behaviour/ecology/molecular biology

predator–prey interactions, infectious haematopoietic necrosis virus, migratory culling, migration ecology, predation risk, disease ecology

**Author for correspondence:**
Nathan B. Furey
e-mail: nathan.furey@unh.edu

# Infected juvenile salmon can experience increased predation during freshwater migration

Nathan B. Furey[1], Arthur L. Bass[2], Kristi M. Miller[3], Shaorong Li[3], Andrew G. Lotto[2], Stephen J. Healy[4], S. Matthew Drenner[5,6] and Scott G. Hinch[2]

[1]Department of Biological Sciences, University of New Hampshire, Durham, NH, USA
[2]Pacific Salmon Ecology and Conservation Laboratory, Department of Forest and Conservation Sciences, University of British Columbia, Vancouver, Canada
[3]Fisheries and Oceans Canada, Molecular Genetics Section, Pacific Biological Station, Nanaimo, Canada
[4]Fisheries and Oceans Canada, Science Branch, Pacific Region, 4160 Marine Dr., West Vancouver, British Columbia, Canada
[5]Stillwater Sciences, 555 W. Fifth St, 35th floor, Los Angeles, CA, USA
[6]Marine Science Institute, University of California Santa Barbara, Santa Barbara, CA, USA

NBF, 0000-0002-8584-7889

Predation risk for animal migrants can be impacted by physical condition. Although size- or condition-based selection is often observed, observing infection-based predation is rare due to the difficulties in assessing infectious agents in predated samples. We examined predation of outmigrating sockeye salmon (*Oncorhynchus nerka*) smolts by bull trout (*Salvelinus confluentus*) in south-central British Columbia, Canada. We used a high-throughput quantitative polymerase chain reaction (qPCR) platform to screen for the presence of 17 infectious agents found in salmon and assess 14 host genes associated with viral responses. In one (2014) of the two years assessed (2014 and 2015), the presence of infectious haematopoietic necrosis virus (IHNv) resulted in 15–26 times greater chance of predation; in 2015 IHNv was absent among all samples, predated or not. Thus, we provide further evidence that infection can impact predation risk in migrants. Some smolts with high IHNv loads also exhibited gene expression profiles consistent with a virus-induced disease state. Nine other infectious agents were observed between the two years, none of which were associated with increased selection by bull trout. In 2014, richness of infectious agents was also associated with greater predation risk. This is a rare demonstration of predator consumption resulting in selection

for prey that carry infectious agents. The mechanism by which this selection occurs is not yet determined. By culling infectious agents from migrant populations, fish predators could provide an ecological benefit to prey.

# 1. Introduction

Predators [1], infectious agents [2,3] and their interaction [4,5] play important roles in structuring communities and ecosystems. Both predators and infectious agents can apply strong selection pressures on prey and hosts, altering population-level phenotypes [4,6–8]. Infection can increase predation risk [9–11], presumably due to decreased ability to detect and/or evade predators, and/or increased conspicuousness to predators [12]. Infectious agents also affect animal migrants [13,14], migrations can act to reduce predation [15,16], and a few studies have found infection to increase predation risk of migrants (e.g. Mesa *et al.* [17], Schreck *et al.* [18] and Hostetter *et al.* [19]).

Pacific salmon (*Oncorhynchus* spp.) are among the most studied animal migrants due to their ecological, economic and cultural value. One of the migrations undertaken during the Pacific salmon life cycle is by juveniles, when smolts leave natal freshwater habitats and migrate downstream to the open ocean. Smolts can experience intense predation during downstream migration [20–22]. Recent research has linked smolt migration survival to the presence and/or prevalence of pathogens [23] and external signs of disease [19,24]. However, infection is merely the presence of a pathogen and does not necessarily indicate disease that could facilitate predation, but disease is difficult to assess in the field [25,26], especially when natural mortality is not observable [25].

Transcriptomics continue to be an increasingly valuable tool in linking animal responses to environmental conditions and other factors [27] and has proven to be a highly sensitive indicator in human disease diagnostics [28–31]. Recently, meta-analysis of multi-cohort microarray data based on six acute and chronic viral diseases revealed a panel of biomarkers consistently associated with viral disease development (VDD) in salmon [32]. Validation of the VDD biomarker panel using independent samples from infectious haematopoietic necrosis virus (IHNv) challenge studies performed across multiple salmon species, and field samples diagnosed pathologically with various viral and non-viral diseases showed that accurate classifications differentiating bacterial versus viral diseases and latent infections versus viral disease could be realized with co-activation of as few as seven VDD biomarkers. Moreover, as demonstrated in human diagnostic studies, the molecular panel could identify disease before clinical or morphological evidence can be observed [32,33], and due to the systemic nature of viral infections, worked well across a range of tissues. The VDD technology has been successfully applied to study disease development pathways for Piscine orthoreovirus (PRv) [33] and has led to the discovery of over a dozen novel viruses in salmon [34,35].

Among sockeye salmon (*Oncorhynchus nerka*) populations of the Fraser River watershed in British Columbia, Canada, the population emigrating from Chilko Lake is among the largest and most intensively studied. Each spring, 10–70 million juvenile sockeye salmon smolts leave the lake and migrate downstream through a gauntlet of binge-feeding bull trout [36] and experience high mortality in the clear, slow-moving waters of the Chilko River [37]. Combining acoustic telemetry with non-lethal biopsies and screening for infectious agents revealed a strong link between mortality of migratory smolts and IHNv [23], but the mechanism of mortality was unable to be determined. IHNv is a coldwater virus found in North America, Europe and Asia [38]. IHNv appears most effective at infecting juvenile fish found in freshwater and at temperatures between 10°C and 12°C [38]. In juvenile sockeye salmon, IHNv can be highly pathogenic [39,40], inducing high rates of mortality. It is suggested that sockeye salmon are natural hosts of IHNv [40] and this virus has been present in Chilko Lake for at least several decades [41].

We assess infection-based predation risk of migrant juvenile sockeye salmon (*Oncorhynchus nerka*) smolts by piscivorous bull trout (*Salvelinus confluentus*) in Chilko Lake. We tested smolt tissue samples using TaqMan assays for 17 infectious agents suspected or known to cause disease in salmon [25], including IHNv. We use a subset of high-performing VDD biomarkers to attempt to link predation and infection with genetic markers of active viral disease states [32].

# 2. Methods

## 2.1. Study area and field sampling

Sampling occurred at the Chilko Lake-River outlet in British Columbia, Canada, where sockeye salmon smolts emigrate downstream each spring and the federal fisheries agency (Fisheries and Oceans Canada) installs a river-wide counting fence to estimate outmigrant abundance. To compare infection status between predated and non-predated smolts, individuals were collected from within bull trout stomachs, as well as at random from the emigrant population (details below) between 30 April and 15 May 2014, and between 19 April and 5 May 2015. Bull trout were captured via dip net or hook and line either at (immediately upstream of) the counting fence or in the 1.3 km stretch between the counting fence and lake outlet. Stomach contents from bull trout were collected via gastric lavage. When possible, freshly ingested smolts were individually wrapped in foil and frozen in liquid nitrogen; when this was not possible, smolts were frozen at −20°C for up to 72 h before transferring to liquid nitrogen or a −80°C freezer for long-term storage; our assessments are not expected to be impacted by this short-term storage at −20°C. Non-predated smolts were collected via dip net at the counting fence at night during the outmigration and selected at random from a small plastic wash basin. Totals of 62 (32 predated, 30 not) and 39 (30 predated, nine not) smolts collected in 2014 and 2015, respectively, were selected for pathogen screening. Sample sizes of non-predated smolts in 2015 were low because the second year of the study was opportunistic with limited funding and the field season was shortened by high flows in the Chilko River that affected other active research. We also wanted to focus on infectious agents in predated fish, rather than broadly characterizing the pathogens found in wild sockeye salmon smolts. Every predated smolt was assigned a condition score as a metric for degree of degradation or digestion such that we could assess the potential effects of sample degradation on infectious agents and biomarker expression. Condition scores ranged between zero (no visible signs of digestion) and six (prey item unidentifiable) as in Furey *et al*. [42]. To maximize the condition of smolts assessed, in 2014, only samples with condition scores between zero and two were selected for molecular work. In 2015, only samples with scores between zero and 1.5 were selected.

## 2.2. Laboratory sampling and analyses

In the laboratory, smolts were dissected to remove gill and liver tissues using aseptic technique. Tissue samples were screened for the presence of 17 infectious agents (table 1; electronic supplementary material, table S1), using high-throughput quantitative real-time reverse transcriptase polymerase chain reaction (ht-qRT-PCR). Infectious agents selected are among those known to infect salmonids worldwide. The biomarkers selected are all among those found to be capable of consistently identifying individuals experiencing viral disease [32]. In addition, 14 host genes found to be a high-performing subset of genes capable of consistently distinguishing a fish in an active viral disease state (i.e. VDD) [32] were assessed (electronic supplementary material, table S2). Individuals in a viral disease state demonstrate powerful co-activation of these VDD genes, which can be identified via strong separation along the first axis of multivariate analyses including expression of groups of VDD genes [32]. One of these assays, HERC6, had low assay efficiency and was excluded, leaving 13 host genes. Three liver samples from predated smolts were removed from analyses due to low reference gene expression.

## 2.3. Molecular assessment of infectious agents and smolt gene expression

PCR was conducted on the Fluidigm BioMark™ HD nanofluidic platform (Fluidigm Corp., South San Francisco, USA). Gill and liver tissues were homogenized separately in TRI reagent (Ambion Inc., Austin, TX) and 1-bromo-3-chloropropane was added to the homogenate. Total RNA was extracted by methods previously described [25,43] using MagMAX™-96 for Microarrays Total RNA Isolation Kits (Applied Biosystems, Foster City, CA, USA) with a Biomek FXP automated liquid-handling instrument (Beckman Coulter, Indianapolis, IN, USA) according to the manufacturer's instructions. The Biomek FXP was also used to automatically normalize total RNA to 1.0 µg. cDNA was synthesized from normalized RNA using SuperScript VILO MasterMix (Invitrogen, CA, USA) following manufacturer's instructions. The nanolitre volume used for each qPCR reaction on the BioMark necessitates a pre-amplification step. Thus, 1.25 µl of cDNA from each sample was

**Table 1.** List of infectious agents assessed in sockeye salmon smolts using qRT-PCR, the percentage of positives recorded across year–tissue combinations (±s.e.), and the odds-ratio of each infectious agent being found in a predated smolt over a non-predated smolt are given. Odds ratios in italics and noted with an asterisk (*) indicate significant Fisher exact test (fdr-corrected $p < 0.05$). Sample sizes are as follows for each year–tissue combination: 2014 predated ($n = 32$ for gills, $n = 31$ for livers); 2014 not predated ($n = 30$ each tissue); 2015 predated ($n = 30$ each tissue); 2015 not predated ($n = 9$ each tissue). Prevalence rates assessed from mixed-tissue samples of Chilko sockeye salmon smolts collected via the Strategic Salmon Health Initiative (SSHI) are given for 2012 ($n = 54$–56 smolts for each assay), 2013 ($n = 85$–89) and 2014 ($n = 21$–30) for comparison.

| infectious agent | assay name | agent | per cent positives (predated/not predated) | | | | odds-ratio (predated over not predated) | | | | prevalence (SSHI samples) | | |
|---|---|---|---|---|---|---|---|---|---|---|---|---|---|
| | | | 2014 gill positives | 2014 liver positives | 2015 gill positives | 2015 liver positives | 2014 gill odds-ratio | 2014 liver odds-ratio | 2015 gill odds-ratio | 2015 liver odds-ratio | 2012 | 2013 | 2014 |
| *Candidatus* Branchiomonas cysticola | c_b_cys | bacteria | 100(±0.0)/ 100(±0.0) | 96.8(±3.2)/ 93.3(4.6±) | 100(±0.0)/ 100(±0.0) | 66.7(±9.1)/ 100(±0.0) | | 2.1 | | 0.0 | 98.2 | 100 | 100 |
| *Ceratomyxa shasta* | ce_sha | myxozoan | | | | | | | | | 0.0 | 0.0 | 0.0 |
| *Dermocystidium salmonis* | de_sal | fungus/ protozoan | | | | | | | | | 1.8 | 0.0 | 0.0 |
| *Flavobacterium psychrophilum* | fl_psy | bacteria | 87.5(±5.8)/ 70(±8.4) | 16.1(±6.6)/ 10(±5.6) | 76.7(±7.7)/ 44.4(±15.0) | 14.8(±6.8)/ 11.1(±6.4) | 2.9 | 1.7 | 3.9 | 1.4 | 5.4 | 6.7 | 17.2 |
| *Candidatus* Syngnamydia salmonis | sch | bacteria | | | | | | | | | 0.0 | 0.0 | 0.0 |
| *Ichthyophthirius multifiliis* | ic_mul | ciliate | 28.1(±7.9)/ 6.7(±4.6) | | 16.7(±6.8)/ 0.0(±0.0) | 7.4(±5.0)/ 0.0(±0.0) | 5.3 | | | | 3.6 | 20.0 | 23.1 |
| Infectious haematopoietic necrosis virus | ihnv | virus | 87.5(±5.8)/ 20(±7.3) | 35.5(±8.6)/ 3.3(±4.0) | | | *25.8** | *15.3** | | | 0.0 | 0.0 | 3.3 |
| *Loma salmonae* (*Loma* spp.) | lo_sal | microsporidium | | | | | | | | | 0.0 | 0.0 | 0.0 |
| Pacific salmon parvovirus | pspv | virus | 9.4(±5.2)/ 0.0(±0.0) | 80.6(±7.1)/ 66.7(±9.2) | 10(±5.5)/ 11.1(±5.9) | 48.1(±9.6)/ 22.2(10.4±) | | 2.1 | 0.9 | 3.2 | 78.6 | 96.6 | 93.3 |
| *Paranucleospora theridion* | pa_ther | microsporidium | | | | 0.0(±0.0)/ | | | | | 0.0 | 1.1 | 0.0 |
| *Parvicapsula minibicornis* | pa_min | myxozoan | | | | 11.1(±5.9) | | | | | 0.0 | 1.1 | 0.0 |

(*Continued.*)

**Table 1.** (*Continued.*)

| infectious agent | assay name | agent | per cent positives (predated/not predated) | | | | odds-ratio (predated over not predated) | | | | prevalence (SSHI samples) | | |
|---|---|---|---|---|---|---|---|---|---|---|---|---|---|
| | | | 2014 gill positives | 2014 liver positives | 2015 gill positives | 2015 liver positives | 2014 gill odds-ratio | 2014 liver odds-ratio | 2015 gill odds-ratio | 2015 liver odds-ratio | 2012 | 2013 | 2014 |
| *Parvicapsula pseudobranchicola* | pa_pse | myxozoan | | | | | | | | | 0.0 | 0.0 | 0.0 |
| *Piscichlamydia salmonis* | pdh_sal | bacteria | 15.6(±6.4)/ 0.0(±0.0) | 3.2(±3.2)/ 0.0(±0.0) | | | | | | | 0.0 | 0.0 | 0.0 |
| Piscine reovirus | prv | virus | | | | | | | | | 0.0 | 0.0 | 0.0 |
| *Tetracapsuloides bryosalmonae* | te_bry | myxozoan | | 6.5(±4.4)/ 0.0(±0.0) | | | | | | | 0.0 | 3.4 | 16.7 |
| Ricksettia-like organism | rlo | | 3.1(±3.1)/ 0.0(±0.0) | | | | | | | | 0.0 | 1.1 | 0.0 |
| *Yersinia ruckeri* | ye_ruc_glnA | bacteria | 9.4(±5.2)/ 0.0(±0.0) | | 13.3(±6.2)/ 0.0(±0.0) | | | | | | 0.0 | 0.0 | 0.0 |

pre-amplified with primer pairs corresponding to all assays in a 5 µl reaction volume using TaqMan Preamp Master Mix (Life Technologies) (see Miller et al. [32]). Unincorporated primers were removed using ExoSAP-IT High-Throughput PCR Product Cleanup (MJS BioLynx Inc., ON, CAN), and samples were diluted 1 : 5 in DNA Suspension Buffer. The assay mix was prepared containing 9 µl primers and 2 µl probes for the TaqMan assays.

All assays were run in duplicate on the BioMark Dynamic Array. A serial dilution of artificial positive constructs (APC clones) of all infectious agent assays was run as six samples. This serial dilution allowed for the calculation of assay efficiency, and the copy numbers of the interest targets. The APC clones contain an additional probe (VIC) that allows for the detection of potential contamination caused by these highly concentrated samples. For biomarkers, assay efficiency was assessed using a five-sample serial dilution of pooled, pre-amplified samples. The serial dilution was created by diluting the pooled sample in DNA suspension buffer. Three reference gene assays (S100 calcium binding protein [786d16.1P], coiled-coil domain-containing protein 84 [COIL], and 39S ribosomal protein L40, mitochondrial precursor [MrpL40]), were included to assess sample quality and normalize biomarker gene data. A 5 µl sample mix was prepared [2.5 ul of TaqMan Gene Expression Master Mix (Life Technologies), 0.25 ul of 20X GE Sample Loading Reagent (Fluidigm), 2.25 ul of pre-amplified cDNA], which was added to each assay inlet of the array following manufacturer's recommendations. After loading the assays and samples into the chip by an IFC controller HX (Fluidigm), PCR was performed with the following conditions: 50°C for 2 min, 95°C for 10 min, followed by 40 cycles of 95°C for 15 s and 60°C for 1 min.

Cycle threshold (Ct) was determined using the Biomark Real-Time PCR analysis software. Reaction curves for each positive sample-assay combination were visually evaluated for abnormal curve shapes, close correspondence between replicates, and the presence of APC contamination as indicated by VIC positives. Using R [44], efficiency was calculated for each assay, results where only one duplicate was positive for a sample-assay combination were removed, limit of detection thresholds (above which, samples were considered negative [32]) applied, VIC positive samples removed and duplicates averaged. Ct scores for infectious agents were converted to RNA copy number per well using the standard curve for each assay.

## 2.4. Reference gene performance and sample degradation potential

For all samples, we assessed the performance of three reference genes (S100 calcium COIL, 786d16.1P and MrpL40) that should be expressed at relatively similar levels among all samples. We wanted to examine their performance due to the possibility of samples degrading while in a bull trout's stomach (which would only affect predated samples). Samples were removed if expression of any reference gene was 1.5 times the interquartile range below the first quartile of gene- and tissue-specific values (e.g. an outlier). Only four samples, one liver sample collected in 2014 and three liver samples collected in 2015, met this criterion and were removed. To further assess the potential effects of sampling in both predated and non-predated samples, we visually assessed the expression of the three reference genes between predated statuses for all year–tissue combinations.

## 2.5. Data analyses

To determine if infectious agents were more prevalent (i.e. greater percentage of samples that were positive) in predated smolts than in smolts caught by dip net, a Fisher's exact test was conducted for each pathogen for each tissue and year, along with the calculation of the odds ratio for infection in predated versus non-predated samples. We used a false-discovery-rate adjusted $\alpha = 0.05$ to assess significance. For any infectious agent found to be more prevalent in predated samples, we determined if fish size (fork length; FL) varied between infection-positive and infection-negative fish using a t-test. When FL was not measured directly, it was estimated from total length (TL) or post-orbital hypural (POH) measurements via regression (Furey 2016, unpublished data). To determine if predated smolts had a greater diversity of infectious agents within their tissues, the Shannon diversity index per sample was calculated using the 'diversity' function in the vegan package [45] in R [44] and compared via a Mann–Whitney U-test on ranks.

To further characterize the relationships among infection, fish length, tissue sampled, and predation, generalized models (GLM) were used. Four global models were constructed, one for each year–tissue combination due to the imbalance in sample sizes of predated and non-predated fish between years and some infectious agents being present in one year and not the other (see Results). Predation status

was the response variable, with smolt FL and presence or absence of infectious agents as explanatory variables. In 2015, 12 smolts did not have any lengths recorded, and these fish were removed from GLM analyses. Two age classes emigrate from Chilko Lake, British Columbia. Age-1 smolts constitute on average approximately 96% of the migrating population, while age-2 are substantially larger but make up approximately 4% of the migration [46]. Of the 32 predated smolts assessed in 2014, eight of them were age-2 (classified as those greater than 116 mm FL; Brian Leaf, DFO, 2016, pers. comm.), all of which were predated. In response, age-2 smolts were removed from 2014 GLM analyses, as they were only present in the predated group (and thus age and FL were confounded). Only infectious agents that were detected in at least one predated and one non-predated smolt for a given tissue–year combination were included to prevent unrealistic logistic regression coefficients. Infectious agents that were found among all samples were also not included. Global models were constructed in R [44]. Candidate models were ranked via the Akaike information criterion corrected for small sample sizes (AICc) using all-subsets regression via the MuMIn package [47] in R [44]. To prevent overfitting due to our small sample sizes, the maximum number of parameters in each candidate model was limited to three (not including the intercept). The model with the lowest AICc was considered further as the most parsimonious and we present all models with $\Delta$AICc < 3. However, we also present averaged coefficients among models with $\Delta$AICc < 3. Coefficients were weighted by AICc weight, recalculated among models with $\Delta$AICc < 3 (table 2, right-hand column). In addition, averaged coefficients were calculated using the zero method, such that if a variable was not retained in an averaged model, a zero was included in the averaging, reducing the effect size.

Ct scores were transformed using a standard curve of known infectious agent RNA concentrations to represent RNA copy number per PCR well. Principal components analysis (PCA) was used to visualize variability in VDD gene expression among samples. Separate PCAs were run for each year–tissue combination (four in total). PCA results were assessed visually to determine relationships between VDD gene expression and both predation and infection status, focusing on groupings of samples along the first two axes. All analyses were completed in R 3.5.1 [44], with PCAs conducted with the 'prcomp' function.

# 3. Results

## 3.1. Infectious agents

Among the 17 infectious agents screened for, 10 (including IHNv) were found to be in sampled smolts between the two years and tissues (table 1). IHNv was only observed in 2014, but its prevalence dramatically differed between predated (87.5% in gill and 35.5% in liver) and non-predated (20% in gill and 3.3% in liver) samples. The odds of IHNv infection in gill was 25.8 times greater for predated than non-predated smolts (Fisher exact test, fdr-corrected $p < 0.0001$) and 15.3 times greater in liver (Fisher exact test, fdr-corrected $p = 0.007$). IHNv prevalence did not differ between age-1 and age-2 predated smolts (electronic supplementary material). $t$-tests comparing mean fork length between fish positive and negative for IHNv in 2014 found no significant difference in size in either gill ($p = 0.75$) or liver tissues ($p = 0.86$). No pathogen aside from IHNv was found to be statistically more prevalent in predated samples than non-predated. 'Candidatus Branchiomonas cysticola' was found in approximately 94% of all samples. Although not significantly so, most observed infectious agents were observed at higher prevalence in predated samples than not, with Flavobacterium psychrophilum being 1.4–3.9 times more likely to be found in predated smolts among all tissue–year combinations. Ichthyophthirius multifiliis was not found in any liver samples in 2014, (and only in two liver samples in 2015, both predated), but in both years of gill samples, the agent was consistently found more often in predated samples. No pathogen was found to be more prevalent in non-predated samples in more than one tissue–year combination (table 1); in the three instances where a pathogen was found more often in non-predated samples, none were statistically significant (all fdr-corrected $p > 0.05$). The Shannon diversity index of infectious agents was significantly greater in predated samples for both gill (Mann–Whitney $U$-test; $p < 0.001$) and liver (Mann–Whitney $U$-test; $p = 0.02$) tissues in 2014 (figure 1). In 2015 samples, the diversity index did not vary between predated and non-predated samples in either tissue (Mann–Whitney $U$-test; $p > 0.05$).

Use of GLMs revealed similar, but also additional, relationships between infection and predation risk (table 2) to the pathogen-by-pathogen approach. IHNv was retained in all 2014 models with $\Delta$AICc < 3, for both gill and liver, with increased predation risk associated with infection. However, the two

**Table 2.** Summary of generalized linear models (GLMs) describing relationships between predation status (binomial) and the presence of infectious agents and fork length (FL). Candidate models are ranked by AICc, and only models with ΔAICc < 3 are shown. The top-ranked model is in italics. First numeric value given for each model is the intercept, and coefficients are shown for each explanatory variable. Infectious agents are labelled as per their assay name (table 1). Positive coefficients indicate increased probability of predation (negative coefficients associated with reduced predation risk). AICc weights are shown for each model, calculated for both all models, but also when only considering models with ΔAICc < 3 (the latter of which was used for calculating model averaged coefficients, shown below candidate models).

| individual model | AICc | ΔAICc | AICc Weight (all models) | AICc Weight (ΔAICc < 3) |
|---|---|---|---|---|
| **2014 - gill** | | | | |
| *~ −2.54 + ihnv(+3.64) + ic_mul(+2.20)* | *46.6* | *0* | *0.33* | *0.38* |
| ~ −1.75 + ihnv(+4.53) + ic_mul(+2.44) + fl_psy(−1.70) | 47.4 | 0.8 | 0.22 | 0.25 |
| ~ −2.08 + ihnv(+3.52) | 48.5 | 1.91 | 0.13 | 0.14 |
| ~ +1.63 + FL(−0.04) + ihnv(+3.60) + ic_mul(+2.07) | 48.6 | 1.95 | 0.13 | 0.14 |
| ~ −1.39 + ihnv(+4.21) + fl_psy(−1.39) | 49.6 | 2.94 | 0.08 | 0.09 |
| averaged model | | | | |
| ~ −1.58 + ihnv(+3.89) + ic_mul(+1.73) + fl_psy(−0.55) + FL(−0.006) | | | | |
| **2014 - liver** | | | | |
| *~ +12.29 + FL(−0.14) + ihnv(+3.56)* | *61.1* | *0.00* | *0.53* | *0.61* |
| ~ +12.22 + FL(−0.14) + ihnv(+3.55) + pspv(+0.16) | 63.3 | 2.29 | 0.17 | 0.19 |
| ~ +12.31 + FL(−0.14) + ihnv(+3.57) + fl_psy(−0.15) | 63.4 | 2.32 | 0.17 | 0.19 |
| averaged model | | | | |
| ~ 12.28 + FL(−0.14) + ihnv(+3.56) + pspv(+0.03) + fl_psy(−0.03) | | | | |
| **2015 - gill** | | | | |
| *~ +22.35 + FL(−0.28) + fl_psy(+2.38)* | *31.4* | *0.00* | *0.59* | *0.63* |
| ~ +24.52 + FL(−0.31) + fl_psy(+2.66) + pspv(−1.10) | 33.4 | 2.08 | 0.21 | 0.22 |
| ~+18.36 + FL(−0.22) | 34.2 | 2.88 | 0.14 | 0.15 |
| averaged model | | | | |
| ~ +22.24 + FL(−0.28) + fl_psy(+2.09) + pspv(−0.24) | | | | |
| **2015 - liver** | | | | |
| *~+18.00 + FL(−0.22)* | *32.8* | *0.00* | *0.38* | *0.40* |
| ~+18.77 + FL(−0.23) + pspv(+1.55) | 32.9 | 0.09 | 0.36 | 0.38 |
| ~+18.06 + FL(−0.22) + fl_psy(+0.18) | 35.3 | 2.47 | 0.11 | 0.12 |
| ~+18.93 + FL(−0.24) + pspv(1.66) + fl_psy(+0.91) | 35.4 | 35.4 | 0.11 | 0.11 |
| averaged model | | | | |
| ~ +18.40 + FL(−0.22) + pspv(+0.77) + fl_psy(+0.12) | | | | |

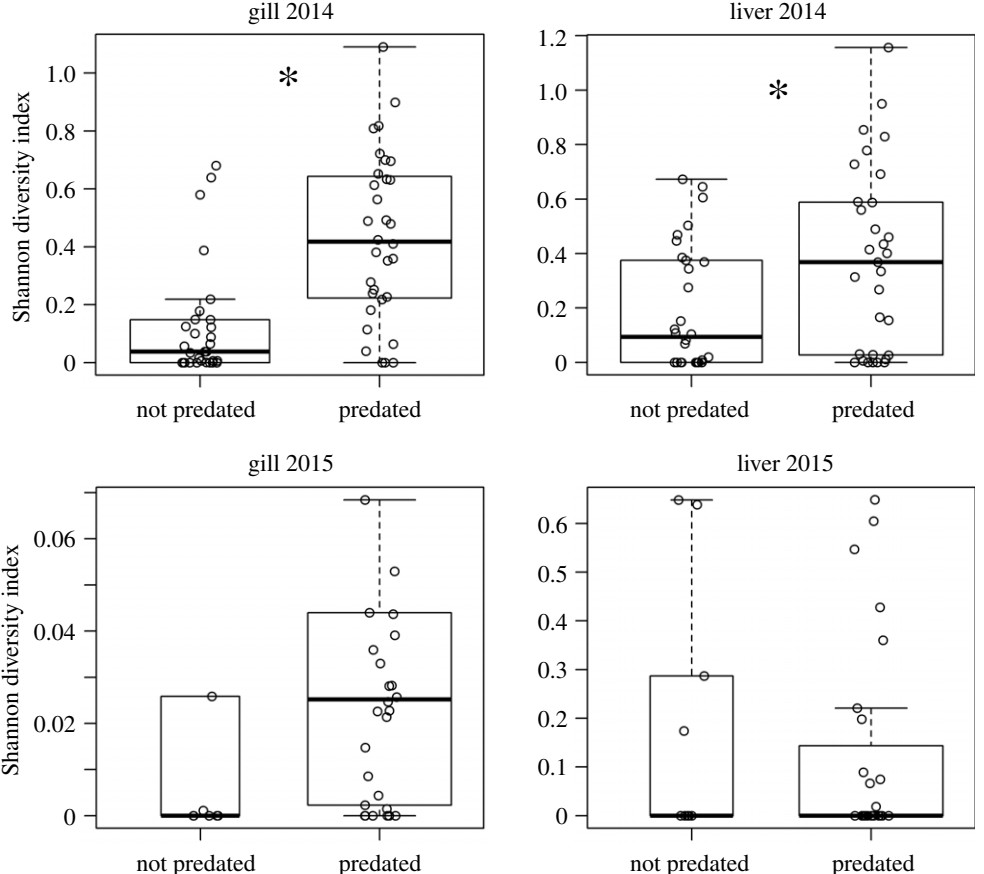

**Figure 1.** Shannon diversity index of infectious agents found in gill and liver tissue of sockeye salmon smolts between those predated and not predated by bull trout. Asterisks indicate a significant difference in median pathogen richness between predated and non-predated groups (Mann–Whitney U-test, $\alpha = 0.05$).

top-ranked 2014-gill models also revealed a potential increased probability of predation for smolts infected with *Ichthyophthirius multifiliis*. *Ichthyophthirius multifiliis* was also more prevalent in predated samples in 2015, but because this infectious agent was absent from non-predated samples in this year (table 1), it was not included in GLMs. The presence of two other pathogens, *Flavobacterium psychrophilum* and Pacific salmon parvovirus, was also retained in some models with ΔAICc < 3, but in general, their coefficients were smaller, and the signs of their coefficients were inconsistent (implying the agent could be associated with other increased or decreased predation risk; table 2).

## 3.2. Fork length and age

Among GLMs, the 2014-liver models and all 2015 models suggested that smaller fish were at greater risk of predation (negative FL coefficient; table 2). This relationship was consistent among year–tissue combinations, with all models ΔAICc < 3 containing FL, including the top models. In 2014 samples, mean FL of smolts did not differ between IHNv+ and IHNv- smolts, in both gill ($t = 0.46$, d.f. = 39, $p = 0.64$) and liver ($t = -0.12$, d.f. = 40, $p = 0.90$) tissues. Similarly, the prevalence of IHNv (0.875) was the same between age-1 (21 of 24) and age-2 (seven of eight) predated smolts in 2014, and thus the inclusion of age-2 fish in our predated sample did not bias IHNv prevalence in predated fish.

## 3.3. Gene expression

PCAs on 2014 VDD gene expression data (the year in which IHNv was present) revealed three smolts that exhibited strong separation along the first PC axis (most positive PC1; figure 2). This strong separation was apparent in both gill and liver tissues (figure 2), and these three same smolts had among the highest tissue-specific loads of IHNv (figure 2). An additional fourth gill 2014 sample exhibited the same strong separation on the first PC axis, but was not included in liver analyses due

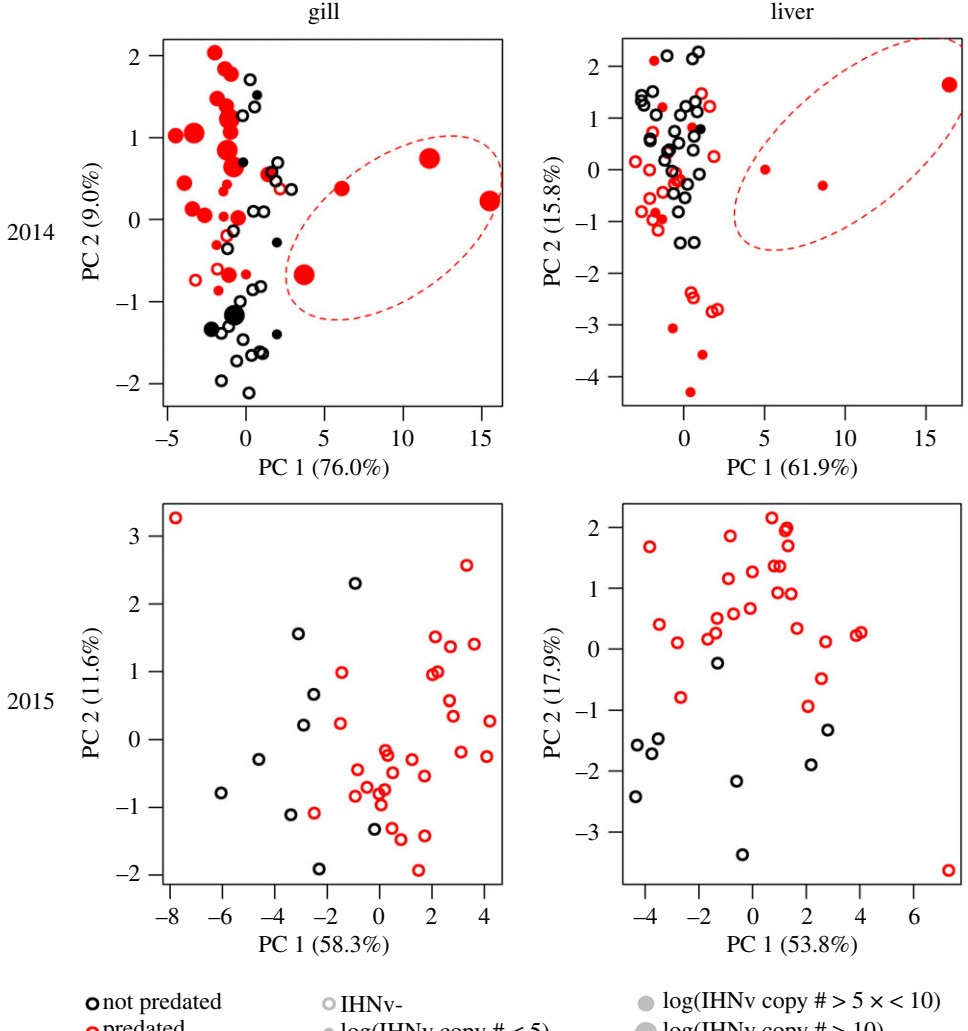

**Figure 2.** PCA of gene expression of 13 genes used in diagnosing viral disease development [40] in sockeye salmon smolt samples from 2014 and 2015 and in gill and liver tissues. Circle size symbolizes IHNv loads (represented as the log of the estimated copy number + 1). Red ellipses enclose the same samples (three in liver samples, with an additional fourth in gill samples) that separate via the first PC axis and have high IHNv loads (IHNv+), potentially indicative of an active disease state. Percentages in parentheses indicate the per cent variability explained among gene expression by that specific axis.

to poor reference gene performance. Aside from these individuals, PCA in both years also demonstrated further shifts in VDD gene expression between predated and non-predated smolts in at least one of the first two PC axes, regardless of year or tissue (figure 2). There was some tissue- and year-specific variability; separation for 2015 gill samples was most clearly along PC1, while the other year–tissue combinations (aside from the three high-IHNv-loaded individuals) demonstrated stronger shifts along PC2 (figure 2).

## 3.4. Sample degradation potential

All three reference genes demonstrated higher expression (lower Ct scores) in non-predated samples in gills for both years (786d16.1P was significantly different in both years, COIL significantly different in 2014, MrpL40 not significantly different in either year; *t*-test, $\alpha = 0.05$; figure 3). Conversely, all three reference genes demonstrated lower expression (higher Ct scores) in non-predated samples in livers in both years (COIL significantly so in both years, MrpL40 in 2015, and 786d16.1P in neither; figure 3).

There was no significant relationship between IHNv loads and condition score for predated, IHN+ smolts for both gill (Pearson correlation = 0.31, d.f. = 26, $t = 1.68$, $p = 0.10$) and liver (Pearson correlation coefficient = 0.22; d.f. = 10, $t = 0.73$, $p = 0.48$). However, IHNv+ gill samples came from predated smolts

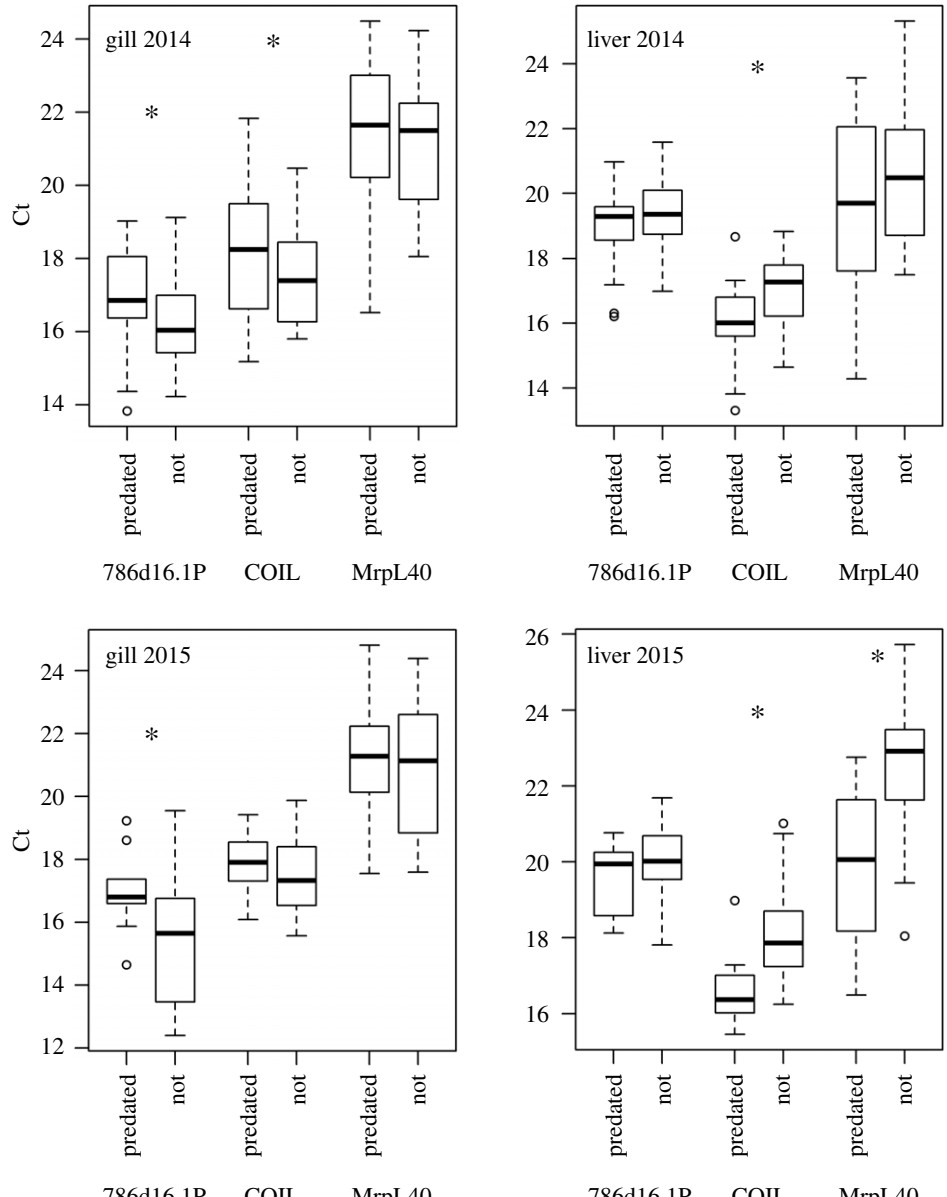

**Figure 3.** Expression levels (via cycle threshold [Ct] values) of three reference genes between years and tissues of juvenile sockeye salmon smolts. 'Predated' indicates predated samples, 'Not' indicates control, or non-predated, sample. Asterisk (*) indicates significant difference in Ct score between predated and non-predated samples for a given reference gene (t-test, $\alpha = 0.05$).

with a significantly higher condition score (i.e. more digested) than predated smolts that were IHNv- (mean score IHNv+ = 1.4, mean score IHNv- = 0.5, t-test, $t = 2.60$, d.f. = 30, $p = 0.01$). However, condition scores did not differ between IHNv+ and IHNv- predated smolt samples in liver samples (mean score IHNv+ = 1.5, mean score IHNv- = 1.1, t-test, $t = 1.60$, d.f. = 29, $p = 0.12$).

## 4. Discussion

IHNv-positive smolts in 2014 were 16–25-times more likely to be predated than not. It is uncommon for studies to make direct links between infection and predation risk outside of experimental settings (but see [9–11,17,25,48]. Field studies on infection-based risk for fishes have focused on avian predators [18,25]. Miller *et al.* [25], used an approach similar to ours to demonstrate pathogen-based predation risk for wild salmon, with rhinoceros auklets (*Cerorhinca monocerata*) feeding more heavily on marine sockeye salmon smolts infected with *Parvicapsula* spp. parasites. Although not focused on predation, Jeffries *et al.* [23] found within our study system that most (greater than 80%) IHNv-positive Chilko sockeye

salmon smolts tracked with acoustic telemetry perished early in the migration, suggesting an association between IHNv infection and smolt mortality, and our results indicate that predation is the probably mechanism for at least a portion of this mortality.

IHNv is a single-stranded RNA virus that generates an acute, systemic disease that causes necrosis of haematopoietic tissues of the kidney and spleen, as well as damage to several other organs [49]. For juvenile sockeye, virulence is high [39] and can result in high mortality [40] 4–20 days after exposure [50], but outbreaks are generally limited to cooler waters below 15°C [38]. IHNv's presence in Chilko Lake has been known for more than 40 years [41]. How infection of IHNv results in increased predation by bull trout remains unclear. It is assumed that these infectious agents either reduce a smolt's probability of escaping a predation attempt when targeted [17], or increase the predator's propensity to target the smolt. Either possibility would probably rely upon changing body coloration [51] or changing swimming behaviour or performance that can occur with infection [52,53]; IHN can result in lethargy, hyperactivity, or erratic swimming [53]. Further work, such as experimental swim trials or high-resolution tracking, is needed to determine the behavioural consequences of infection in migratory smolts, and how this might result in increased predation risk. Such research would further develop our understanding of how infections and movements, including migrations, interact to affect individuals, populations and communities [14,54].

Although IHNv demonstrated the strongest links between predation risk and infection, *Ichthyophthirius multifiliis* was also associated with increased predation risk via GLMs for 2014 gill samples. *Ichthyophthirius multifiliis* was only found in predated samples in both gill and liver tissues in 2015, and thus an odds ratio could not be calculated and was not included in 2015 GLMs, but in 2014 gill samples, this infectious agent was associated with an approximately fivefold increase in predation risk. This freshwater ciliate can induce mortality in fishes [55,56], including documented epizootics in a wild population of spawning Fraser River sockeye salmon [57]. The parasite targets epithelial tissue, and damage to gills leads to oxygen starvation and acidosis [56]. Thus, *I. multifiliis* can reduce swimming capacity of hosts [58]. In contrast to IHNv, the likelihood of infection with this globally distributed parasite increases with rising water temperature (as a result of reduced generation time; [56,59]).

Both *Flavobacterium psychrophilum* and Pacific salmon parvovirus were retained in GLMs, but inconsistently. *Flavobacterium psychrophilum* is a bacterium that causes disease in a wide variety of salmonids and can induce high rates of mortality, but pathogenicity varies widely among strains [60]. We found *F. psychrophilum* to be more prevalent in predated samples consistently among tissues and years, but not significantly so. Furthermore, GLM coefficients for this infectious agent were negative in 2014, suggesting predation risk was lower with infection. However, these negative coefficients only occur in models that also contain IHNv as an explanatory variable (which was retained in all high-ranking models); thus, these coefficients refer to the impact of *F. psychrophilum* on predation risk in the absence of IHNv. These small but negative coefficients are thus probably due to small sample sizes and we caution overinterpretation. Similar to our Fisher exact tests, it has been found to be weakly, but not significantly, associated with mortality in tagged Chilko sockeye salmon smolts [61]. Pacific salmon parvovirus is a recently discovered virus [43] and it is unclear if and how it causes disease [25], but has been observed in sockeye salmon [32,62,63]. However, Stevenson *et al.* [61] did not observe this virus in Chilko sockeye salmon smolt gill tissues.

The presence of an infectious agent, without an indication of tissue damage or an immunological response (such as the VDD gene panel used in this study), is not evidence of infectious disease. Therefore, unsurprisingly, most of the infectious agents detected in this study were not associated with increased predation risk. Furthermore, the virulence of an infection is dependent upon the interaction of aspects of the host, its environment and the pathogen. Salmon populations that have coevolved with endemic pathogens may be immunologically equipped to resist physiological impairment [64] and some pathogens may disrupt homeostasis primarily in the context of environmental stressors, a pertinent example being the importance of cool temperatures for IHNv virulence [40].

In addition to the prevalence of specific agents, the diversity of infectious agents detected was higher in predated samples in both tissues in 2014. Similarly, rhinoceros auklets fed more heavily on sockeye salmon smolts with higher pathogen richness [25]. Although the mechanism for a correlation between pathogen diversity and predation status is unclear, we hypothesize that smolts with greater diversity of infectious agents are probably physiologically compromised. Although diversity metrics (or other metrics such as relative infection burden [43]) can describe the variability in infections in terms of the presence and load of multiple infections, infectious agents can interact in complex ways. In certain

circumstances co-infection can exacerbate existing or generate new physiological issues for the host [65,66] or even mediate impacts through competitive or antagonistic interactions [67–69]. Thus, further work should focus not only on specific infections or the number of unique infections, but also the combination of infections and their loads.

Regardless of the mechanism, we provide evidence that infections can increase predation risk of fish in the wild. Predation on juvenile salmonids has long been of interest, with research focused on quantifying the number of salmon lost via avian [70,71] predators and piscivorous fishes [72,73] alike. However, it appears in this system that the impacts of predation by bull trout and infection are not additive sources of mortality, but rather compensatory. There is increasing recognition that predators of salmon exert selective pressures [19,25,74], but it remains difficult to quantify the interactions among various biological and environmental conditions influencing mortality [25].

Our assessment of infectious agent influences on predation risk is dependent upon multiple assumptions, including that once ingested, an infected smolt cannot infect others. IHNv-infected smolts, however, were in worse condition (a proxy for longer duration in the gut) than those that were not infected (in gill samples, but not liver samples), which may be evidence of transmission post-ingestion. If cross-contamination of IHNv within the gut does occur, it could be through the gills, which were the only externally exposed tissue sampled. As IHNv can be present in mucus [75], it is plausible that cross-contamination could occur (subsequently increasing the prevalence of infectious agent-positive fish in the predated sample). Restricting sampling to only internal organs in future studies could minimize this risk. With cessation of circulation post-mortem within the fish, we feel it is highly unlikely that an infection could travel between gills and liver once in the bull trout's stomach. Cross-contamination after ingestion would be more likely if infectious agents could persist and proliferate after host death. Stomach acid, however, is a hostile environment that is thought to have evolved in vertebrates not only to aid digestion, but to protect against infectious agents [76,77], which would help to prevent productivity after ingestion. It is also possible that we observed greater prevalence of IHNv in gills rather than livers because heavy infections in the gills represented a more developed infection where the virus can be detected in all tissues, if the liver degrades more quickly post-mortem. Examining multiple tissues simultaneously may also assist in determining infection or disease progression.

Another assumption of our study is that IHNv is not transferred from bull trout to ingested smolts. Although IHNv can infect a variety of North American salmonids [78], to our knowledge, it has never been documented in bull trout, albeit implicated in a historical population collapse in Lake Chelan, Washington [79]. Susceptibility to IHNv is species- and experience-dependent, with other chars exhibiting more resilience than sockeye salmon [78]. If bull trout exhibit similar resilience, it seems likely that their infection rates and loads would be low relative to those observed in sockeye salmon smolts. If bull trout are susceptible to IHNv or any other screened infectious agent, it is certainly feasible for these fish to become infected due to repeated exposures via feeding on smolts during the outmigration. Ingestion of a virus can possibly result in infection transmission [80], leading to concerns over the use of wild baitfish in hatcheries or moving baitfish into new systems [81]. It remains unknown, however, if the ingestion of a smolt would provide an appropriate mechanism for infectious agent transfer from bull trout to smolts, and thus further research could address the validity of this assumption. Regardless, our work presents compelling evidence for the influence for fish health to impact predation risk.

Lastly, IHNv infection does not appear to be confounded by smolt size or age. IHNv affects fish quickly [40], and thus feeding might not be impacted for a long enough duration to generate size differences among infected and uninfected smolts. Similarly, IHNv had equal prevalence in predated samples between the two age classes of smolts emigrating the lake. Thus, IHNv infection probably affects predation risk independent of size, which commonly correlates with survival in juvenile fishes [82].

Even though IHNv was not confounded by size, our analyses found evidence of size-based selection, with bull trout consuming smaller fish, supporting earlier findings in this system [42]. Increased size of fish can both reduce potential gape-limited predators and improve ability to evade predators [82]. Bull trout are probably not affected by gape, and thus size-based predation risk is probably due to increased swimming performance of larger smolts. Smaller sockeye salmon smolts are also disproportionately fed upon by rhinoceros auklets in the marine environment [74], and thus larger smolt sizes may continually be selected for throughout both freshwater and marine portions of the outmigration. However, we acknowledge our sample size is small for investigating size-based predation risk as this paper focuses more on the role of infection.

Gene expression of markers shown to be predictive of VDD [32] differed between predated and non-predated smolts. In particular, three individuals with high IHN loads in 2014 separated clearly along the

first PC axis in both tissues (and a fourth gill sample), a signature observed in other IHNv-infected fish known to be in a viral disease state [32]. Thus, these individuals, all predated, were probably experiencing consequences of disease, an anecdotal but rare link between disease and predation. There was also some separation between other predated and non-predated individuals via PCA (i.e. 2015 samples when IHNv was not present) that could possibly be due to an undetected infectious agent. However, we hesitate to attribute these differences to predation selection, as these differences were of smaller magnitude, and we cannot discount the possibility that gene expression was affected by sample degradation as IHNv+ smolts were in worse condition than those IHNv- (see below). However, we are confident that the strong response from the four fish with high IHNv loads is not due to degradation, as these samples separate from non-predated from other predated samples in the opposite direction along the first PC axis and to a much larger degree (we also observe strong separation when we conduct a PCA on the predated samples only, providing further evidence of a biologically relevant signal; electronic supplementary material, figure S1). Recent work assessing gene expression in gill biopsies on smolts tracked with acoustic telemetry found high IHNv loads to be associated with VDD genes and the first PC axis, but did not find IHNv presence to correlate with survival [61], unlike this study and Jeffries *et al.* [23]. However, Stevenson *et al.* [61] tagged fewer fish with biopsies relative to Jeffries *et al.* [23] and still found age-2 fish that perished in the first 14 km of migration to have high PC1 scores that were associated with elevated IHNv loads [61]. Thus, more work is needed to determine the dynamics of IHNv in the system and the interannual variability in its impacts on smolts.

Reference genes demonstrated that predated samples had lower expression that non-predated samples in gills, but higher expression in liver, although most values were highly overlapping. Although it is difficult to explain why one tissue would react differently than the other regarding gene expression, the lower expression of predated gill samples could be the result of sample degradation. The gills, being an external tissue, would be more exposed to the bull trout's stomach acids and digestive processes than the liver tissue. Sample degradation, or any factor that would result in a shift of gene expression between predated and non-predated samples, would affect our ability to test for predation-based impacts. For example, we see consistent shifts in gene expression based on predation status using PCA, but we cannot demonstrate that these differences are not due to sample degradation alone. The separation between predation statuses apparent via PCA could be attributed to differences in gene performance in the assays or could reflect cellular post-mortem transcriptional shifts, which have been documented to occur in zebrafish, mice and humans [33,83]. However, we see much larger separation in multivariate space regarding VDD gene expression in four samples with high IHNv loads in 2014, that load within the PCA in an opposite direction from other predated samples. We also still see strong separation of these same individuals along the first PC axis when conducting a PCA on only predated fish, indicating unique gene expression regardless of predation status (electronic supplementary material, figure S1). This panel has also been effectively applied to recently dead and live sampled farmed salmon to differentiate fish in an active viral disease state, with findings validated through pathology, providing evidence that these signatures are retained after death [33]. Other recent works suggest that RNA can indeed remain intact post-mortem, although the responses are gene-specific [83,84]. Therefore, we are confident these three or four samples are indeed expressing the screened VDD genes in a distinct matter. If post-mortem sample degradation is a factor for at least some host genes, we do not expect infectious agents to be as adversely affected, as microbes can survive passage through the gut of a predator, and therefore can continually produce mRNA transcripts, maintaining our ability to detect their presence after death of the host. In addition, tissue selection may also affect ability to detect and assess infection and needs to be considered when interpreting each infectious agent. For example, the kidneys would be more ideal for further assessments of IHNv, given that this virus causes disease within this tissue.

In conclusion, we provide evidence that specific infections can be associated with higher predation risks in wild fish, suggesting compensatory mortality. Predation may therefore aid 'migratory culling' [13,14], where the physiological impacts of infection prevent successful migration in some individuals, reducing pathogen prevalence, burdens and transmission in the population. Indeed, Mesa *et al.* [17] suggested that avian predation on smolts with BKD may explain why high infectious loads of *Renibacterium salmoninarum* are relatively rare in the Columbia River. The potential for migratory culling has important implications for management such as predator control [85]. If fish are compromised upon migration, survival may be poor regardless of predators. Thus, control of native predators may not have the intended effects on prey [86] and it is important to attempt to identify selection processes predators place on prey such as juvenile salmon (i.e. [12,74]). The ability for predators to facilitate or affect migratory culling is probably dependent upon the specific qualities of

the predators, the migrants and their movement behaviours, the infection(s) and experienced environmental conditions. For instance, the ability of the pathogen to spread before predation, or potential for other forms of transmission (prey to predator, or vertically during other life stages) are likely to affect potential for predation-assisted migratory culling. More broadly, it appears imperative to include infectious agents within monitoring of important fish populations, particularly with the possibility for individual host–infection relationships to interact with climate change and warming waters, with some infections potentially becoming less prevalent (such as IHNv, generally limited to colder waters [38]), and others more [87,88], such as *Ichthyophthirius multifiliis* [87].

Ethics. This research was approved by the University of British Columbia Animal Ethics Committee (animal care permit: A11-0125) in accordance with the Canadian Council of Animal Care.
Data accessibility. Data are available via Dryad ([89] doi:10.5061/dryad.12jm63xw2).
Authors' contributions. N.B.F., A.L.B., K.M.M. and S.G.H. conceived and planned the work. N.B.F., A.L.B., S.J.H., A.G.L. and S.M.D. contributed to field sampling. A.L.B., S.L. and K.M.M. led laboratory processing. N.B.F. and A.L.B. conducted analyses. All authors wrote, edited and gave final approval for submission of the manuscript.
Competing interests. We have no competing interests.
Funding. Work was supported by the Pacific Salmon Foundation and the Salish Sea Marine Survival Project (contribution #55), Canada's Ocean Tracking Network, Genome British Columbia (Strategic Salmon Health Initiative), MITACS Accelerate program, NSERC Discovery grant to S.G.H., Canada Foundation for Innovation and Fisheries Society of the British Islands small grants program.
Acknowledgements. We thank G. William, C. Middleton and V. Minke-Martin for field assistance, A. Tabata for database assistance, the Tsilhqot'in National Government and Xeni Gwet'in First Nation for field access, and Fisheries and Oceans Canada Stock Assessment. Some data provided in table 1 were provided by the Strategic Salmon Health Initiative (SSHI) funded by Genome British Columbia, Pacific Salmon Foundation, and Fisheries and Oceans, Canada. We thank T. Ming and K. Kaukinen for analysing SSHI samples, and D. Patterson and field crews for collection of SSHI samples.

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
