## [Peer Review File · Royal Society Open Science]

Review History

RSOS-201522.R0 (Original submission)

Review form: Reviewer 1

Is the manuscript scientifically sound in its present form?

Yes

Are the interpretations and conclusions justified by the results?

Yes

Is the language acceptable?

Yes

Do you have any ethical concerns with this paper?

Yes

Have you any concerns about statistical analyses in this paper?

No

Recommendation?

Major revision is needed (please make suggestions in comments)

Comments to the Author(s)

I reviewed this paper in an earlier stage, I still think it's excellent and very interesting and the authors responded to most of my initial comments. I suggest some minor revisions on this version.

Minor Comments

L47- rather than saying "few studies", it might be better to quickly summarise what the three studies that have studied this have concluded

L64- given that this virus is a focal point of the analysis, it would be ideal to expand, here or elsewhere, on the ecology of this virus so far as it is known

L67- a bit of a throw in to the introduction given that this has not been brought up at all yet- how does it relate? I suggest not to simply delete this, but work in earlier in the introduction how host gene expression can be affected by various biotic and abiotic factors that can affect survival

L75- in what way was it randomized? There is always selection bias with capture gears that should be acknowledged

L80- clarify what information loss is anticipated for smolts frozen at minus twenty

L82- again could slower or shallower swimming individuals be more vulnerable to dipnet capture? Is there any way to know? Just curious.

L158- what is prevalence here? Abundance or LOD value?

L199- have read this paragraph four times and find it quite difficult to decipher what is being communicated.. consider revising for clarity

L222- should be i.e. (in other words) not e.g. (for example)

L253- discuss whether this is additive or compensatory mortality

L257- alt+248 will give you the symbol for degrees

L261- detectability of the smolt?

L268- any other of the work from the Miller lab that has revealed the spatial or species distribution of this virus?

L274- is this related to relative infection burden, defined in several papers by Teffer and Bass?

L372- this discussion is lacking a discussion on the fundamentals of predation, especially the role of compensatory compared to additive mortality and what the implications are for salmon ecology. There are several other papers about salmon predation and the role of predators.. for Atlantic salmon there are some papers on cormorant predation as well as trout and cod, and striped bass predation. There is a Wood et al. paper re- merganser predation on Pacific salmon. There is a lot of anti-predator narratives that are clearly informed by the findings here that should be addressed more explicitly in the discussion.

L379- Lennox et al. Biol Conserv provides a comprehensive discussion of this

Review form: Reviewer 2**Is the manuscript scientifically sound in its present form?**

No

Are the interpretations and conclusions justified by the results?

No

Is the language acceptable?

Yes

Do you have any ethical concerns with this paper?

No

Have you any concerns about statistical analyses in this paper?

Yes

Recommendation?

Major revision is needed (please make suggestions in comments)

Comments to the Author(s)

Furey et al examine the differential consumption of infected salmon smolt by bull trout in British Columbia, Canada by screening consumed smolts for the presence of 17 infectious agents. As I mentioned in my last review, I think this is an interesting study that has some very interesting implications. However, I think the reviewers raised a number of valid concerns in the last review that were not adequately addressed by the authors. In a few cases, I learned more about the study, and information relevant to the study, from the responses to the reviewers rather than from the manuscript. In this review, I tried to point out where that information would be useful to have in the text. As a result, I still think this paper has a number of weaknesses that need to be addressed.

My one major concern is that the statistical analysis needs to be changed from multiple Fishers exact tests to a mixed effects glm approach with model selection (e.g. information criterion) to account for the lack of independence. My more specific comments related to the manuscript are below.

Introduction

Lines 64-65 – There is only one sentence for IHNV in the introduction, whereas it seems like IHNV is the major focal disease of this paper based on later descriptions in the results and discussion. I suggest setting up this disease as a major focus by providing a broader description of this disease, its life history, and its impacts on salmonids.

Methods

Lines 82-83 – I appreciate that this was an opportunistic study and there was insufficient funding to collect adequate samples in 2015. However, this should be acknowledged in the text. It would be valuable to make a comment here about how the sample sizes were decided upon and, in the discussion, make a comment that it would be valuable to have more samples of the non-predated fish to have a better sample of the prevalence of the disease throughout the population. I realize that you couldn't sample more of the non-predated smolts due to funding issues, which is no fault of your own, but is still a weakness of your study. I think we should all acknowledge the weaknesses of our studies in our papers so that the next studies that build on our research can use our experiences to improve their designs.

Lines 91-92 – I previously made the suggestion to add some indication of what the prevalence of the diseases throughout Chilko Lake to Table one. The authors ignored this suggestion, but I still think this would be extremely important to have since your sample sizes of the non-predated fish were so small. I noticed in response #10 to referee 2 that interannual variability is being assessed using monitoring programs. In that same response, you also cite some of your own groups work

in the system that provides some estimates of disease prevalence rates, so it appears these data exist.

Lines 97-98 – I had no idea that the VDD panel was not an ‘accepted or standard approach to classifying fish as diseased or not’ until I read the comments of reviewer four. Since that is what you are using to assess the disease state of your fish, and the entire conclusions of your paper depend on the diseased state of consumed and non-consumed fish, that seems like a key piece of information that should better described in the paper. I recommend the authors spend a few sentences in the methods briefly summarizing the important conclusions from Miller et al. 2017 (rather than chastising reviewer #4 that they need to read Miller et al. 2017). Much of the information that I think would be important for readers to have appears in response #19 to reviewer 4. In my opinion, a well written paper is one where there is sufficient information to justify the work without having to go and read another paper.

Lines 147-148 - How did you select these genes? Was this based on previous research that indicated these genes would not be degraded across samples? If not, was it based on the analysis of the genes from this study? If so, how can you be positive that this isn't some other artifact of the five samples that you ended up removing?

Lines 158-160 – Running a single Fisher’s exact test for each pathogen for each tissue and year is statistically inappropriate. First of all, you have tissues which are collected from the same fish. Because they are collected from the same fish they are not independent from each other. This could potentially help to address some of the concerns of reviewer #4 comment 10 – where you have different responses of the same tissue within the same fish. Then you have pathogens that are collected from the same river in the same year. I suspect that different pathogens may be correlated with environmental conditions (i.e. temperature, flow, etc); therefore, it is likely that different pathogens likely have a higher occurrence in one year than in another year. This needs to be analyzed in a mixed effects logistic regression model with individual fish as a random effect and year, pathogen, and tissue as fixed effects. The authors should also include fish length as a covariate, since the authors also run a posthoc test comparing the fork length of IHN+ and IHN- fish. This would also make sense since fish size is certainly related to fish predation and may also be related to susceptibility to disease. A glm framework would be necessary to tease out these idiosyncrasies. As the authors suggested, you can also look at interactions between year and pathogen, but based on your small sample size, I’m almost positive you will not have a sufficient sample size to detect those interactions. The wonderful thing about using AIC to select the most parsimonious model is that it will only let you fit as complex a model as your data will allow. As the authors state, the output from a logistic regression can be expressed as odds ratios, which is the same as what they express here.

Line 164 – I agree with reviewer #4 that species richness has a specific definition in ecology. The count of the number of infectious agents per sample may not be the most appropriate response to assess if species richness had an effect. I think Shannon’s diversity index, which takes into account both the count and the abundance (which in your case would probably be cycle threshold) would probably be more appropriate.

Line 175 – This is the first place in the text where I got the impression that IHNv is a focal disease of this paper. I had to go back to the introduction to realize that there was one sentence where you specifically mention this disease. As someone who isn’t specifically familiar with this disease, I suggest spending a little more time in the introduction to describe the importance of this disease and the possible population level implications it may have for salmon.

Line 228-229 – As I previously suggest, fork length should be included as a covariate in the mixed effects logistic regression.

Discussion

Lines 254-268 – Some of this paragraph should be moved to the introduction

Tables

Table 1 – a percent without a sample size is not very valuable. Furthermore, please calculate the standard errors for these percentages. There is a simple equation to calculate the standard error of a proportion.

Supplemental material

.csv files – my comment regarding the metadata for the supplemental material was primarily regarding the .csv files. I would like you to provide a excel or text file (.doc or .txt) that describes the content of all the columns in each of your .csv files in relatively easy to understand language.

Decision letter (RSOS-201522.R0)

Dear Dr Furey,

The Editors assigned to your paper RSOS-201522 "Infected juvenile salmon can experience increased predation during freshwater migration" have now received comments from reviewers and would like you to revise the paper in accordance with the reviewer comments and any comments from the Editors. Please note this decision does not guarantee eventual acceptance.

Please submit your revised manuscript and required files (see below) no later than 21 days from today's (ie 21-Sep-2020) date. Note: the ScholarOne system will 'lock' if submission of the revision is attempted 21 or more days after the deadline. If you do not think you will be able to meet this deadline please contact the editorial office immediately.

Kind regards,
Lianne Parkhouse
Editorial Coordinator
Royal Society Open Science
openscience@royalsociety.org

on behalf of the Associate Editor, and Professor Kevin Padian (Subject Editor)
openscience@royalsociety.org

Associate Editor Comments to Author:

Thank you for the transfer of this paper. Two of the original reviewers have assessed the submission and the changes you have made. One is broadly of the view the paper is on the right track; however, the second strongly feels that you have not engaged satisfactorily with the queries raised in the earlier round of review. We would like you to take their concerns seriously and would highlight that, unless there are exceptional reasons for doing so, we do not routinely permit multiple rounds of major revision: indeed, if the reviewers are not persuaded that you are taking steps to address their concerns in the revision, it is possible your paper will be rejected. With this in mind, please do your best to respond to their concerns both in a tracked-changes version of your revision and also a clear point-by-point response, so the editors and reviewers can see how you tackled the critiques. Good luck and we look forward to reading your revised paper in due course.

Reviewer comments to Author:

Reviewer: 1
Comments to the Author(s)

I reviewed this paper in an earlier stage, I still think it's excellent and very interesting and the authors responded to most of my initial comments. I suggest some minor revisions on this version.

Minor Comments

L47- rather than saying "few studies", it might be better to quickly summarise what the three studies that have studied this have concluded

L64- given that this virus is a focal point of the analysis, it would be ideal to expand, here or elsewhere, on the ecology of this virus so far as it is known

L67- a bit of a throw in to the introduction given that this has not been brought up at all yet- how does it relate? I suggest not to simply delete this, but work in earlier in the introduction how host gene expression can be affected by various biotic and abiotic factors that can affect survival

L75- in what way was it randomized? There is always selection bias with capture gears that should be acknowledged

L80- clarify what information loss is anticipated for smolts frozen at minus twenty

L82- again could slower or shallower swimming individuals be more vulnerable to dipnet capture? Is there any way to know? Just curious.

L158- what is prevalence here? Abundance or LOD value?

L199- have read this paragraph four times and find it quite difficult to decipher what is being communicated.. consider revising for clarity

L222- should be i.e. (in other words) not e.g. (for example)

L253- discuss whether this is additive or compensatory mortality

L257- alt+248 will give you the symbol for degrees

L261- detectability of the smolt?

L268- any other of the work from the Miller lab that has revealed the spatial or species distribution of this virus?

L274- is this related to relative infection burden, defined in several papers by Teffer and Bass?

L372- this discussion is lacking a discussion on the fundamentals of predation, especially the role of compensatory compared to additive mortality and what the implications are for salmon ecology. There are several other papers about salmon predation and the role of predators.. for Atlantic salmon there are some papers on cormorant predation as well as trout and cod, and striped bass predation. There is a Wood et al. paper re- merganser predation on Pacific salmon. There is a lot of anti-predator narratives that are clearly informed by the findings here that should be addressed more explicitly in the discussion.

L379- Lennox et al. Biol Conserv provides a comprehensive discussion of this

Reviewer: 2

Comments to the Author(s)

Furey et al examine the differential consumption of infected salmon smolt by bull trout in British Columbia, Canada by screening consumed smolts for the presence of 17 infectious agents. As I mentioned in my last review, I think this is an interesting study that has some very interesting implications. However, I think the reviewers raised a number of valid concerns in the last review that were not adequately addressed by the authors. In a few cases, I learned more about the study, and information relevant to the study, from the responses to the reviewers rather than from the manuscript. In this review, I tried to point out where that information would be useful to have in the text. As a result, I still think this paper has a number of weaknesses that need to be addressed.

My one major concern is that the statistical analysis needs to be changed from multiple Fishers exact tests to a mixed effects glm approach with model selection (e.g. information criterion) to account for the lack of independence. My more specific comments related to the manuscript are below.

Introduction

Lines 64-65 – There is only one sentence for IHNV in the introduction, whereas it seems like IHNV is the major focal disease of this paper based on later descriptions in the results and discussion. I suggestion setting up this disease as a major focus by providing a broader description of this disease, its life history, and its impacts on salmonids.

Methods

Lines 82-83 – I appreciate that this was an opportunistic study and there was insufficient funding to collect adequate samples in 2015. However, this should be acknowledged in the text. It would be valuable to make a comment here about how the sample sizes were decided upon and, in the discussion, make a comment that it would be valuable to have more samples of the non-predated fish to have a better sample of the prevalence of the disease throughout the population. I realize that you couldn't sample more of the non-predated smolts due to funding issues, which is no fault of your own, but is still a weakness of your study. I think we should all acknowledge the

weaknesses of our studies in our papers so that the next studies that build on our research can use our experiences to improve their designs.

Lines 91-92 – I previously made the suggestion to add some indication of what the prevalence of the diseases throughout Chilko Lake to Table one. The authors ignored this suggestion, but I still think this would be extremely important to have since your sample sizes of the non-predated fish were so small. I noticed in response #10 to referee 2 that interannual variability is being assessed using monitoring programs. In that same response, you also cite some of your own groups work in the system that provides some estimates of disease prevalence rates, so it appears these data exist.

Lines 97-98 – I had no idea that the VDD panel was not an ‘accepted or standard approach to classifying fish as diseased or not’ until I read the comments of reviewer four. Since that is what you are using to assess the disease state of your fish, and the entire conclusions of your paper depend on the diseased state of consumed and non-consumed fish, that seems like a key piece of information that should better described in the paper. I recommend the authors spend a few sentences in the methods briefly summarizing the important conclusions from Miller et al. 2017 (rather than chastising reviewer #4 that they need to read Miller et al. 2017). Much of the information that I think would be important for readers to have appears in response #19 to reviewer 4. In my opinion, a well written paper is one where there is sufficient information to justify the work without having to go and read another paper.

Lines 147-148 - How did you select these genes? Was this based on previous research that indicated these genes would not be degraded across samples? If not, was it based on the analysis of the genes from this study? If so, how can you be positive that this isn't some other artifact of the five samples that you ended up removing?

Lines 158-160 – Running a single Fisher’s exact test for each pathogen for each tissue and year is statistically inappropriate. First of all, you have tissues which are collected from the same fish. Because they are collected from the same fish they are not independent from each other. This could potentially help to address some of the concerns of reviewer #4 comment 10 – where you have different responses of the same tissue within the same fish. Then you have pathogens that are collected from the same river in the same year. I suspect that different pathogens may be correlated with environmental conditions (i.e. temperature, flow, etc); therefore, it is likely that different pathogens likely have a higher occurrence in one year than in another year. This needs to be analyzed in a mixed effects logistic regression model with individual fish as a random effect and year, pathogen, and tissue as fixed effects. The authors should also include fish length as a covariate, since the authors also run a posthoc test comparing the fork length of IHN+ and IHN- fish. This would also make sense since fish size is certainly related to fish predation and may also be related to susceptibility to disease. A glm framework would be necessary to tease out these idiosyncrasies. As the authors suggested, you can also look at interactions between year and pathogen, but based on your small sample size, I’m almost positive you will not have a sufficient sample size to detect those interactions. The wonderful thing about using AIC to select the most parsimonious model is that it will only let you fit as complex a model as your data will allow. As the authors state, the output from a logistic regression can be expressed as odds ratios, which is the same as what they express here.

Line 164 – I agree with reviewer #4 that species richness has a specific definition in ecology. The count of the number of infectious agents per sample may not be the most appropriate response to assess if species richness had an effect. I think Shannon’s diversity index, which takes into account both the count and the abundance (which in your case would probably be cycle threshold) would probably be more appropriate.

Line 175 – This is the first place in the text where I got the impression that IHNv is a focal disease of this paper. I had to go back to the introduction to realize that there was one sentence where you specifically mention this disease. As someone who isn't specifically familiar with this disease, I suggest spending a little more time in the introduction to describe the importance of this disease and the possible population level implications it may have for salmon.

Line 228-229 – As I previously suggest, fork length should be included as a covariate in the mixed effects logistic regression.

Discussion

Lines 254-268 – Some of this paragraph should be moved to the introduction

Tables

Table 1 – a percent without a sample size is not very valuable. Furthermore, please calculate the standard errors for these percentages. There is a simple equation to calculate the standard error of a proportion.

Supplemental material

.csv files – my comment regarding the metadata for the supplemental material was primarily regarding the .csv files. I would like you to provide a excel or text file (.doc or .txt) that describes the content of all the columns in each of your .csv files in relatively easy to understand language.

===PREPARING YOUR MANUSCRIPT===

If you have been asked to revise the written English in your submission as a condition of publication, you must do so, and you are expected to provide evidence that you have received language editing support. The journal would prefer that you use a professional language editing service and provide a certificate of editing, but a signed letter from a colleague who is a native speaker of English is acceptable. Note the journal has arranged a number of discounts for authors

using professional language editing services
(<https://royalsociety.org/journals/authors/benefits/language-editing/>).

===PREPARING YOUR REVISION IN SCHOLARONE===

<https://royalsociety.org/journals/authors/author-guidelines/#supplementary-material> to include a suitable title and informative caption. An example of appropriate titling and captioning may be found at https://figshare.com/articles/Table_S2_from_Is_there_a_trade-

off_between_peak_performance_and_performance_breadth_across_temperatures_for_aerobic_sc
ope_in_teleost_fishes_/3843624.

Author's Response to Decision Letter for (RSOS-201522.R0)

See Appendix A.

RSOS-201522.R1 (Revision)

Review form: Reviewer 1

Is the manuscript scientifically sound in its present form?

Yes

Are the interpretations and conclusions justified by the results?

Yes

Is the language acceptable?

Yes

Do you have any ethical concerns with this paper?

No

Have you any concerns about statistical analyses in this paper?

No

Recommendation?

Accept as is

Comments to the Author(s)

Well done- really enjoy this paper

Review form: Reviewer 2

Is the manuscript scientifically sound in its present form?

Yes

Are the interpretations and conclusions justified by the results?

Yes

Is the language acceptable?

Yes

Do you have any ethical concerns with this paper?

No

Have you any concerns about statistical analyses in this paper?

Yes

Recommendation?

Major revision is needed (please make suggestions in comments)

Comments to the Author(s)

In general, Furey et al. did a good job addressing my comments, especially in the introduction. I still have some large concerns with the analysis, that I believe the authors can address relatively quickly, if given the opportunity. I also have some concerns that the authors tend to oversell their results in the discussion. Specifically, the results show that the only disease that appeared to increase the susceptibility to predation was IHNV, which was only present in one year. In contrast, other diseases that appeared to be much more prevalent, did not increase predation rates. However, the authors did not mention this dichotomy in the discussion at all. I think this is a major oversight that can have some major implications. I recommend the authors temper their conclusions in the discussion to better represent that they only found that one disease increased predation rates, while other diseases appeared to have minimal effect.

Perhaps there are differences in these diseases and the way that they influence fish behavior that influence the fish's susceptibility to predation.

Specific comments:

Line 124 – Was using fish in the best digestion condition necessary to prevent degradation of the disease DNA? Could this have influenced your analysis later where you compared condition of the consumed and non-consumed fish?

Line 195-198 – Based on the discussion that we've been having; it is apparent that this is a complex dataset that requires some careful consideration in how it is analyzed. I appreciate that the authors put the effort into attempting the mixed effects model, unfortunately without success. However, I still don't believe that individual Fisher's exact test for each tissue and year is the correct way to analyze these data. The main problem with the GLM that I suggested appeared to be the year effect, due to the singularity issue. I will list what I think is required at a minimum and then make some additional recommendations for some added complexity that I think would improve the analysis:

1) At a minimum the authors should fit a logistic regression where the response is whether or not a fish was predated and the covariates are: a) whether or not that fish was infected with the single disease (e.g. IHNV) the authors want to test, b) fish length, c) the tissue (gill or liver) tested, d) a year effect for the diseases that occur over multiple years. Fish length needs to be included in this GLM, rather than using a second t-test later as the authors currently do. Fitting one model to test whether disease increases predation risk and a second to test if there is an effect of length on predation is inappropriate, because any results from these tests will give you false precision since you are doing two separate tests and assuming independence between them. However, it is the same fish getting eaten, so they cannot be independent. If the authors have further difficulty getting the models to converge, I highly encourage them to consult with a statistician or quantitative ecologist that can help them through the analysis.

2) It might also be interesting to try to fit a model that includes multiple diseases, but I recommend that the authors only include the most prevalent diseases (e.g. Candidatus

Branchiomonas cisticola, Flavobacterium psychrophilum, ichthyophthirius multifiliis, Infectious hematopoietic necrosis virus, Pacific salmon parvovirus). That will greatly reduce the number of parameters in your model, since it appears fairly obvious without using statistics that none of the other diseases will come out as significant.

3) Another option, if you did want to show the effect of all disease on predation risk, would be to fit a multivariate GLM. But that would be considerably more complex and not necessary for your purposes.

Lines 220-224: See my recommendation above about testing the effects of length on IHN infection. If the authors want to disentangle the effect of fish length and disease on predation rates, these need to be included in the same model. Currently, the authors are testing the hypothesis that there is no difference in length between IHN infected fish. But, it is still possible that the consumed IHN fish were smaller than all other fish.

Line 275-280: It's unclear to me how these tests differs from the tests the authors describe on lines 220-224.

Lines 302-316: I think somewhere in here you should comment on the differences between IHNv and the other diseases. You observed an increased risk of predation with IHNv, but not with any of the other infections, based on what you know of these diseases, can you formulate some hypotheses about why you observed those results?

Lines 330-331: Specifically, you provide evidence that infection with one specific disease can increase risk to fish in the wild. In fact, two other diseases, that appear to have higher prevalence in your samples and in the system, didn't have any impact on predation. It seems like you are ignoring that result to focus on the single positive result that you had. I find it really interesting that there appear to be some diseases that don't increase the risk of predation. I think that dichotomy, that some diseases do increase the risk of predation while some may not, should be addressed in the discussion.

Lines 430: Again, you are overselling your results a little. You didn't find that 'specific infections can be associated with higher predation risks', but rather that a single infection was associated with a higher predation risk while multiple others were not.

Table 1: I appreciate that the authors added the extra information that was requested, but that generally requires adjusting the table to accommodate the additional information. This table is now a little difficult to comprehend with the way it is arranged. They should to play around with formatting to make it easier for the reader to digest.

Table 1: Do you have sample sizes for these other studies? Are there any confidence intervals for these prevalence rates?

Decision letter (RSOS-201522.R1)

Dear Dr Furey

The Editors assigned to your paper RSOS-201522.R1 "Infected juvenile salmon can experience increased predation during freshwater migration" have now received comments from reviewers and would like you to revise the paper in accordance with the reviewer comments and any comments from the Editors. Please note this decision does not guarantee eventual acceptance.

Please submit your revised manuscript and required files (see below) no later than 21 days from today's (ie 11-Dec-2020) date. Note: the ScholarOne system will 'lock' if submission of the revision is attempted 21 or more days after the deadline. If you do not think you will be able to meet this deadline please contact the editorial office immediately.

Best regards,

on behalf of the Associate Editor and Professor Kevin Padian (Subject Editor)
openscience@royalsociety.org

Associate Editor Comments to Author:

Given that the authors seem to have tried hard to improve the paper but one of the reviewers still has some concerns, it is fair to offer the authors a final opportunity to revise, but they should be aware this is the final opportunity they will have.

Reviewer comments to Author:

Reviewer: 2

Comments to the Author(s)

In general, Furey et al. did a good job addressing my comments, especially in the introduction. I still have some large concerns with the analysis, that I believe the authors can address relatively quickly, if given the opportunity. I also have some concerns that the authors tend to oversell their

results in the discussion. Specifically, the results show that the only disease that appeared to increase the susceptibility to predation was IHNV, which was only present in one year. In contrast, other diseases that appeared to be much more prevalent, did not increase predation rates. However, the authors did not mention this dichotomy in the discussion at all. I think this is a major oversight that can have some major implications. I recommend the authors temper their conclusions in the discussion to better represent that they only found that one disease increased predation rates, while other diseases appeared to have minimal effect. Perhaps there are differences in these diseases and the way that they influence fish behavior that influence the fish's susceptibility to predation.

Specific comments:

Line 124 – Was using fish in the best digestion condition necessary to prevent degradation of the disease DNA? Could this have influenced your analysis later where you compared condition of the consumed and non-consumed fish?

Line 195-198 – Based on the discussion that we've been having; it is apparent that this is a complex dataset that requires some careful consideration in how it is analyzed. I appreciate that the authors put the effort into attempting the mixed effects model, unfortunately without success. However, I still don't believe that individual Fisher's exact test for each tissue and year is the correct way to analyze these data. The main problem with the GLM that I suggested appeared to be the year effect, due to the singularity issue. I will list what I think is required at a minimum and then make some additional recommendations for some added complexity that I think would improve the analysis:

- 1) At a minimum the authors should fit a logistic regression where the response is whether or not a fish was predated and the covariates are: a) whether or not that fish was infected with the single disease (e.g. IHNV) the authors want to test, b) fish length, c) the tissue (gill or liver) tested, d) a year effect for the diseases that occur over multiple years. Fish length needs to be included in this GLM, rather than using a second t-test later as the authors currently do. Fitting one model to test whether disease increases predation risk and a second to test if there is an effect of length on predation is inappropriate, because any results from these tests will give you false precision since you are doing two separate tests and assuming independence between them. However, it is the same fish getting eaten, so they cannot be independent. If the authors have further difficulty getting the models to converge, I highly encourage them to consult with a statistician or quantitative ecologist that can help them through the analysis.
- 2) It might also be interesting to try to fit a model that includes multiple diseases, but I recommend that the authors only include the most prevalent diseases (e.g. *Candidatus Branchiomonas cisticola*, *Flavobacterium psychrophilum*, *ichthyophthirius multifiliis*, Infectious hematopoietic necrosis virus, Pacific salmon parvovirus). That will greatly reduce the number of parameters in your model, since it appears fairly obvious without using statistics that none of the other diseases will come out as significant.
- 3) Another option, if you did want to show the effect of all disease on predation risk, would be to fit a multivariate GLM. But that would be considerably more complex and not necessary for your purposes.

Lines 220-224: See my recommendation above about testing the effects of length on IHN infection. If the authors want to disentangle the effect of fish length and disease on predation rates, these need to be included in the same model. Currently, the authors are testing the hypothesis that there is no difference in length between IHN infected fish. But, it is still possible that the consumed IHN fish were smaller than all other fish.

Line 275-280: It's unclear to me how these tests differs from the tests the authors describe on lines 220-224.

Lines 302-316: I think somewhere in here you should comment on the differences between IHNv and the other diseases. You observed an increased risk of predation with IHNv, but not with any of the other infections, based on what you know of these diseases, can you formulate some hypotheses about why you observed those results?

Lines 330-331: Specifically, you provide evidence that infection with one specific disease can increase risk to fish in the wild. In fact, two other diseases, that appear to have higher prevalence in your samples and in the system, didn't have any impact on predation. It seems like you are ignoring that result to focus on the single positive result that you had. I find it really interesting that there appear to be some diseases that don't increase the risk of predation. I think that dichotomy, that some diseases do increase the risk of predation while some may not, should be addressed in the discussion.

Lines 430: Again, you are overselling your results a little. You didn't find that 'specific infections can be associated with higher predation risks', but rather that a single infection was associated with a higher predation risk while multiple others were not.

Table 1: I appreciate that the authors added the extra information that was requested, but that generally requires adjusting the table to accommodate the additional information. This table is now a little difficult to comprehend with the way it is arranged. They should to play around with formatting to make it easier for the reader to digest.

Table 1: Do you have sample sizes for these other studies? Are there any confidence intervals for these prevalence rates?

Reviewer: 1

Comments to the Author(s)

well done- really enjoy this paper

===PREPARING YOUR MANUSCRIPT===

===PREPARING YOUR REVISION IN SCHOLARONE===

Author's Response to Decision Letter for (RSOS-201522.R1)

See Appendix B.

RSOS-201522.R2 (Revision)

Review form: Reviewer 2

Is the manuscript scientifically sound in its present form?

No

Are the interpretations and conclusions justified by the results?

No

Is the language acceptable?

Yes

Do you have any ethical concerns with this paper?

No

Have you any concerns about statistical analyses in this paper?

Yes

Recommendation?

Major revision is needed (please make suggestions in comments)

Comments to the Author(s)

I appreciate that the authors added the GLM analysis that was requested. I think this has greatly improved the quality of the statistics and our ability to interpret the results. However, I have some major concerns remaining, primarily with the way the logistic regression GLM was conducted:

1) Now that the GLM has been conducted, I don't think the individual Fisher exact tests are needed. I think the results from these tests are repetitive and simply serve to confuse the readers.

My guess is that the author's want to include these tests to highlight the odds ratios; however, as I mention in the attached file (Appendix C), odds ratios can be easily calculated by exponentiating the coefficients of a logistic regression.

2) There were some obvious problems with the coefficients of the top logistic regression models that were presented. Some of the covariates in these models have coefficients over 15, meaning they had odds ratios over 3 million!! This is obviously unrealistic. After doing a little investigating by looking at Table 1, I realized all these covariates with large coefficients either had 0% or 100% predated or not predated. That means there were either no values in the numerator or denominator of the odds ratio (just like you couldn't calculate the odds ratio for those diseases in those tissues in those years). Since the coefficient estimates in the logistic regression of the logs of the odds ratio, the coefficient estimates for these covariates aren't realistic. In other words, for your logistic regression model, you can't include any of the diseases for any of the tissues in any year that you couldn't calculate an odds ratio in table 1.

3) The best practice for model selection isn't to just interpret the top model, but to either use model averaging or to pick the most parsimonious model from your top model set. Things may change after you modify which diseases to include in your models, but, currently most of your top models are subsets of one of the top models (i.e., they include all the same covariate plus some some additional ones). If that continues to be the case, you should just use that most parsimonious model as your top model.

Decision letter (RSOS-201522.R2)

Dear Dr Furey

On behalf of the Editors, we are pleased to inform you that your Manuscript RSOS-201522.R2 "Infected juvenile salmon can experience increased predation during freshwater migration" has been accepted for publication in Royal Society Open Science subject to minor revision in accordance with the referees' reports. Please find the referees' comments along with any feedback from the Editors below my signature.

Please submit your revised manuscript and required files (see below) no later than 7 days from today's (ie 23-Feb-2021) date. Note: the ScholarOne system will 'lock' if submission of the revision is attempted 7 or more days after the deadline. If you do not think you will be able to meet this deadline please contact the editorial office immediately.

on behalf of Prof Kevin Padian (Subject Editor)
 openscience@royalsociety.org

Associate Editor Comments to Author:

This paper represents something of a tricky call for the editors. On the one hand, it seems clear the authors are doing their best to meet the concerns raised by the referee, but the referee has a number of outstanding concerns regarding the statistical treatment of work. As the authors have had a number of opportunities to revise, and the referee has - likewise - had a number of opportunities to review, it is not clear how productive continued review-revise-review is going to be. Instead, we are going to make the call that the authors should do what they can to address the remaining concerns in a final revision, and this revision will be assessed by the editors alone - if the latter are satisfied that the paper is publishable, it will be accepted for publication: any remaining concerns that the reviewer and the wider community may have at this stage can then be discussed openly with the paper and data accessible to all. The editors thank the reviewers for their support and the authors for their engagement with the process.

Reviewer comments to Author:

Reviewer: 2

Comments to the Author(s)

I appreciate that the authors added the GLM analysis that was requested. I think this has greatly improved the quality of the statistics and our ability to interpret the results. However, I have some major concerns remaining, primarily with the way the logistic regression GLM was conducted:

- 1) Now that the GLM has been conducted, I don't think the individual Fisher exact tests are needed. I think the results from these tests are repetitive and simply serve to confuse the readers. My guess is that the author's want to include these tests to highlight the odds ratios; however, as I mention in the attached file, odds ratios can be easily calculated by exponentiating the coefficients of a logistic regression.
- 2) There were some obvious problems with the coefficients of the top logistic regression models that were presented. Some of the covariates in these models have coefficients over 15, meaning they had odds ratios over 3 million!! This is obviously unrealistic. After doing a little investigating by looking at Table 1, I realized all these covariates with large coefficients either had 0% or 100% predated or not predated. That means there were either no values in the numerator or denominator of the odds ratio (just like you couldn't calculate the odds ratio for those diseases in those tissues in those years). Since the coefficient estimates in the logistic regression of the logs of the odds ratio, the coefficient estimates for these covariates aren't realistic. In other words, for your logistic regression model, you can't include any of the diseases for any of the tissues in any year that you couldn't calculate an odds ratio in table 1.
- 3) The best practice for model selection isn't to just interpret the top model, but to either use model averaging or to pick the most parsimonious model from your top model set. Things may change after you modify which diseases to include in your models, but, currently most of your top models are subsets of one of the top models (i.e., they include all the same covariate plus

some some additional ones). If that continues to be the case, you should just use that most parsimonious model as your top model.

===PREPARING YOUR MANUSCRIPT===

===PREPARING YOUR REVISION IN SCHOLARONE===

- 1) One version identifying all the changes that have been made (for instance, in coloured highlight, in bold text, or tracked changes);
 - 2) A 'clean' version of the new manuscript that incorporates the changes made, but does not highlight them.
 - An individual file of each figure (EPS or print-quality PDF preferred [either format should be produced directly from original creation package], or original software format).
 - An editable file of each table (.doc, .docx, .xls, .xlsx, or .csv).
 - An editable file of all figure and table captions.
- Note: you may upload the figure, table, and caption files in a single Zip folder.
- Any electronic supplementary material (ESM).
 - If you are requesting a discretionary waiver for the article processing charge, the waiver form must be included at this step.
 - If you are providing image files for potential cover images, please upload these at this step, and inform the editorial office you have done so. You must hold the copyright to any image provided.
 - A copy of your point-by-point response to referees and Editors. This will expedite the preparation of your proof.

- Ensure that your data access statement meets the requirements at <https://royalsociety.org/journals/authors/author-guidelines/#data>. You should ensure that you cite the dataset in your reference list. If you have deposited data etc in the Dryad repository, please only include the 'For publication' link at this stage. You should remove the 'For review' link.
- If you are requesting an article processing charge waiver, you must select the relevant waiver option (if requesting a discretionary waiver, the form should have been uploaded at Step 3 'File upload' above).
- If you have uploaded ESM files, please ensure you follow the guidance at <https://royalsociety.org/journals/authors/author-guidelines/#supplementary-material> to include a suitable title and informative caption. An example of appropriate titling and captioning may be found at [https://figshare.com/articles/Table_S2_from_Is_there_a_trade-off_between_peak_performance_and_performance_breadth_across_temperatures_for_aerobic_sc ope_in_teleost_fishes_/3843624](https://figshare.com/articles/Table_S2_from_Is_there_a_trade-off_between_peak_performance_and_performance_breadth_across_temperatures_for_aerobic_scope_in_teleost_fishes_/3843624).

Author's Response to Decision Letter for (RSOS-201522.R2)

See Appendix D.

Decision letter (RSOS-201522.R3)

Dear Dr Furey,

It is a pleasure to accept your manuscript entitled "Infected juvenile salmon can experience increased predation during freshwater migration" in its current form for publication in Royal Society Open Science.

You can expect to receive a proof of your article in the near future. Please contact the editorial office (openscience@royalsociety.org) and the production office (openscience_proofs@royalsociety.org) to let us know if you are likely to be away from e-mail contact – if you are going to be away, please nominate a co-author (if available) to manage the proofing process, and ensure they are copied into your email to the journal.

on behalf of Prof Kevin Padian (Subject Editor)
openscience@royalsociety.org

Appendix A

Handling Editor comments:

Associate Editor Comments to Author:

Thank you for the transfer of this paper. Two of the original reviewers have assessed the submission and the changes you have made. One is broadly of the view the paper is on the right track; however, the second strongly feels that you have not engaged satisfactorily with the queries raised in the earlier round of review. We would like you to take their concerns seriously and would highlight that, unless there are exceptional reasons for doing so, we do not routinely permit multiple rounds of major revision: indeed, if the reviewers are not persuaded that you are taking steps to address their concerns in the revision, it is possible your paper will be rejected. With this in mind, please do your best to respond to their concerns both in a tracked-changes version of your revision and also a clear point-by-point response, so the editors and reviewers can see how you tackled the critiques. Good luck and we look forward to reading your revised paper in due course.

Response: Thank you for providing the reviewer comments. We are happy to hear that both reviewers find value in the paper. We have taken careful care to respond to each comment made by each reviewer, documented below. We feel we have done everything possible to meet the reviewers' requests as closely as possible.

Please note that all line numbers mentioned in response to reviewer comments refer to line numbers in the "track-changes" document, not the "clean" version.

Reviewer: 1

Comment #1: I reviewed this paper in an earlier stage, I still think it's excellent and very interesting and the authors responded to most of my initial comments. I suggest some minor revisions on this version.

Response #1: We appreciate that the reviewer finds the paper to be excellent and interesting. We also appreciate that the reviewer acknowledges we took their suggestions to heart.

Minor Comments

Comment #2: L47- rather than saying "few studies", it might be better to quickly summarise what the three studies that have studied this have concluded

Response #2: We have modified the sentence to summarize that these studies found a link between infection and predation risk.

Comment #3: L64- given that this virus is a focal point of the analysis, it would be ideal to expand, here or elsewhere, on the ecology of this virus so far as it is known

Response #3: We have text in the Introduction that incorporates more background on IHNV (L82-87).

Comment #4: L67- a bit of a throw in to the introduction given that this has not been brought up at all yet- how does it relate? I suggest not to simply delete this, but work in earlier in the introduction how host gene expression can be affected by various biotic and abiotic factors that can affect survival

Response #4: We have added a paragraph in the Introduction that more fully the VDD panel, including its development, validation, and utility (L59-74).

Comment #5: L75- in what way was it randomized? There is always selection bias with capture gears that should be acknowledged

Response #5: In addition to stating that smolts were captured via dipnet (L113) we have added text to also state smolts were pulled at random from a wash bin (L114).

Comment #6: L80- clarify what information loss is anticipated for smolts frozen at minus twenty

Response #6: We have added text to state we do not expect information loss due to the short time (up to 72 hours) smolts were left at -20 before transferring to liquid nitrogen or -80 freezer for long term storage (L111-113).

Comment #7: L82- again could slower or shallower swimming individuals be more vulnerable to dipnet capture? Is there any way to know? Just curious.

Response #7: We don't think there would be any way to know for sure, but certainly using a dipnet could target slower individuals. If our samples of non-predated smolts is indeed biased towards slow-moving or otherwise compromised individuals, it is possible that these fish have higher impairments than seen in the population. This introduced bias would probably act to dampen the effect sizes we observed (rather than increase) when comparing predated and non-predated fish, which were quite strong, particularly with IHNV. We do also feel that these potential biases towards slow-moving and shallower swimming individuals are likely small. Waters are shallow (< 1 m) so the dipnet is generally sampling the upper half of the water column (rather than a small percentage). In addition, the smolt migrations can be very dense, with the river practically boiling with smolts (sometimes breaking the dipnet). Given the densities of

smolts in the river and in our dipnet, it certainly appears to be as random of a sample as possible while still using capture (but we admit this is speculation).

Comment #8: L158- what is prevalence here? Abundance or LOD value?

Response #8: We have edited the sentence to make it clear that prevalence refers to percentage of smolts that were positive for an agent (not load).

Comment #9: L199- have read this paragraph four times and find it quite difficult to decipher what is being communicated.. consider revising for clarity

Response #9: We agree that this paragraph could be improved. We have re-written to improve clarity (L243-250).

Comment #10: L222- should be i.e. (in other words) not e.g. (for example)

Response #10: Good catch! Edit made.

Comment #11: L253- discuss whether this is additive or compensatory mortality

Response #11: Although we feel it would be too speculative to assign all of this mortality as compensatory, we have amended this statement to more clearly link that we feel the mortality observed in Jeffries et al (2014) was at least partially explained by predation in our study. But see our response to Comment #16 regarding additional discussion of predation on salmonids.

Comment #12: L257- alt+248 will give you the symbol for degrees

Response #12: Thank you; we are using the proper symbol now.

Comment #13: L261- detectability of the smolt?

Response #13: We believe the detectability of the smolt is included in “the predator’s propensity to target the smolt” (greater detectability, greater propensity). No edits made based on this comment.

Comment #14: L268- any other of the work from the Miller lab that has revealed the spatial or species distribution of this virus?

Response #14: We have added additional text to the Introduction (where we felt it fit a bit better and in response to other reviewer comments), to state that IHNV’s current broader geographic range (North America, Europe, and Asia) and that the virus largely infects fish in freshwater and is most effective at infecting fish at temperatures 10-12 (L82-86).

Comment #15: L274- is this related to relative infection burden, defined in several papers by Teffer and Bass?

Response #15: Relative infection burden (RIB) would be another way to assess these data; however Reviewer 2 specifically requested a Shannon diversity index. So we have amended our methods and results, along with this paragraph in the discussion, to reflect our new analyses. However, in response to this comment we do add that RIB exists as another metric with the relevant citation (L323).

Comment #16: L372- this discussion is lacking a discussion on the fundamentals of predation, especially the role of compensatory compared to additive mortality and what the implications are for salmon ecology. There are several other papers about salmon predation and the role of predators.. for Atlantic salmon there are some papers on cormorant predation as well as trout and cod, and striped bass predation. There is a Wood et al. paper re- merganser predation on Pacific salmon. There is a lot of anti-predator narratives that are clearly informed by the findings here that should be addressed more explicitly in the discussion.

Response #16: We have added an additional paragraph (L330-337) in the Discussion dedicated to this topic, including citing some of the papers/systems mentioned by the reviewer, to state how predation can be compensatory, but that these interactions are difficult to quantify. We also made some minor changes to the final paragraph to help tie in this earlier paragraph (L439-440).

Comment #17: L379- Lennox et al. Biol Conserv provides a comprehensive discussion of this

Response #17: This is a good and very relevant reference; it has been added.

Reviewer: 2

Comments to the Author(s)

Comment #1: Furey et al examine the differential consumption of infected salmon smolt by bull trout in British Columbia, Canada by screening consumed smolts for the presence of 17 infectious agents. As I mentioned in my last review, I think this is an interesting study that has some very interesting implications. However, I think the reviewers raised a number of valid concerns in the last review that were not adequately addressed by the authors. In a few cases, I learned more about the study, and information relevant to the study, from the responses to the reviewers rather than from the manuscript. In this review, I tried to point out where that information would be useful to have in the text. As a result, I still think this paper has a number of weaknesses that need to be addressed. My one major concern is that the statistical analysis needs to be changed from multiple

Fishers exact tests to a mixed effects glm approach with model selection (e.g. information criterion) to account for the lack of independence. My more specific comments related to the manuscript are below.

Response #1: We appreciate that the reviewer still finds this study interesting with very interesting implications. We have attempted to address the weaknesses pointed out by the reviewer (detailed in our responses below), including to take care to insert salient points into the manuscript itself in addition to our direct response to the reviewer. We attempted to use a glmm approach, as requested by the reviewer, but it was not successful due to limitations of our data (primarily singularity issues that result due to unbalanced sampling design and interannual variability in pathogen prevalence); we detail these issues fully in Response #7 below. We hope that we are able to demonstrate that we took the reviewer's suggestion to heart, attempted to completely redo our analyses, but simply could not due to limitations of our data.

Introduction

Comment #2: Lines 64-65 – There is only one sentence for IHNv in the introduction, whereas it seems like IHNv is the major focal disease of this paper based on later descriptions in the results and discussion. I suggestion setting up this disease as a major focus by providing a broader description of this disease, its life history, and its impacts on salmonids.

Response #2: We agree, as did Reviewer 1. We have added text in the Introduction to provide better background on INHv (L82-87).

Methods

Comment #3: Lines 82-83 – I appreciate that this was an opportunistic study and there was insufficient funding to collect adequate samples in 2015. However, this should be acknowledged in the text. It would be valuable to make a comment here about how the sample sizes were decided upon and, in the discussion, make a comment that it would be valuable to have more samples of the non-predated fish to have a better sample of the prevalence of the disease throughout the population. I realize that you couldn't sample more of the non-predated smolts due to funding issues, which is no fault of your own, but is still a weakness of your study. I think we should all acknowledge the weaknesses of our studies in our papers so that the next studies that build on our research can use our experiences to improve their designs.

Response #3: We have now stated in the Methods why the sample size of non-predated fish was so low – funding and a field season cut short by high waters (which affected our other research in the system) (L116-118).

Comment #4: Lines 91-92 – I previously made the suggestion to add some indication of what the prevalence of the diseases throughout Chilko Lake to Table one. The authors ignored this suggestion, but I still think this would be extremely important to have since your sample sizes of the non-predated fish were so small. I noticed in response #10 to referee 2 that interannual variability is being assessed using monitoring programs. In that same response, you also cite some of your own groups work in the system that provides some estimates of disease prevalence rates, so it appears these data exist.

Response #4: We have added a couple of columns to Table 1 to show the prevalence rates of pathogens observed in either Jeffries et al. (2014) or Stevenson et al. (2020). Please note that these are only for a limited number of pathogens and sample sizes in these studies were low in some years. We do not consider these, from a sockeye salmon population perspective, a comprehensive examination or screening of pathogens. Samples have been taken for monitoring over the past several years as part of the Strategic Salmon Health Initiative but these data are not yet available to publish. This is why we stated in our previous response to the reviewer that the broader prevalence of infectious agents in the lake environment is unknown.

Comment #5: Lines 97-98 – I had no idea that the VDD panel was not an ‘accepted or standard approach to classifying fish as diseased or not’ until I read the comments of reviewer four. Since that is what you are using to assess the disease state of your fish, and the entire conclusions of your paper depend on the diseased state of consumed and non-consumed fish, that seems like a key piece of information that should better described in the paper. I recommend the authors spend a few sentences in the methods briefly summarizing the important conclusions from Miller et al. 2017 (rather than chastising reviewer #4 that they need to read Miller et al. 2017). Much of the information that I think would be important for readers to have appears in response #19 to reviewer 4. In my opinion, a well written paper is one where there is sufficient information to justify the work without having to go and read another paper.

Response #5: We have added a paragraph to the Introduction (L59-74; we felt it was better placed here than in the Methods and more likely to be digested by the reader, no pun intended) that introduces and defines the VDD approach and how it was validated in Miller et al. to hopefully give the reader a better understanding and confidence in the approach. Human diagnostics are also moving to similar biomarker-based approaches, and it was, in fact, developments in the human diagnostics field that spurred the development of the VDD panel in salmon. Interestingly, half of the biomarkers that are predictive of a viral disease state in salmon are shared with those uncovered to recognize respiratory viral infections in humans, and differentiate them from bacterial respiratory infections. Simply, we used a more modern approach, with a

precedence in human medicine, and made substantial, and peer-reviewed, efforts to validate the approach, especially for application with IHNV. More of this detail is now in the paper.

We also apologize if our response to the reviewer was seen as chastising, that was not the intent. However, we firmly believe that the VDD approach, although a more modern and less used approach relative to traditional epidemiological research, is indeed a validated (and thus accepted) approach.

Comment #6: Lines 147-148 - How did you select these genes? Was this based on previous research that indicated these genes would not be degraded across samples? If not, was it based on the analysis of the genes from this study? If so, how can you be positive that this isn't some other artifact of the five samples that you ended up removing?

Response #6: The Reference genes were originally developed in the Miller laboratory almost a decade ago based on extensive analysis of in-house microarray datasets across multiple tissues, species, and experimental studies. They have been applied as TaqMan assays in many of our transcriptomic studies (e.g. Miller et al. 2014, Jeffries et al. 2014), where we often additionally apply normfinder or other similar software to ensure that they are behaving as expected. We have no evidence that these genes were not behaving as expected.

Comment #7: Lines 158-160 – Running a single Fisher’s exact test for each pathogen for each tissue and year is statistically inappropriate. First of all, you have tissues which are collected from the same fish. Because they are collected from the same fish they are not independent from each other. This could potentially help to address some of the concerns of reviewer #4 comment 10 – where you have different responses of the same tissue within the same fish. Then you have pathogens that are collected from the same river in the same year. I suspect that different pathogens may be correlated with environmental conditions (i.e. temperature, flow, etc); therefore, it is likely that different pathogens likely have a higher occurrence in one year than in another year. This needs to be analyzed in a mixed effects logistic regression model with individual fish as a random effect and year, pathogen, and tissue as fixed effects. The authors should also include fish length as a covariate, since the authors also run a posthoc test comparing the fork length of IHN+ and IHN- fish. This would also make sense since fish size is certainly related to fish predation and may also be related to susceptibility to disease. A glm framework would be necessary to tease out these idiosyncrasies. As the authors suggested, you can also look at interactions between year and pathogen, but based on your small sample size, I’m almost positive you will not have a sufficient sample size to detect those interactions. The wonderful thing about using AIC to select the most parsimonious model is that it will

only let you fit as complex a model as your data will allow. As the authors state, the output from a logistic regression can be expressed as odds ratios, which is the same as what they express here.

Response #7: There are several pieces to unravel here, and we will do our best to address each point (but felt this entire section was motivated by one issue – our statistical approach). Simply, we attempted to follow the reviewer’s suggestion, but the data are not amenable to this glmm approach. Based on our best interpretation of the reviewer’s suggestion, we would run models of the form (including pathogens a – i):

Predation(1/0) ~ infection(pathogen_a)+infection(pathogen_b)+...+infection(pathogen_i) + Year + FL + Tissue + (FishID_random).

This framework is problematic given our dataset for several reasons. In fact, we attempted to run some of these models, and experienced several problems, detailed below.

- ***Most importantly, given that some pathogens are only seen in certain tissues or in certain years, we have issues of singularity. Models become singular if the parameter estimates are on the boundary of the feasible parameter space – variances of one or more linear combinations of effects are zero or very close to zero (paraphrased from lme4 package helpfiles). In more practical terms, singularity can indicate overfitting of low-power models (such as ours) and increase chances of numerical errors. Our most important pathogen, IHNv, is a perfect example. When attempting to run a GLMM as constructed above, we run into singularity because IHNv is only found in one of years, and not the other (so the model cannot determine the impact of the infection vs year on predation). We tried to run these models in response to the reviewer comment, and they simply would often not converge or result in singularity (so they would run, but difficult to trust the results). If we then ran models only for pathogens and years in which they were present, which would alleviate some of these singularity issues, we would then be left with varying sample sizes among models, rendering AIC and the desired approach from the reviewer having little utility. We hope the reviewer understands that we spent a substantial amount of time developing and attempting these models, but it was clear our data do not have the sample size (nor consistent enough prevalence of all pathogens between years and tissues) to use this approach. Further justification against this approach is given below.***
- ***Given that we screened 17 pathogens (10 of which were observed), either the presence of each pathogen would be included as its own explanatory variable in the same global model as stated in our theoretical formulation above (which we***

- also do not have the power for), or we would need to run 10 separate global models (as we did with IHNv noted above), and then conduct subsequent model selection for each. This would quickly turn the entire paper into a modelling exercise, and I'd argue with greater issues of multiple comparisons than our approach has currently (where we help account for this by using a false discovery rate-adjusted p-value).*
- *Adding year as an explanatory variable in models predicting predation status is also problematic because our sample sizes (of predated vs not predated fish) are not balanced between the two years. Our models would suggest that a fish is more likely to be predated in the second year, simply because a greater proportion of our samples were indeed from bull trout stomachs (because we were financially limited in running further non-predated sample and our field season was cut short due to high flow conditions in the Chilkco River). This was a large reason why we ran analyses on separate years. The alternative would be to have infection status be the response variable, with predation status and year (as well as the others the reviewer desired) as explanatory variables, but this would be investigating a fundamentally different hypothesis (what influences probability of infection, rather than predation).*
 - *We also respectfully disagree that lack of independence is a substantial issue here. Non-independence is of greatest issue when the assumption of independence is broken within an analysis; here we have separated out each tissue into their own analyses (as the reviewer, and other reviewers, have stated, we expect different responses from infectious agents in different tissues). To account for this properly, it would require an interaction term between the specific infectious agent and tissue, which would further exacerbate our low power (which the reviewers again acknowledged). Similarly, the reviewer argues we need to include year as a covariate into a glmm to determine if prevalence rates differ among years. Excluding the year in analyses would indeed be an issue if we conducted analyses that combined data between years; however we clearly demonstrate the interannual variability in infectious agent prevalence between both years (Table 1). And we have demonstrated previously why year as a covariate in a glmm framework is problematic.*
 - *In response, we instead use Fisher's exact tests on tissue-year combinations with post-hoc analyses to investigate potentially confounding factors (fish length). We do account for the repeated testing by using a false discovery rate (fdr). We would also like to note that in terms of identifying large effects, our approach was successful, highlighting the importance of IHNv, and using the fdr-adjusted p-values prevented us from overinterpretation of other infectious agents for which the effects were smaller.*

Comment #8: Line 164 – I agree with reviewer #4 that species richness has a specific definition in ecology. The count of the number of infectious agents per sample may not be the most appropriate response to assess if species richness had an effect. I think Shannon’s diversity index, which takes into account both the count and the abundance (which in your case would probably be cycle threshold) would probably be more appropriate.

Response #8: We have now replaced the unique number of infectious agents with a Shannon diversity index (new Figure 1). The results, in terms of overall trends, are the same (greater infectious agent diversity in both tissues in 2014, with no significant differences in either tissue in 2015). Methods (L201-204), Results (L235-241), and Discussion have been edited to reflect this change in methods.

Comment #9: Line 175 – This is the first place in the text where I got the impression that IHNv is a focal disease of this paper. I had to go back to the introduction to realize that there was one sentence where you specifically mention this disease. As someone who isn’t specifically familiar with this disease, I suggest spending a little more time in the introduction to describe the importance of this disease and the possible population level implications it may have for salmon.

Response #9: We agree (and Reviewer #1 did as well). We have added additional text to the Introduction (L82-87) to better introduce IHNv to complement the text already in the Discussion.

Comment #10: Line 228-229 – As I previously suggest, fork length should be included as a covariate in the mixed effects logistic regression.

Response #10: Please see Response #7 above.

Discussion

Comment #11: Lines 254-268 – Some of this paragraph should be moved to the introduction

Response #11: We have now better introduced IHNv in the Introduction (L82-87).

Tables

Comment #12: Table 1 – a percent without a sample size is not very valuable. Furthermore, please calculate the standard errors for these percentages. There is a simple equation to calculate the standard error of a proportion.

Response #12: Sample sizes have been added to the table caption. Standard errors have also been added.

Supplemental material

Comment #13: .csv files – my comment regarding the metadata for the supplemental material was primarily regarding the .csv files. I would like you to provide a excel or text file (.doc or .txt) that describes the content of all the columns in each of your .csv files in relatively easy to understand language.

Response #13: We apologize if our metadata were not able to be found. In response to the reviewer’s comment in the previous draft, we did develop an Excel file that did provide a “dictionary” for each column as the first sheet. The name of this file, submitted to Dryad, is “SampleMetadata_UsedforPub” and the first sheet is titled “Dictionary” and can still be accessed for review. This is my first time using Dryad for a submitted paper, so I apologize if these materials were not properly available to reviewers. The data submission can also be accessed via this link:

https://datadryad.org/stash/share/-0kK4Evaal9gKPdeFOGhJWz_e_JvuK0BQFYK5eMnZQM

Appendix B

Associate Editor Comments to Author:

Given that the authors seem to have tried hard to improve the paper but one of the reviewers still has some concerns, it is fair to offer the authors a final opportunity to revise, but they should be aware this is the final opportunity they will have.

Response: Thank you, and we appreciate that our hard work has been evident. We understand that this is our final opportunity, and hope the reviewer appreciates the lengths we went to meet their demands. We also hope the Associate Editor and reviewer can appreciate the value of this story – that even with a complicated system and limited sample size, we were able to observe this link between infection and predation. Regardless, we appreciate the efforts of all of the reviewers that have been involved with this manuscript.

All line numbers referred to in our response below correspond to the track-changes document (rather than the “clean” version).

Reviewer comments to Author:

Reviewer: 2

Comments to the Author(s)

Comment #1: In general, Furey et al. did a good job addressing my comments, especially in the introduction. I still have some large concerns with the analysis, that I believe the authors can address relatively quickly, if given the opportunity. I also have some concerns that the authors tend to oversell their results in the discussion. Specifically, the results show that the only disease that appeared to increase the susceptibility to predation was IHNV, which was only present in one year. In contrast, other diseases that appeared to be much more prevalent, did not increase predation rates. However, the authors did not mention this dichotomy in the discussion at all. I think this is a major oversight that can have some major implications. I recommend the authors temper their conclusions in the discussion to better represent that they only found that one disease increased predation rates, while other diseases appeared to have minimal effect.

Perhaps there are differences in these diseases and the way that they influence fish behavior that influence the fish's susceptibility to predation.

Response #1: We appreciate the reviewer felt that overall, we did a good job in addressing their comments. Focusing on IHNV's impacts to us is not “overselling” the results relative to the pathogens that are not linked to mortality, but rather we focus on this result because it is so striking (and makes sense, given the literature on this infectious agent). Rather than temper our conclusions, we instead add text to the Discussion that clearly recognizes that most infectious agents did not result in increased predation risk, which is expected (L359-367); we agree this is an important addition that we overlooked. Please note that infection does not equate into disease (all of us, and animals, have several infectious agents in our systems, but disease is only experienced at specific agent-load levels). So our results demonstrate that most infections we saw did not result in increased predation risk, rather than disease. We also add text in the Discussion (L338-358) to place some of the new results

(see response to comments below) in context, including increased discussion of other pathogens.

Specific comments:

Comment #2: Line 124 – Was using fish in the best digestion condition necessary to prevent degradation of the disease DNA? Could this have influenced your analysis later where you compared condition of the consumed and non-consumed fish?

Response #2: Yes, we wanted to minimize the potential degradation of the infectious agent by selecting the best digestion individuals. Yes, it is possible that even worse condition individuals could have experienced further degradation than we observed. We discuss the potential impacts of our sampling methodology, and potential biases due to degradation, (L388-405 and 458-487).

Comment #3: Line 195-198 – Based on the discussion that we've been having; it is apparent that this is a complex dataset that requires some careful consideration in how it is analyzed. I appreciate that the authors put the effort into attempting the mixed effects model, unfortunately without success. However, I still don't believe that individual Fisher's exact test for each tissue and year is the correct way to analyze these data. The main problem with the GLM that I suggested appeared to be the year effect, due to the singularity issue. I will list what I think is required at a minimum and then make some additional recommendations for some added complexity that I think would improve the analysis:

1) At a minimum the authors should fit a logistic regression where the response is whether or not a fish was predated and the covariates are: a) whether or not that fish was infected with the single disease (e.g. IHNV) the authors want to test, b) fish length, c) the tissue (gill or liver) tested, d) a year effect for the diseases that occur over multiple years. Fish length needs to be included in this GLM, rather than using a second t-test later as the authors currently do. Fitting one model to test whether disease increases predation risk and a second to test if there is an effect of length on predation is inappropriate, because any results from these tests will give you false precision since you are doing two separate tests and assuming independence between them. However, it is the same fish getting eaten, so they cannot be independent. If the authors have further difficulty getting the models to converge, I highly encourage them to consult with a statistician or quantitative ecologist that can help them through the analysis.

2) It might also be interesting to try to fit a model that includes multiple diseases, but I recommend that the authors only include the most prevalent diseases (e.g. *Candidatus Branchiomonas cisticola*, *Flavobacterium psychrophilum*, *ichthyophthirius multifiliis*, *Infectious hematopoietic necrosis virus*, *Pacific salmon parvovirus*). That will greatly reduce the number of parameters in your model, since it appears fairly obvious without using statistics that none of the other diseases will come out as significant.

3) Another option, if you did want to show the effect of all disease on predation risk, would be to fit a multivariate GLM. But that would be considerably more complex and not necessary for your purposes.

Response #3: We have conducted additional analyses. However, they are not exactly as prescribed by the reviewer, for the reasons we discuss below (as well as in the Methods; L119-217, Results L250-263, and Discussion L338-358 and 426-434. The reviewer requested both a global model including both years "for diseases that occur over multiple years" but also to "try a model that includes multiple diseases." Even here, it is unclear how many models the reviewer actually wants presented, and recognizes the complexity of our data. This also speaks to (as noted in our previous revision and

response to reviewers, and by the reviewer above) that because some pathogens are only found in one year or the other, it is difficult to assess multiple pathogens simultaneously AND include year as a covariate. It's also unclear what adding year as a covariate would provide beyond our current analyses (the reader can easily assess the relative impacts of an agent on predation risk, and overall prevalence, between both years).

- *Including tissue is nonsensical as an explanatory variable for models attempting to explain probability of predation (predation binary as response variable). The coefficients from this value would simply reflect the number of samples taken for each predation group for each tissue. It would not reflect differences in the relationship between predation probability and infectious agents between tissues without including an interaction (agent1 * tissue), and given our sample size, we cannot include interactions between each agent and tissue.*
- *Only including the "most prevalent" pathogens is also not a sound a priori modeling decision. The most virulent pathogens generally occur at lower prevalences (because in many cases, except at extremely high host densities, hosts perish rapidly and are unable to pass on the infection). As stated in our paper, there is other work pointing to IHNV affecting survival, with population-level prevalence rates at <15%. In reality, the fact that a given pathogen occurs at high prevalence is a likely indication that it is NOT virulent. Furthermore, it is simply not good science to hand-pick the pathogens to assess, particularly when the reviewer wants a more comprehensive analysis than what we have presented previously.*
- *Similarly, although adding FL is a good idea to a modeling framework, this only acts to assess the independent impact of fish length on predation risk – understanding how the relationship between predation ~ FL is affected by pathogens would require interaction terms (which due to sample size, we cannot explore) or further post-hoc assessments such as those we provided (size distributions of infection-positive vs infection-negative fish).*

In light of this, while also attempting to provide a more comprehensive analysis as requested by the reviewer, we added the following generalized linear modelling (GLM) framework to our paper (also described in the Methods L 199-217).

- *Four global models were constructed, one for each year-tissue combination (so 2014-gill, 2014-liver, 2015-gill, and 2015-liver)*
- *Predation status was the response variable (as requested)*
- *Explanatory variables included: FL and presence/absence of infectious agents. Only infectious agents that were detected at least twice in a given tissue-year combination were included (this helped ensure a large enough sample size to have faith in a result in as consistent of a manner as possible). Infectious agents that were found among all samples, predated and not, were not included (as these would thus have no impact on predation risk).*
- *Another confounding factor in smolt lengths is smolt age. Two age classes emigrate from Chilko Lake, with Age-1 smolts constituting on average ~96% of the migrating population, while age-2 are substantially larger but make up ~4% of the migration. Thus length is confounded by age. Age 2 fish were only sampled in 2014, with 8 of the 32 predated smolts being age-2 (no control fish were age 2). Thus, age-2 smolts were removed from 2014 GLM analyses, as they were only present in the predated group (and thus age and FL were confounded).*

- ***We used all subsets regression to rank candidate models via AICc. But to prevent overfitting due to our small sample sizes, the maximum number of parameters in each candidate model was limited to three (not including the intercept).***

Overall, these models still identified the main result – that IHNv strongly increased predation risk. However, some other interesting results emerged, including smaller smolts at higher risk of predation, and potential increase in predation risk associated with Ichthyophthirius multifiliis. Please see our new Results (L250-258) and Table 2) and Discussion (L338-358) on these topics. These models do represent an improvement to the paper. However, we feel these analyses work best in addition to, rather than in replacement of, our former results. This is largely due to the inability to include all pathogens within global models (and given this is the broadest published screening of infectious agents in this population to-date, it is important to publish the prevalence rates and odds-ratio associated with predation in a straightforward manner) and that we had to do further subsetting of the data to run the models.

Comment #4: Lines 220-224: See my recommendation above about testing the effects of length on IHN infection. If the authors want to disentangle the effect of fish length and disease on predation rates, these need to be included in the same model. Currently, the authors are testing the hypothesis that there is no difference in length between IHN infected fish. But, it is still possible that the consumed IHN fish were smaller than all other fish.

Response #4: Please see our new GLM analyses and response to the broader comment. We do see evidence of size-based selection, but it still appears that this effect is independent of IHNv infection (which is logical, based on the speed at which IHNv causes disease, as described in our paper). Please note that even the reviewer’s suggested modelling framework would have not identified if “consumed IHN+ fish were smaller than all other fish” without including an interaction term, which our study sample size simply would not allow.

Comment #5: Line 275-280: It’s unclear to me how these tests differs from the tests the authors describe on lines 220-224.

Response #5: We are confused by this comment, because lines 220-224 referred to comparisons of fish length, while lines 275-280 referred to comparisons of fish condition. No changes to the text have been made from this comment.

Comment #6: Lines 302-316: I think somewhere in here you should comment on the differences between IHNv and the other diseases. You observed an increased risk of predation with IHNv, but not with any of the other infections, based on what you know of these diseases, can you formulate some hypotheses about why you observed those results?

Response #6: In our Discussion, we do have a paragraph describing why IHNv is unique – in terms of its ability to infect, cause disease, and affect mortality of juvenile sockeye salmon. Simply, IHNv has long been known to cause acute disease and mortality, particularly in juvenile salmonids, relative to many of the other infectious agents we screened (L323-327). The infectious agents we screen are quite diverse, and thus should not be expected to behave similarly (some are viruses, others bacteria, others parasites). However, in response to this comment as well as a previous one, we have added text in the Discussion to clearly acknowledge that most infectious agents do not cause an increase in predation

risk (L359-367).

Comment #7: Lines 330-331: Specifically, you provide evidence that infection with one specific disease can increase risk to fish in the wild. In fact, two other diseases, that appear to have higher prevalence in your samples and in the system, didn't have any impact on predation. It seems like you are ignoring that result to focus on the single positive result that you had. I find it really interesting that there appear to be some diseases that don't increase the risk of predation. I think that dichotomy, that some diseases do increase the risk of predation while some may not, should be addressed in the discussion.

Response #7: It is important to make the clear distinction between an infection and disease (see L55-58 and L359-360). Infection is simply when a pathogen (something that could cause disease) is present. Infection can occur without disease (similar to how many with COVID19 are asymptomatic). Every animal has several infections at any given time, but that does not mean they are diseased. Disease is when an organism's function is affected by the presence of an infection. Although our use of VDD genes allows us to identify potential smolts that are experiencing disease, the reviewer here is focusing our prevalence rates of infections. It is not surprising, rather expected, that infectious agents can be present without increasing predation risk. Particularly, when infectious agents are at very high prevalence rates (90+% as we observe in the couple pathogens noted by the reviewer), is highly likely they do not cause disease in that given host (unless we were witnessing an epidemic before our eyes) or at least not strong enough disease to impact survival (think of the common cold). We have added text to recognize that most infectious agents will not increase predation risk (L359-367).

Comment #8: Lines 430: Again, you are overselling your results a little. You didn't find that 'specific infections can be associated with higher predation risks', but rather that a single infection was associated with a higher predation risk while multiple others were not.

Response #8: We do not understand this comment. IHNv is a 'specific infection' – we do not claim that many or all infections result in increased predation risk. We did not edit the text based on this comment. In addition, the new models requested by the reviewer suggest at least one other pathogen could be linked to predation.

Comment #9: Table 1: I appreciate that the authors added the extra information that was requested, but that generally requires adjusting the table to accommodate the additional information. This table is now a little difficult to comprehend with the way it is arranged. They should to play around with formatting to make it easier for the reader to digest.

Response #9: We have made additional adjustments in Word (changing column widths throughout, further reducing font size) but indeed a lot of information was asked for. We are hopeful that further organization can be done at the typesetting phase, if we are fortunate enough to publish.

Comment #10: Table 1: Do you have sample sizes for these other studies? Are there any confidence intervals for these prevalence rates?

Response #10: We have decided to replace info from these others studies, that felt awkward, with results from provincial screening of infectious agents in this population of juvenile sockeye salmon smolts from mixed tissues (via the Strategic Salmon Health Initiative). We were able to acquire these data between the previous revision and now, and permission to use here. See amendments to Table 1. We include ranges of sample sizes in the Table caption. However, due to space constraints (already

noted by the reviewer in Comment #9), we did not include confidence intervals of these prevalence rates (but with proportions these can be calculated from the sample size).

Reviewer: 1

Comments to the Author(s)

Comment: well done- really enjoy this paper

Response: We are glad that someone did enjoy the paper. Thank you for your continued support of this paper and seeing value in it.

Appendix C**ROYAL SOCIETY
OPEN SCIENCE****Infected juvenile salmon can experience increased
predation during freshwater migration**

Journal:	Royal Society Open Science
Manuscript ID	RSOS-201522.R2
Article Type:	Research
Date Submitted by the Author:	26-Jan-2021
Complete List of Authors:	Furey, Nathan; University of New Hampshire, Biological Sciences Bass, Arthur; The University of British Columbia, Forest and Conservation Sciences Miller, Kristi; Fisheries and Oceans Canada - Pacific Biological Station, Molecular Genetics Section Li, Shaorong; Fisheries and Oceans Canada - Pacific Biological Station, Molecular Genetics Section Lotto, Andrew; The University of British Columbia, Forest and Conservation Sciences Healy, Stephen; Fisheries and Oceans Canada Pacific Region, Science Branch Drenner, S; Stillwater Sciences; University of California Santa Barbara Hinch, Scott; The University of British Columbia, Forest and Conservation Sciences
Subject:	behaviour < BIOLOGY, ecology < BIOLOGY, molecular biology < BIOLOGY
Keywords:	predator-prey interactions, infectious hematopoietic virus, migratory culling, migration ecology, predation risk, Pacific salmon
Subject Category:	Organismal and Evolutionary Biology

Author-supplied statements

Relevant information will appear here if provided.

Ethics

Does your article include research that required ethical approval or permits?:

Yes

Statement (if applicable):

This research was approved by the University of British Columbia Animal Ethics Committee (animal care permit: A11-0125) in accordance with the Canadian Council of Animal Care.

Data

It is a condition of publication that data, code and materials supporting your paper are made publicly available. Does your paper present new data?:

Yes

Statement (if applicable):

Data are deposited at the Dryad Digital Repository (<https://doi.org/10.5061/dryad.12jm63xw2>).

URL for review: https://datadryad.org/stash/share/0kK4EvaaL9gKPdeFQGhJWz_e_JvuK0BQFYK5eMnZQM

Conflict of interest

I/We declare we have no competing interests

Statement (if applicable):

CUST_STATE_CONFLICT :No data available.

Authors' contributions

This paper has multiple authors and our individual contributions were as below

Statement (if applicable):

NBF, ALB, KMM, and SGH conceived and planned the work. NBF, ALB, SJH, AGL, and SMD contributed to field sampling. ALB, SL, KMM led laboratory processing. NBF and ALB conducted analyses. All authors wrote, edited, and gave final approval for submission of the manuscript.

1 **Infected juvenile salmon can experience increased predation during**
2 **freshwater migration**

3 Nathan B. Furey^{*a}, Arthur L. Bass^b, Kristi M. Miller^c, Shaorong Li^c, Andrew G. Lotto^b, Stephen
4 J. Healy^d, S. Matthew Drenner^{ef}, and Scott G. Hinch^b

5
6 ^a Department of Biological Sciences, University of New Hampshire, Durham, USA

7 ^b Department of Forest and Conservation Sciences, University of British Columbia, Vancouver,
8 Canada

9 ^c Fisheries and Oceans Canada, Molecular Genetics Section, Pacific Biological Station,
10 Nanaimo, Canada

11 ^d Fisheries and Oceans Canada, Science Branch, Pacific Region, 4160 Marine Dr., West
12 Vancouver, BC, V7V 1N6, Canada

13 ^e Stillwater Sciences, 555 W. Fifth St, 35th floor, Los Angeles, CA 90013; Marine Science
14 Institute

15 ^f University of California Santa Barbara, Santa Barbara, USA

16 * Corresponding author: Nathan.Furey@unh.edu

18

Abstract

Predation risk for animal migrants can be impacted by physical condition. Although size- or condition-based selection is often observed, observing infection-based predation is rare due to the difficulties in assessing infectious agents in predated samples. We examined predation of outmigrating sockeye salmon (*Oncorhynchus nerka*) smolts by bull trout (*Salvelinus confluentus*) in southcentral British Columbia, Canada. We used a high-throughput quantitative polymerase chain reaction (qPCR) platform to screen for the presence of 17 infectious agents found in salmon and assess 14 host genes associated with viral responses. In one (2014) of the two years assessed (2014 and 2015), presence of infectious haematopoietic necrosis virus (IHNV) resulted in 16-25 times greater chance of predation; in 2015 IHNV was absent among all samples, predated or not. Thus, we provide further evidence that infection can impact predation risk in migrants. Some smolts with high IHNV loads also exhibited gene expression profiles consistent with a virus-induced disease state. Nine other infectious agents were observed between the two years, none of which were associated with increased selection by bull trout. In 2014, richness of infectious agents was also associated with greater predation risk. This is a rare demonstration of predator consumption resulting in selection for prey that carry infectious agents. The mechanism by which this selection occurs is not yet determined. By culling infectious agents from migrant populations, fish predators could provide an ecological benefit to prey.

Key-words

Predator-prey interactions, infectious hematopoietic virus, migratory culling, migration ecology, predation risk, Pacific salmon, pathogens, disease ecology

40

[revised manuscript text omitted]

showed positive relationship between predation and presence of *Ichthyophthirius multifiliis*
(Table 2). Lastly, the top-ranked 2015-liver model also suggested that infection with *Candidatus*
*Branchiomonas cysticola* was associated with reduced chance of predation risk, as it was found
in 100% of predated samples, but only two-thirds of predated samples (Table 1; Table 2).

Fork length and age

Among GLMs, the 2014-liver models and all 2015 models suggested that smaller fish were at
greater risk of predation (negative FL coefficient; Table 2). This relationship was consistent

among year-tissue combinations, with all models $\Delta AICc < 3$ containing FL, including the top
models. In 2014 samples, mean FL of smolts did not differ between IHN+ and IHN- smolts, in
both gill ($t = 0.46$, $df = 39$, $P = 0.64$), and liver ($t = -0.12$, $df = 40$, $P = 0.90$) tissues. Similarly, the
prevalence of IHN (0.875) was the same between age-1 (21 of 24) and age-2 (7 of 8) predated
smolts in 2014, and thus the inclusion of age-2 fish in our predated sample did not bias IHN
prevalence in predated fish.

Gene expression

PCAs on 2014 VDD gene expression data (the year in which IHNv was present) revealed three
smolts that exhibited strong separation along the first PC axis (most positive PC1; Figure 2). This
strong separation was apparent in both gill and liver tissues (Figure 2), and these three same
smolts had among the highest tissue-specific loads of IHNv (Figure 2). An additional fourth gill
2014 sample exhibited the same strong separation on the first PC axis, but was not included in
liver analyses due to poor reference gene performance. Aside from these individuals, PCA in
both years also demonstrated further shifts in VDD gene expression between predated and non-
predated smolts in at least one of the first two PC axes, regardless of year or tissue (Figure 2).
There was some tissue- and year-specific variability; separation for 2015 gill samples was most
clearly along PC1, while the other year-tissue combinations (aside from the three high-IHN-
loaded individuals) demonstrated stronger shifts along PC2 (Figure 2).

Sample degradation potential

All three reference genes demonstrated higher expression (lower Ct scores) in non-predated
samples in gills for both years (786d16.1P was significantly different in both years, COIL

significantly different in 2014, MrpL40 not significantly different in either year; t-test, $\alpha = 0.05$;
Figure 3). Conversely, all three reference genes demonstrated lower expression (higher Ct
scores) in non-predated samples in livers in both years (COIL significantly so in both years,
MrpL40 in 2015, and 786d16.1P in neither; Figure 3).

There was no significant relationship between IHN loads and condition score for predated, IHN+
smolts for both gill (Pearson correlation = 0.31, $df = 26$, $t = 1.68$, $P = 0.10$) and liver (Pearson
correlation coefficient = 0.22; $df = 10$, $t = 0.73$, $P = 0.48$). However, IHN+ gill samples came
from predated smolts with a significantly higher condition score (i.e. more digested) than
predated smolts that were IHN- (mean score IHN+ = 1.4, mean score IHN- = 0.5, t-test, $t = 2.60$,
$df=30$, $P = 0.01$). However, condition scores did not differ between IHN+ and IHN- predated
smolt samples in liver samples (mean score IHN+ = 1.5, mean score IHN- = 1.1, t-test, $t = 1.60$,
$df = 29$, $P = 0.12$).

**Potential interactions between IHN infection and size.**

~~To determine if IHN infection was confounded with fish size (as mortality in fish is often size-~~
~~selective; [46]), we compared fork length (FL) of IHN+ and IHN- smolts in 2014 samples. FL~~
~~was either measured directly, or estimated based on post-orbital hypural length or total length via~~
~~regression (Furey, unpublished data). Mean FL of smolts did not differ between IHN+ and IHN-~~
~~smolts, in both gill ($t = 0.46$, $df = 39$, $P = 0.64$), and liver ($t = -0.12$, $df = 40$, $P = 0.90$) tissues.~~

**Age differences**

~~Two age classes emigrate from Chilko Lake, British Columbia. Age-1 smolts constitute~~
~~on average ~96% of the migrating population, while age-2 are substantially larger but make up~~
~~~4% of the migration [47]. Of the 32 predated smolts assessed in 2014, 8 of them were age-2~~

~~(classified as those >116 mm FL; Brian Leaf, DFO, pers. comm.), all of which were predated.~~
~~The prevalence of IHNV (0.875) was the same between age-1 (21 of 24) and age-2 (7 of 8)~~
~~predated smolts in 2014, and thus the inclusion of age-2 fish in our predated sample did not bias~~
~~IHNV prevalence in predated fish.~~

Discussion

IHNV-positive smolts in 2014 were 16-to-25-times more likely to be predated than not. It is
uncommon for studies to make direct links between infection and predation risk outside of
experimental settings (but see [9,11,17,25,48,49]). Field studies on infection-based risk for fishes
have focused on avian predators [18,25]. Miller *et al.* [25], used an approach similar to ours to
demonstrate pathogen-based predation risk for wild salmon, with rhinoceros auklets (*Cerorhinca*
*monocerata*) feeding more heavily on marine sockeye salmon smolts infected with *Parvicapsula*
*spp.* parasites. Although not focused on predation, Jeffries *et al.* [23] found within our study
system that most (>80%) IHNV-positive Chilko sockeye salmon smolts tracked with acoustic
telemetry perished early in the migration, suggesting an association between IHNV infection and
smolt mortality, and our results indicate that predation is the likely mechanism for at least a
portion of this mortality.

IHNV is a single-stranded RNA virus that generates an acute, systemic disease that causes
necrosis of hematopoietic tissues of the kidney and spleen, as well as damage to several other
organs [50]. For juvenile sockeye, virulence is high [39] and can result in high mortality [40] 4 –
20 days after exposure [51], but outbreaks are generally limited to cooler waters below 15°C
[38]. IHNV's presence in Chilko Lake has been known for >40 years [41]. How infection of

328 IHNv results in increased predation by bull trout remains unclear. It is assumed that these
329 infectious agents either reduce a smolt's probability of escaping a predation attempt when
targeted [17], or increase the predator's propensity to target the smolt. Either possibility would
probably rely upon changing body coloration [52] or changing swimming behavior or
performance that can occur with infection [53,54]; IHN can result in lethargy, hyperactivity, or
erratic swimming [54]. Further work, such as experimental swim trials or high-resolution
tracking, is needed to determine the behavioral consequences of infection in migratory smolts,
and how this might result in increased predation risk. Such research would further develop our
understanding of how infections and movements, including migrations, interact to affect
individuals, populations, and communities [14,55].

Although IHNv demonstrated the strongest links between predation risk and infection,
Ichthyophthirius multifiliis -was also associated with increased predation risk via GLMs in three
of the four year-tissue combinations. Ichthyophthirius multifiliis was only found in predated
samples in both gill and liver tissues in 2015, and thus an odds ratio could not be calculated, but
in 2014 gill samples, this infectious agent was associated with a ~5-fold increase in predation
risk. This freshwater ciliate can induce mortality in fishes [56,57], including documented
epizootics in a wild population of spawning Fraser River sockeye salmon [58]. The parasite
targets epithelial tissue, and damage to gills leads to oxygen starvation and acidosis [57]. Thus,
Ichthyophthirius multifiliis can reduce swimming capacity of hosts [59]. In contrast to IHNv, the
likelihood of infection with this globally-distributed parasite increases with rising water
temperature (as a result of reduced generation time; [57,60]).

Lastly, one model (representing liver samples in 2015) suggested that infection of *Candidatus*
*Branchiomonas cysticola* resulted in reduced predation risk. But this infectious agent was quite
prevalent among all samples, with a prevalence rate between 90-100% except for predated liver
samples (67%), including 100% prevalence in predated and non-predated gill samples. Thus, our
results likely reflect an ubiquitous infectious agent in this population and caution
overinterpreting of the GLM result implying reduced predation risk. Multiple studies from our
research group have found high incidence of this pathogen with no accompanying physiological
or survival impact [61 – 63], including in this population of sockeye salmon smolts [64] and
research from Norway suggested that despite its 100% prevalence in Atlantic salmon gill
epitheliocysts, this bacteria was not associated with gill disease [65].

[revised manuscript text omitted]

(doi:10.1093/conphys/cox036)
33. Di Cicco E, Ferguson HW, Kaukinen KH, Schulze AD, Li S, Tabata A, Günther OP,
Mordecai G, Suttle CA, Miller KM. 2018. The same strain of Piscine orthoreovirus (PRV-
1) is involved in the development of different, but related, diseases in Atlantic and Pacific
Salmon in British Columbia. *Facets* 3:599-641.
34. Mordecai GJ et al. 2019 Endangered wild salmon infected by newly discovered viruses.
*eLife* 8, e47615. (doi:10.7554/eLife.47615)
35. Mordecai GJ et al. In press. Discovery and surveillance of viruses from salmon in British
Columbia using viral immune-response biomarkers, metatranscriptomics and high-
throughput RT-PCR. *Virus Evol* (doi:10.1093/ve/veaa069)
36. Furey NB, Hinch SG, Mesa MG, Beauchamp DA. 2016 Piscivorous fish exhibit
temperature-influenced binge feeding during an annual prey pulse. *J. Anim. Ecol.* 85,
1307-1317. (doi:10.1111/1365-2656.12565)
37. Clark TD et al. 2016 Tracking wild sockeye salmon smolts to the ocean reveals distinct
regions of nocturnal movement and high mortality. *Ecol. Appl.* 26, 959-978.
(doi:10.14286/2015CLARKTCHILKO)
38. Dixon P, Paley R, Alegria-Moran R, Oidtmann B. 2016 Epidemiological characteristics of
infectious hematopoietic necrosis virus (IHNV): a review. *Vet Res* 47, 63.
(doi:10.1186/s13567-016-0341-1)
39. Miller K, Traxler G, Kaukinen K, Li S, Richard J, Ginther N. 2007 Salmonid host
response to infectious hematopoietic necrosis (IHN) virus: Cellular receptors, viral
control, and novel pathways of defence. *Aquaculture* 272, 217–237.
(doi:10.1016/j.aquaculture.2007.08.041)
40. Lapatra S. 1998 Factors affecting pathogenicity of infectious hematopoietic necrosis virus
(IHNV) for salmonid fish. *J. Aquat. Anim. Health* 10, 121–131. (doi:10.1577/1548-
8667(1998)010<0121>
41. Williams I V, Amend DF. 1976 A natural epizootic of infectious hematopoietic necrosis
in fry of sockeye salmon (*Oncorhynchus nerka*) at Chilko Lake, British Columbia. *J. Fish.*
*Res. Board Canada* 33, 1564–1567.
42. Furey NB, Hinch SG, Lotto AG, Beauchamp DA. 2015 Extensive Feeding on Sockeye
Salmon *Oncorhynchus Nerka* Smolts by Bull Trout *Salvelinus Confluentus* during Initial
Outmigration into a Small, Unregulated and Inland British Columbia River. *Journal of*
*Fish Biology* 86, 392–401. (<https://doi.org/10.1111/jfb.12567>)
43. Miller KM, Li S, Kaukinen KH, Ginther N, Hammill E, Curtis JM, Patterson DA,
Sierocinski T, Donnison L, Pavlidis P, Hinch SG. 2011. Genomic signatures predict
migration and spawning failure in wild Canadian salmon. *Science* 331(6014):214-217.

44. R Core Team. 2018. R: A language and environment for statistical computing. R
Foundation for Statistical Computing, Vienna, Austria. URL <https://www.R-project.org/>
45. Oksanen J, Guillaume Blanchet F, Friendly M, Kindt R, Legendre P, McGlinn D, Minchin
PR, O'Hara RB, Simpson, GL, Solymos, P, Stevens MHH, Szoecs E, Wagner H. 2019.
vegan: Community Ecology Package. R package version 2.5-6. [https://CRAN.R-](https://CRAN.R-project.org/package=vegan)
[project.org/package=vegan](https://CRAN.R-project.org/package=vegan)
46. Irvine JR, Akenhead SA. 2013 Understanding Smolt Survival Trends in Sockeye Salmon.
*Marine and Coastal Fisheries* **5**, 303–328. (doi:10.1080/19425120.2013.831002)
47. [Barton, K. 2019 MuMIn: Multi-Model Inference. R package version 1.43.6.](https://CRAN.R-project.org/package=MuMIn)
<https://CRAN.R-project.org/package=MuMIn>
48. Hudson PJ, Dobson AP, Newborn D. 1992 Do Parasites make Prey Vulnerable to
Predation? Red Grouse and Parasites. *J. Anim. Ecol.* **61**, 681–692. (doi:10.2307/5623)
49. Genovart M, Negre N, Tavecchia G, Bistuer A, Parpal L, Oro D. 2010 The Young, the
Weak and the Sick: Evidence of Natural Selection by Predation. *PLOS ONE* **5**, e9774.
(doi:10.1371/journal.pone.0009774)
50. Bootland LM, Leong J-AC. 1999 Infectious haematopoietic necrosis virus. In *Fish*
*Diseases and Disorders* (eds P Woo, J Leatherland, D Bruno), pp. 66–109.
51. Kim CH, Dummer DM, Chiou PP, Leong JA. 1999 Truncated particles produced in fish
surviving infectious hematopoietic necrosis virus infection: mediators of persistence? *J.*
*Virol.* **73**, 843–9.
52. LaPatra SE, Barone L, Jones GR, Zon LI. 2000 Effects of Infectious Hematopoietic
Necrosis Virus and Infectious Pancreatic Necrosis Virus Infection on Hematopoietic
Precursors of the Zebrafish. *Blood Cells, Molecules, and Diseases* **26**, 445–452.
(doi:10.1006/bcmd.2000.0320)
53. Munday BL, Kwang J, Moody N. 2002 Betanodavirus infections of teleost fish: a review.
*J. Fish Dis.* **25**, 127–142.
54. Tierney KB, Farrell AP. 2004 The relationships between fish health, metabolic rate,
swimming performance and recovery in return-run sockeye salmon, *Oncorhynchus nerka*
(Walbaum). *J. Fish Dis.* **27**, 663–671. (doi:10.1111/j.1365-2761.2004.00590.x)
55. Binning SA, Shaw AK, Roche DG. 2017 Parasites and Host Performance: Incorporating
Infection into Our Understanding of Animal Movement. *Integr Comp Biol* **57**, 267–280.
(doi:10.1093/icb/ix024)
56. [Xu D-H, Klesius PH, Peatman E, Liu Z. 2011 Susceptibility of channel catfish, blue catfish](https://doi.org/10.1016/j.aquaculture.2010.10.012)
[and channel×blue catfish hybrid to *Ichthyophthirius multifiliis*. *Aquaculture* 311, 25–30.](https://doi.org/10.1016/j.aquaculture.2010.10.012)
[\(doi:10.1016/j.aquaculture.2010.10.012\)](https://doi.org/10.1016/j.aquaculture.2010.10.012)
57. [Dickerson, HW. 2011. *Ichthyophthirius multifiliis*. In Woo, P. T., & Buchmann, K. \(Eds.\).](https://doi.org/10.1016/j.aquaculture.2010.10.012)
[\(2011\). *Fish parasites: pathobiology and protection*. CABI.](https://doi.org/10.1016/j.aquaculture.2010.10.012)
58. [Traxler GS, Richard J, McDonald TE. 1998 *Ichthyophthirius multifiliis* \(Ich\) Epizootics in](https://doi.org/10.1577/1548-8667(1998)010<0143:IMIEIS>2.0.CO;2)
[Spawning Sockeye Salmon in British Columbia, Canada. *Journal of Aquatic Animal Health*](https://doi.org/10.1577/1548-8667(1998)010<0143:IMIEIS>2.0.CO;2)
[10, 143–151. \(doi:10.1577/1548-8667\(1998\)010<0143:IMIEIS>2.0.CO;2\)](https://doi.org/10.1577/1548-8667(1998)010<0143:IMIEIS>2.0.CO;2)
59. [Münderle M, Sures B, Taraschewski H. 2004 Influence of *Anguillicola crassus* \(Nematoda\)](https://doi.org/10.3354/dao060133)
[and *Ichthyophthirius multifiliis* \(Ciliophora\) on swimming activity of European eel *Anguilla*](https://doi.org/10.3354/dao060133)
[*anguilla*. *Diseases of Aquatic Organisms* 60, 133–139. \(doi:10.3354/dao060133\)](https://doi.org/10.3354/dao060133)

60. Maceda-Veiga A, Salvadó H, Vinyoles D, Sostoa AD. 2009 Outbreaks of *Ichthyophthirius*
multifiliis in Redtail Barbs *Barbus haasi* in a Mediterranean Stream during Drought. *Journal*
of Aquatic Animal Health 21, 189–194. (doi:10.1577/H08-054.1)
61. Healy SJ, Hinch SG, Bass AL, Furey NB, Welch DW, Rechisky EL, Eliason EJ, Lotto AG,
Miller KM. 2018 Transcriptome profiles relate to migration fate in hatchery steelhead
(*Oncorhynchus mykiss*) smolts. *Canadian Journal of Fisheries and Aquatic Sciences*
(doi:10.1139/cjfas-2017-0424)
62. Teffer AK, Bass AL, Miller KM, Patterson DA, Juanes F, Hinch SG. 2018 Infections,
fisheries capture, temperature, and host responses: multistressor influences on survival and
behaviour of adult Chinook salmon. *Canadian Journal of Fisheries and Aquatic Sciences*
(doi:10.1139/cjfas-2017-0491)
5663. Bass, A.L., Hinch, S.G., Teffer, A.K., Patterson, D.A. and Miller, K.M., 2019. Fisheries
capture and infectious agents are associated with travel rate and survival of Chinook
salmon during spawning migration. *Fish. Res.* 209, 156-166.
64. Stevenson CF, Bass AL, Furey NB, Miller KM, Li S, Rechisky EL, Porter AD, Welch
DW, Hinch SG. 2019 Infectious agents and gene expression differ between sockeye
salmon (*Oncorhynchus nerka*) smolt age classes but do not predict migration survival.
*Can. J. Fish. Aquat. Sci.* (doi: 10.1139/cjfas-2019-0113)
65. Gunnarsson GS, Karlsbakk E, Blindheim S, Plarre H, Imsland AK, Handeland S, Sveier H,
Nylund A. 2017 Temporal changes in infections with some pathogens associated with gill
disease in farmed Atlantic salmon (*Salmo salar* L). *Aquaculture* 468, 126–134.
(doi:10.1016/j.aquaculture.2016.10.011)
66. Bartholomew JL. 1998 Host Resistance to Infection by the Myxosporean Parasite
*Ceratomyxa shasta*: A Review. *Journal of Aquatic Animal Health* 10, 112–120.
(doi:10.1577/1548-8667(1998)010<0112:HRTIBT>2.0.CO;2)
5767. Kotob MH, Menanteau-Ledouble S, Kumar G, Abdelzaher M, El-Matbouli M. 2016 The
impact of co-infections on fish: a review. *Vet. Res.* 47, 1–12. (doi:10.1186/s13567-016-
0383-4)
5868. Bordes F, Morand S. 2011 The impact of multiple infections on wild animal hosts: a
review. *Infect. Ecol. Epidemiol.* 1, 7346. (doi:10.3402/iee.v1i0.7346)
5969. Lello J, Boag B, Fenton A, Stevenson IR, Hudson PJ. 2004 Competition and mutualism
among the gut helminths of a mammalian host. *Nature* 428, 840–844.
(doi:10.1038/nature02490)
6070. Natsopoulou ME, McMahon DP, Doublet V, Bryden J, Paxton RJ. 2015 Interspecific
competition in honeybee intracellular gut parasites is asymmetric and favours the spread
of an emerging infectious disease. *Proceedings of the Royal Society B: Biological*
*Sciences* 282, 20141896. (doi:10.1098/rspb.2014.1896)
6471. Telfer S, Lambin X, Birtles R, Beldomenico P, Burthe S, Paterson S, Begon M. 2010
Species Interactions in a Parasite Community Drive Infection Risk in a Wildlife
Population. *Science* 330, 243–246. (doi:10.1126/science.1190333)
6272. Evans AF et al. 2016 Avian Predation on Juvenile Salmonids: Spatial and Temporal
Analysis Based on Acoustic and Passive Integrated Transponder Tags. *Trans. Am. Fish.*
*Soc.* 145, 860–877. (doi:10.1080/00028487.2016.1150881)
6373. Wood CC. 1987 Predation of Juvenile Pacific Salmon by the Common Merganser (*Mergus*
merganser) on Eastern Vancouver Island. I: Predation during the Seaward Migration. *Can.*

[revised manuscript text omitted]

Tables

Table 1: List of infectious agents assessed in sockeye salmon smolts using qRT-PCR, the percentage of positives recorded across year-tissue combinations (\pm SE), and the odds-ratio of each infectious agent being found in a predated smolt over a non-predated smolt are given. Odds ratios in bold and noted with an asterisk(*) indicate significant Fisher exact test (fdr-corrected $P < 0.05$). Sample sizes are as follows for each year-tissue combination: 2014 predated (n = 32 for gills, n = 31 for livers); 2014 not predated (n = 30 each tissue); 2015 predated (n = 30 each tissue); 2015 not predated (n = 9 each tissue). For pPrevalence rates assessed from mixed-tissue samples in other studies of Chilko sockeye salmon smolts collected via the Strategic Salmon Health Initiative (SSHI) are given for 2012 (n = 54 – 56 smolts for each assay), 2013 (n = 85 – 89), and 2014 (n = 21 – 30) for comparison, only infectious agents that had at least one positive in a smolt are included.

Infectious agent	Assay name	Agent	Percent positives (predated / not predated)				odds-ratio (predated over not predated)				Prevalence (SSHI samples)		
			2014 gill positives	2014 liver positives	2015 gill positives	2015 liver positives	2014 gill odds-ratio	2014 liver odds-ratio	2015 gill odds-ratio	2015 liver odds-ratio	2012	2013	2014
Candidatus Branchiomonas cysticola	c_b_cys	Bacteria	100(\pm 0.0)	96.8(\pm3.2)	100(\pm 0.0)	66.7(\pm9.1)		2.1		0.0	98.2	100	100
Ceratomyxa shasta	ce_sha	Myxozoan									0.0	0.0	0.0
Dermocystidium salmonis	de_sal	Fungus/ Protozoan									1.8	0.0	0.0
Flavobacterium psychrophilum	fl_psy	Bacteria	87.5(\pm 5.8)	16.1(\pm 6.6)	76.7(\pm 7.7)	14.8(\pm 6.8)	2.9	1.7	3.9	1.4	5.4	6.7	17.2

	Candidatus	sch	Bacteria									0.0	0.0	0.0
	Syngnamydia													
	salmonis													
	Ichthyophthirius	ic_mul	Ciliate	28.1(±7.9)		16.7(±6.8)	7.4(±5.0)/	5.3				3.6	20.0	23.1
	multifiliis)/6.7(±4.6))/0.0(±0.0)	0.0(±0.0)							
	Infectious	ihn	Virus	87.5(±5.8)	35.5(±8.6)					25.8*	15.3*	0.0	0.0	3.3
	hematopoietic)/20(±7.3)	/3.3(±4.0)									
	necrosis virus)										
	Loma salmonae	lo_sal	Microspori									0.0	0.0	0.0
	(Loma Spp)		dium											
	Pacific salmon	pspv	Virus	9.4(±5.2)/	80.6(±7.1)	10(±5.5)/	48.1(±9.6)		2.1	0.9	3.2	78.6	96.6	93.3
	parvovirus			0.0(±0.0)	/66.7(±9.2)	11.1(±5.9))/22.2(10.4±)							
	Paranucleospora	pa_ther	Microspori									0.0	1.1	0.0
	theridion		dium											
	Parvicapsula	pa_min	Myxozoan				0.0(±0.0)/					0.0	1.1	0.0
	minibicornis						11.1(±5.9)							
	Parvicapsula	pa_pse	Myxozoan)					0.0	0.0	0.0
	pseudobranchic													
	ola													
	Piscichlamydia	pch_sal	Bacteria	15.6(±6.4)	3.2(±3.2)/							0.0	0.0	0.0
	salmonis)/0.0(±0.0)	0.0(±0.0)									
	Piscine reovirus	prv	Virus)								0.0	0.0	0.0
	Tetracapsuloides	te_bry	Myxozoan		6.5(±4.4)/							0.0	3.4	16.7
	bryosalmonae				0.0(±0.0)									
	Rickettsia-like	rlo		3.1(±3.1)/								0.0	1.1	0.0
	organism			0.0(±0.0)										
	Yersinia ruckeri	ye_ruc_glnA	Bacteria	9.4(±5.2)/		13.3(±6.2)						0.0	0.0	0.0
				0.0(±0.0))/0.0(±0.0))							

**Table 2:** Summary of generalized linear models (GLMs) describing relationships
 between predation status (binomial) and the presence of infectious agents and fork length
 (FL). Candidate models are ranked by AICc, and only models with $\Delta AICc < 3$ are shown.
 The top-ranked model is in bold. First numeric value given for each model is the
 intercept, and coefficients are shown for each explanatory variable. Infectious agents are
 labelled as per their assay name (Table 1). Positive coefficients indicate increased
 probability of predation (negative coefficients associated with reduced predation risk).

2014 - Gill			
Model	AICc	$\Delta AICc$	
$\sim -2.54 + \text{ihnv}(+3.64) + \text{ic_mul}(+2.20)$	46.6	0	
$\sim -1.75 + \text{ihnv}(+4.53) + \text{ic_mul}(+2.44) + \text{fl_psy}(-1.70)$	47.4	0.8	
$\sim -2.54 + \text{ihnv}(+3.49) + \text{ic_mul}(+2.20) + \text{pch_sal}(+16.14)$	47.7	1.04	
$\sim -2.56 + \text{ihnv}(+3.51) + \text{ic_mul}(+2.25) + \text{pspv}(+16.61)$	47.7	1.12	
$\sim -2.08 + \text{ihnv}(+3.52)$	48.5	1.91	
$\sim +1.63 + \text{FL}(-0.04) + \text{ihnv}(+3.60) + \text{ic_mul}(+2.07)$	48.6	1.95	
$\sim -2.08 + \text{ihnv}(+3.36) + \text{pch_sal}(+16.29)$	49.4	2.79	
$\sim -1.39 + \text{ihnv}(+4.21) + \text{fl_psy}(-1.39)$	49.6	2.94	

2014 - Liver			
Model	AICc	$\Delta AICc$	
$\sim +12.29 + \text{FL}(-0.14) + \text{ihnv}(+3.56)$	61.1	0	
$\sim -4.12 + \text{FL}(-0.14) + \text{ihnv}(+3.50) + \text{cb_cys}(+16.64)$	62	0.91	
$\sim +12.22 + \text{FL}(-0.14) + \text{ihnv}(+3.55) + \text{pspv}(+0.16)$	63.3	2.29	
$\sim +12.31 + \text{FL}(-0.14) + \text{ihnv}(+3.57) + \text{fl_psy}(-0.15)$	63.4	2.32	

2015 - Gill			
Model	AICc	$\Delta AICc$	
$\sim +22.21 + \text{FL}(-0.27) + \text{ic_mul}(+18.63)$	31	0	
$\sim +22.56 + \text{FL}(-0.29) + \text{ic_mul}(+17.77) + \text{fl_psy}(+1.66)$	31.2	0.19	
$\sim +23.11 + \text{FL}(-0.29) + \text{ic_mul}(+19.91) + \text{ye_ruc_glA}(+18.88)$	31.3	0.35	
$\sim +22.35 + \text{FL}(-0.28) + \text{fl_psy}(+2.38)$	31.4	0.39	
$\sim +22.10 + \text{FL}(-0.27) + \text{fl_psy}(+2.25) + \text{ye_ruc_glA}(+16.74)$	33	2.08	
$\sim +24.52 + \text{FL}(-0.31) + \text{fl_psy}(+2.66) + \text{pspv}(-1.10)$	33.4	2.47	
$\sim +22.36 + \text{FL}(-0.28) + \text{ic_mul}(18.71) + \text{pspv}(+0.43)$	33.5	2.5	

2015 - Liver

825

Model	AICc	ΔAICc
$\sim +39.19 + FL(-0.25) + \text{cb_cys}(-19.14) + \text{ic_mul}(+20.09)$	29.3	0
$\sim +35.77 + FL(-0.21) + \text{cb_cys}(-18.84)$	29.8	0.57
$\sim +36.07 + FL(-0.22) + \text{cb_cys}(-18.61) + \text{pspv}(+1.35)$	31.1	1.78

Figure Legends

**Figure 1** Shannon diversity index of infectious agents found in gill and liver tissue of
sockeye salmon smolts between those predated and not predated by bull trout. Asterisks
indicate a significant difference in median pathogen richness between predated and non-
predated groups (Mann Whitney U-test, $\alpha = 0.05$).

**Figure 2** PCA of gene expression of 13 genes used in diagnosing viral disease
development [40] in sockeye salmon smolt samples from 2014 and 2015 and in gill and
835 liver tissues. Circle size symbolizes IHNV loads (represented as the log of the estimated
copy number + 1). Red ellipses enclose the same samples (three in liver samples, with an
additional fourth in gill samples) that separate via the first PC axis and have high IHNV
loads (IHNV+), potentially indicative of an active disease state. Percentages in
parentheses indicate the percent variability explained among gene expression by that
specific axis.

**Figure 3:** Expression levels (via cycle threshold [Ct] values) of three reference genes
between years and tissues of juvenile sockeye salmon smolts. “Predated” indicates
predated samples, “Not” indicates control, or non-predated, sample. Asterisk (*) indicates
significant difference in Ct score between predated and non-predated samples for a given
reference gene (t-test, $\alpha = 0.05$).

Figure 1 Shannon diversity index of infectious agents found in gill and liver tissue of sockeye salmon smolts between those predated and not predated by bull trout. Asterisks indicate a significant difference in median pathogen richness between predated and non-predated groups (Mann-Whitney U-test, $\alpha = 0.05$).

177x152mm (300 x 300 DPI)

Figure 2 PCA of gene expression of 13 genes used in diagnosing viral disease development [40] in sockeye salmon smolt samples from 2014 and 2015 and in gill and liver tissues. Circle size symbolizes IHNv loads (represented as the log of the estimated copy number + 1). Red ellipses enclose the same samples (three in liver samples, with an additional fourth in gill samples) that separate via the first PC axis and have high IHNv loads (IHNv+), potentially indicative of an active disease state. Percentages in parentheses indicate the percent variability explained among gene expression by that specific axis.

177x203mm (300 x 300 DPI)

Figure 3: Expression levels (via cycle threshold [Ct] values) of three reference genes between years and tissues of juvenile sockeye salmon smolts. "Predated" indicates predated samples, "Not" indicates control, or non-predated, sample. Asterisk (*) indicates significant difference in Ct score between predated and non-predated samples for a given reference gene (t-test, $\alpha = 0.05$).

152x203mm (300 x 300 DPI)

1 **Infected juvenile salmon can experience increased predation during**
2 **freshwater migration**

3 Nathan B. Furey^{*a}, Arthur L. Bass^b, Kristi M. Miller^c, Shaorong Li^c, Andrew G. Lotto^b, Stephen
4 J. Healy^d, S. Matthew Drenner^{ef}, and Scott G. Hinch^b

5
6 ^a Department of Biological Sciences, University of New Hampshire, Durham, USA

7 ^b Department of Forest and Conservation Sciences, University of British Columbia, Vancouver,
8 Canada

9 ^c Fisheries and Oceans Canada, Molecular Genetics Section, Pacific Biological Station,
10 Nanaimo, Canada

11 ^d Fisheries and Oceans Canada, Science Branch, Pacific Region, 4160 Marine Dr., West
12 Vancouver, BC, V7V 1N6, Canada

13 ^e Stillwater Sciences, 555 W. Fifth St, 35th floor, Los Angeles, CA 90013; Marine Science
14 Institute

15 ^f University of California Santa Barbara, Santa Barbara, USA

16 * Corresponding author: Nathan.Furey@unh.edu

18

Abstract

Predation risk for animal migrants can be impacted by physical condition. Although size- or condition-based selection is often observed, observing infection-based predation is rare due to the difficulties in assessing infectious agents in predated samples. We examined predation of outmigrating sockeye salmon (*Oncorhynchus nerka*) smolts by bull trout (*Salvelinus confluentus*) in southcentral British Columbia, Canada. We used a high-throughput quantitative polymerase chain reaction (qPCR) platform to screen for the presence of 17 infectious agents found in salmon and assess 14 host genes associated with viral responses. In one (2014) of the two years assessed (2014 and 2015), presence of infectious haematopoietic necrosis virus (IHNV) resulted in 16-25 times greater chance of predation; in 2015 IHNV was absent among all samples, predated or not. Thus, we provide further evidence that infection can impact predation risk in migrants. Some smolts with high IHNV loads also exhibited gene expression profiles consistent with a virus-induced disease state. Nine other infectious agents were observed between the two years, none of which were associated with increased selection by bull trout. In 2014, richness of infectious agents was also associated with greater predation risk. This is a rare demonstration of predator consumption resulting in selection for prey that carry infectious agents. The mechanism by which this selection occurs is not yet determined. By culling infectious agents from migrant populations, fish predators could provide an ecological benefit to prey.

Key-words

Predator-prey interactions, infectious hematopoietic virus, migratory culling, migration ecology, predation risk, Pacific salmon, pathogens, disease ecology

40

[revised manuscript text omitted]

showed positive relationship between predation and presence of *Ichthyophthirius multifiliis*
(Table 2). Lastly, the top-ranked 2015-liver model also suggested that infection with *Candidatus*
*Branchiomonas cysticola* was associated with reduced chance of predation risk, as it was found
in 100% of predated samples, but only two-thirds of predated samples (Table 1; Table 2).

**Fork length and age**

Among GLMs, the 2014-liver models and all 2015 models suggested that smaller fish were at
greater risk of predation (negative FL coefficient; Table 2). This relationship was consistent

among year-tissue combinations, with all models $\Delta AICc < 3$ containing FL, including the top
models. In 2014 samples, mean FL of smolts did not differ between IHN+ and IHN- smolts, in
both gill ($t = 0.46$, $df = 39$, $P = 0.64$), and liver ($t = -0.12$, $df = 40$, $P = 0.90$) tissues. Similarly, the
prevalence of IHN (0.875) was the same between age-1 (21 of 24) and age-2 (7 of 8) predated
smolts in 2014, and thus the inclusion of age-2 fish in our predated sample did not bias IHN
prevalence in predated fish.

**Gene expression**

[revised manuscript text omitted]

Lastly, one model (representing liver samples in 2015) suggested that infection of *Candidatus*
*Branchiomonas cysticola* resulted in reduced predation risk. But this infectious agent was quite
prevalent among all samples, with a prevalence rate between 90-100% except for predated liver
samples (67%), including 100% prevalence in predated and non-predated gill samples. Thus, our
results likely reflect an ubiquitous infectious agent in this population and caution
overinterpreting of the GLM result implying reduced predation risk. Multiple studies from our
research group have found high incidence of this pathogen with no accompanying physiological
or survival impact [61 – 63], including in this population of sockeye salmon smolts [64] and
research from Norway suggested that despite its 100% prevalence in Atlantic salmon gill
epitheliocysts, this bacteria was not associated with gill disease [65].

[revised manuscript text omitted]

(doi:10.1093/conphys/cox036)
33. Di Cicco E, Ferguson HW, Kaukinen KH, Schulze AD, Li S, Tabata A, Günther OP,
Mordecai G, Suttle CA, Miller KM. 2018. The same strain of Piscine orthoreovirus (PRV-
1) is involved in the development of different, but related, diseases in Atlantic and Pacific
Salmon in British Columbia. *Facets* 3:599-641.
34. Mordecai GJ et al. 2019 Endangered wild salmon infected by newly discovered viruses.
*eLife* 8, e47615. (doi:10.7554/eLife.47615)
35. Mordecai GJ et al. In press. Discovery and surveillance of viruses from salmon in British
Columbia using viral immune-response biomarkers, metatranscriptomics and high-
throughput RT-PCR. *Virus Evol* (doi:10.1093/ve/veaa069)
36. Furey NB, Hinch SG, Mesa MG, Beauchamp DA. 2016 Piscivorous fish exhibit
temperature-influenced binge feeding during an annual prey pulse. *J. Anim. Ecol.* 85,
1307-1317. (doi:10.1111/1365-2656.12565)
37. Clark TD et al. 2016 Tracking wild sockeye salmon smolts to the ocean reveals distinct
regions of nocturnal movement and high mortality. *Ecol. Appl.* 26, 959-978.
(doi:10.14286/2015CLARKTCHILKO)
38. Dixon P, Paley R, Alegria-Moran R, Oidtmann B. 2016 Epidemiological characteristics of
infectious hematopoietic necrosis virus (IHNV): a review. *Vet Res* 47, 63.
(doi:10.1186/s13567-016-0341-1)
39. Miller K, Traxler G, Kaukinen K, Li S, Richard J, Ginther N. 2007 Salmonid host
response to infectious hematopoietic necrosis (IHN) virus: Cellular receptors, viral
control, and novel pathways of defence. *Aquaculture* 272, 217–237.
(doi:10.1016/j.aquaculture.2007.08.041)
40. Lapatra S. 1998 Factors affecting pathogenicity of infectious hematopoietic necrosis virus
(IHNV) for salmonid fish. *J. Aquat. Anim. Health* 10, 121–131. (doi:10.1577/1548-
8667(1998)010<0121>
41. Williams I V, Amend DF. 1976 A natural epizootic of infectious hematopoietic necrosis
in fry of sockeye salmon (*Oncorhynchus nerka*) at Chilko Lake, British Columbia. *J. Fish.*
*Res. Board Canada* 33, 1564–1567.
42. Furey NB, Hinch SG, Lotto AG, Beauchamp DA. 2015 Extensive Feeding on Sockeye
Salmon *Oncorhynchus Nerka* Smolts by Bull Trout *Salvelinus Confluentus* during Initial
Outmigration into a Small, Unregulated and Inland British Columbia River. *Journal of*
*Fish Biology* 86, 392–401. (<https://doi.org/10.1111/jfb.12567>)
43. Miller KM, Li S, Kaukinen KH, Ginther N, Hammill E, Curtis JM, Patterson DA,
Sierocinski T, Donnison L, Pavlidis P, Hinch SG. 2011. Genomic signatures predict
migration and spawning failure in wild Canadian salmon. *Science* 331(6014):214-217.

44. R Core Team. 2018. R: A language and environment for statistical computing. R
Foundation for Statistical Computing, Vienna, Austria. URL <https://www.R-project.org/>
45. Oksanen J, Guillaume Blanchet F, Friendly M, Kindt R, Legendre P, McGlinn D, Minchin
PR, O'Hara RB, Simpson, GL, Solymos, P, Stevens MHH, Szoecs E, Wagner H. 2019.
vegan: Community Ecology Package. R package version 2.5-6. [https://CRAN.R-](https://CRAN.R-project.org/package=vegan)
[project.org/package=vegan](https://CRAN.R-project.org/package=vegan)
46. Irvine JR, Akenhead SA. 2013 Understanding Smolt Survival Trends in Sockeye Salmon.
*Marine and Coastal Fisheries* **5**, 303–328. (doi:10.1080/19425120.2013.831002)
47. Barton, K. 2019 MuMIn: Multi-Model Inference. R package version 1.43.6.
<https://CRAN.R-project.org/package=MuMIn>
48. Hudson PJ, Dobson AP, Newborn D. 1992 Do Parasites make Prey Vulnerable to
Predation? Red Grouse and Parasites. *J. Anim. Ecol.* **61**, 681–692. (doi:10.2307/5623)
49. Genovart M, Negre N, Tavecchia G, Bistuer A, Parpal L, Oro D. 2010 The Young, the
Weak and the Sick: Evidence of Natural Selection by Predation. *PLOS ONE* **5**, e9774.
(doi:10.1371/journal.pone.0009774)
50. Bootland LM, Leong J-AC. 1999 Infectious haematopoietic necrosis virus. In *Fish*
*Diseases and Disorders* (eds P Woo, J Leatherland, D Bruno), pp. 66–109.
51. Kim CH, Dummer DM, Chiou PP, Leong JA. 1999 Truncated particles produced in fish
surviving infectious hematopoietic necrosis virus infection: mediators of persistence? *J.*
*Virol.* **73**, 843–9.
52. LaPatra SE, Barone L, Jones GR, Zon LI. 2000 Effects of Infectious Hematopoietic
Necrosis Virus and Infectious Pancreatic Necrosis Virus Infection on Hematopoietic
Precursors of the Zebrafish. *Blood Cells, Molecules, and Diseases* **26**, 445–452.
(doi:10.1006/bcmd.2000.0320)
53. Munday BL, Kwang J, Moody N. 2002 Betanodavirus infections of teleost fish: a review.
*J. Fish Dis.* **25**, 127–142.
54. Tierney KB, Farrell AP. 2004 The relationships between fish health, metabolic rate,
swimming performance and recovery in return-run sockeye salmon, *Oncorhynchus nerka*
(Walbaum). *J. Fish Dis.* **27**, 663–671. (doi:10.1111/j.1365-2761.2004.00590.x)
55. Binning SA, Shaw AK, Roche DG. 2017 Parasites and Host Performance: Incorporating
Infection into Our Understanding of Animal Movement. *Integr Comp Biol* **57**, 267–280.
(doi:10.1093/icb/icx024)
56. Xu D-H, Klesius PH, Peatman E, Liu Z. 2011 Susceptibility of channel catfish, blue catfish
and channel×blue catfish hybrid to *Ichthyophthirius multifiliis*. *Aquaculture* **311**, 25–30.
(doi:10.1016/j.aquaculture.2010.10.012)
57. Dickerson, HW. 2011. *Ichthyophthirius multifiliis*. In Woo, P. T., & Buchmann, K. (Eds.).
(2011). *Fish parasites: pathobiology and protection*. CABI.
58. Traxler GS, Richard J, McDonald TE. 1998 *Ichthyophthirius multifiliis* (Ich) Epizootics in
Spawning Sockeye Salmon in British Columbia, Canada. *Journal of Aquatic Animal Health*
**10**, 143–151. (doi:10.1577/1548-8667(1998)010<0143:IMIEIS>2.0.CO;2)
59. Münderle M, Sures B, Taraschewski H. 2004 Influence of *Anguillicola crassus* (Nematoda)
and *Ichthyophthirius multifiliis* (Ciliophora) on swimming activity of European eel *Anguilla*
*anguilla*. *Diseases of Aquatic Organisms* **60**, 133–139. (doi:10.3354/dao060133)

60. Maceda-Veiga A, Salvadó H, Vinyoles D, Sostoa AD. 2009 Outbreaks of *Ichthyophthirius*
*multifiliis* in Redtail Barbs *Barbus haasi* in a Mediterranean Stream during Drought. *Journal*
*of Aquatic Animal Health* 21, 189–194. (doi:10.1577/H08-054.1)
61. Healy SJ, Hinch SG, Bass AL, Furey NB, Welch DW, Rechisky EL, Eliason EJ, Lotto AG,
Miller KM. 2018 Transcriptome profiles relate to migration fate in hatchery steelhead
(*Oncorhynchus mykiss*) smolts. *Canadian Journal of Fisheries and Aquatic Sciences*
(doi:10.1139/cjfas-2017-0424)
62. Teffer AK, Bass AL, Miller KM, Patterson DA, Juanes F, Hinch SG. 2018 Infections,
fisheries capture, temperature, and host responses: multistressor influences on survival and
behaviour of adult Chinook salmon. *Canadian Journal of Fisheries and Aquatic Sciences*
(doi:10.1139/cjfas-2017-0491)
63. Bass, A.L., Hinch, S.G., Teffer, A.K., Patterson, D.A. and Miller, K.M., 2019. Fisheries
capture and infectious agents are associated with travel rate and survival of Chinook
salmon during spawning migration. *Fish. Res.* **209**, 156-166.
64. Stevenson CF, Bass AL, Furey NB, Miller KM, Li S, Rechisky EL, Porter AD, Welch
DW, Hinch SG. 2019 Infectious agents and gene expression differ between sockeye
salmon (*Oncorhynchus nerka*) smolt age classes but do not predict migration survival.
*Can. J. Fish. Aquat. Sci.* (doi: 10.1139/cjfas-2019-0113)
65. Gunnarsson GS, Karlsbakk E, Blindheim S, Plarre H, Imsland AK, Handeland S, Sveier H,
Nylund A. 2017 Temporal changes in infections with some pathogens associated with gill
disease in farmed Atlantic salmon (*Salmo salar* L). *Aquaculture* 468, 126–134.
(doi:10.1016/j.aquaculture.2016.10.011)
66. Bartholomew JL. 1998 Host Resistance to Infection by the Myxosporean Parasite
*Ceratomyxa shasta*: A Review. *Journal of Aquatic Animal Health* 10, 112–120.
(doi:10.1577/1548-8667(1998)010<0112:HRTIBT>2.0.CO;2)
67. Kotob MH, Menanteau-Ledouble S, Kumar G, Abdelzaher M, El-Matbouli M. 2016 The
impact of co-infections on fish: a review. *Vet. Res.* **47**, 1–12. (doi:10.1186/s13567-016-
0383-4)
68. Bordes F, Morand S. 2011 The impact of multiple infections on wild animal hosts: a
review. *Infect. Ecol. Epidemiol.* **1**, 7346. (doi:10.3402/iee.v1i0.7346)
69. Lello J, Boag B, Fenton A, Stevenson IR, Hudson PJ. 2004 Competition and mutualism
among the gut helminths of a mammalian host. *Nature* **428**, 840–844.
(doi:10.1038/nature02490)
70. Natsopoulou ME, McMahan DP, Doublet V, Bryden J, Paxton RJ. 2015 Interspecific
competition in honeybee intracellular gut parasites is asymmetric and favours the spread
of an emerging infectious disease. *Proceedings of the Royal Society B: Biological*
*Sciences* **282**, 20141896. (doi:10.1098/rspb.2014.1896)
71. Telfer S, Lambin X, Birtles R, Beldomenico P, Burthe S, Paterson S, Begon M. 2010
Species Interactions in a Parasite Community Drive Infection Risk in a Wildlife
Population. *Science* **330**, 243–246. (doi:10.1126/science.1190333)
72. Evans AF et al. 2016 Avian Predation on Juvenile Salmonids: Spatial and Temporal Analysis
Based on Acoustic and Passive Integrated Transponder Tags. *Trans. Am. Fish. Soc.* 145,
860–877. (doi:10.1080/00028487.2016.1150881)
73. Wood CC. 1987 Predation of Juvenile Pacific Salmon by the Common Merganser (*Mergus*
*merganser*) on Eastern Vancouver Island. I: Predation during the Seaward Migration. *Can.*

[revised manuscript text omitted]

Tables

Table 1: List of infectious agents assessed in sockeye salmon smolts using qRT-PCR, the percentage of positives recorded across year-tissue combinations (\pm SE), and the odds-ratio of each infectious agent being found in a predated smolt over a non-predated smolt are given. Odds ratios in bold and noted with an asterisk(*) indicate significant Fisher exact test (fdr-corrected $P < 0.05$). Sample sizes are as follows for each year-tissue combination: 2014 predated (n = 32 for gills, n = 31 for livers); 2014 not predated (n = 30 each tissue); 2015 predated (n = 30 each tissue); 2015 not predated (n = 9 each tissue). Prevalence rates assessed from mixed-tissue samples of Chilko sockeye salmon smolts collected via the Strategic Salmon Health Initiative (SSHI) are given for 2012 (n = 54 – 56 smolts for each assay), 2013 (n = 85 – 89), and 2014 (n = 21 – 30) for comparison

Infectious agent	Assay name	Agent	Percent positives (predated / not predated)				odds-ratio (predated over not predated)				Prevalence (SSHI samples)		
			2014 gill positives	2014 liver positives	2015 gill positives	2015 liver positives	2014 gill odds-ratio	2014 liver odds-ratio	2015 gill odds-ratio	2015 liver odds-ratio	2012	2013	2014
Candidatus Branchiomonas cysticola	c_b_cys	Bacteria	100(\pm 0.0) /100(\pm 0.0)	96.8(\pm 3.2) /93.3(4.6 \pm)	100(\pm 0.0) /100(\pm 0.0)	66.7(\pm 9.1) /100(\pm 0.0)	2.1			0.0	98.2	100	100
Ceratomyxa shasta	ce_sha	Myxozoan									0.0	0.0	0.0
Dermocystidium salmonis	de_sal	Fungus/ Protozoan									1.8	0.0	0.0
Flavobacterium psychrophilum	fl_psy	Bacteria	87.5(\pm 5.8) /70(\pm 8.4)	16.1(\pm 6.6) /10(\pm 5.6)	76.7(\pm 7.7) /44.4(\pm 5.0)	14.8(\pm 6.8) /11.1(\pm 6.4)	2.9	1.7	3.9	1.4	5.4	6.7	17.2
Candidatus	sch	Bacteria									0.0	0.0	0.0

Syngnamydia
salmonis
Ichthyophthirius	ic_mul	Ciliate	28.1(±7.9		16.7(±6.8	7.4(±5.0)/	5.3			3.6	20.0	23.1
multifiliis) / 6.7(±4.6) / 0.0(±0.0	0.0(±0.0)
9))							
Infectious	ihnv	Virus	87.5(±5.8	35.5(±8.6)			25.8*	15.3*		0.0	0.0	3.3
hematopoietic) / 20(±7.3) / 3.3(±4.0)
necrosis virus)
Loma salmonae	lo_sal	Microspori								0.0	0.0	0.0
(Loma Spp)		dium
Pacific salmon	pspv	Virus	9.4(±5.2)/	80.6(±7.1)	10(±5.5)/	48.1(±9.6	2.1	0.9	3.2	78.6	96.6	93.3
parvovirus			0.0(±0.0)) / 66.7(±9.2) / 11.1(±5.9) / 22.2(10.
17))	4±)							
Paranucleospora	pa_ther	Microspori								0.0	1.1	0.0
theridion		dium
Parvicapsula	pa_min	Myxozoan				0.0(±0.0)/				0.0	1.1	0.0
minibicornis						11.1(±5.9
22)						
Parvicapsula	pa_pse	Myxozoan								0.0	0.0	0.0
pseudobranchic
ola
Piscichlamydia	pch_sal	Bacteria	15.6(±6.4	3.2(±3.2)/						0.0	0.0	0.0
salmonis) / 0.0(±0.0) / 0.0(±0.0)
28)									
Piscine reovirus	prv	Virus								0.0	0.0	0.0
Tetracapsuloides	te_bry	Myxozoan		6.5(±4.4)/						0.0	3.4	16.7
bryosalmonae				0.0(±0.0)
Ricksettia-like	rlo		3.1(±3.1)/							0.0	1.1	0.0
organism			0.0(±0.0)
Yersinia ruckeri	ye_ruc_gl	Bacteria	9.4(±5.2)/		13.3(±6.2					0.0	0.0	0.0
nA		0.0(±0.0)) / 0.0(±0.0
36)							

**Table 2:** Summary of generalized linear models (GLMs) describing relationships
 between predation status (binomial) and the presence of infectious agents and fork length
 (FL). Candidate models are ranked by AICc, and only models with $\Delta\text{AICc} < 3$ are shown.
 The top-ranked model is in bold. First numeric value given for each model is the
 intercept, and coefficients are shown for each explanatory variable. Infectious agents are
 labelled as per their assay name (Table 1). Positive coefficients indicate increased
 probability of predation (negative coefficients associated with reduced predation risk).

2014 - Gill			
Model	AICc	ΔAICc	
~ -2.54 + ihnv(+3.64) + ic_mul(+2.20)	46.6	0	
~ -1.75 + ihnv (+4.53) + ic_mul(+2.44) + fl_psy(-1.70)	47.4	0.8	
~ -2.54 + ihnv(+3.49) + ic_mul(+2.20) + pch_sal(+16.14)	47.7	1.04	
~ -2.56 + ihnv(+3.51) + ic_mul(+2.25) + pspv(+16.61)	47.7	1.12	
~ -2.08 + ihnv(+3.52)	48.5	1.91	
~ +1.63 + FL(-0.04) + ihnv(+3.60) + ic_mul(+2.07)	48.6	1.95	
~ -2.08 + ihnv(+3.36) + pch_sal(+16.29)	49.4	2.79	
~ -1.39 + ihnv(+4.21) + fl_psy(-1.39)	49.6	2.94	
2014 - Liver			
Model	AICc	ΔAICc	
~ +12.29 + FL(-0.14) + ihnv(+3.56)	61.1	0	
~ -4.12 + FL(-0.14) + ihnv(+3.50) + cb_cys(+16.64)	62	0.91	
~ +12.22 + FL(-0.14) + ihnv(+3.55) + pspv(+0.16)	63.3	2.29	
~ +12.31 + FL(-0.14) + ihnv(+3.57) + fl_psy(-0.15)	63.4	2.32	
2015 - Gill			
Model	AICc	ΔAICc	
~ +22.21 + FL(-0.27) + ic_mul(+18.63)	31	0	
~ +22.56 + FL(-0.29) + ic_mul(+17.77) + fl_psy(+1.66)	31.2	0.19	
~ +23.11 + FL(-0.29) + ic_mul(+19.91) + ye_ruc_gIA(+18.88)	31.3	0.35	
~ +22.35 + FL(-0.28) + fl_psy(+2.38)	31.4	0.39	
~ +22.10 + FL(-0.27) + fl_psy(+2.25) + ye_ruc_gIA(+16.74)	33	2.08	
~ +24.52 + FL(-0.31) + fl_psy(+2.66) + pspv(-1.10)	33.4	2.47	
~ +22.36 + FL(-0.28) + ic_mul(18.71) + pspv(+0.43)	33.5	2.5	
2015 - Liver			

Model	AICc	Δ AICc
$\sim +39.19 + \text{FL}(-0.25) + \text{cb_cys}(-19.14) + \text{ic_mul}(+20.09)$	29.3	0
$\sim +35.77 + \text{FL}(-0.21) + \text{cb_cys}(-18.84)$	29.8	0.57
$\sim +36.07 + \text{FL}(-0.22) + \text{cb_cys}(-18.61) + \text{pspv}(+1.35)$	31.1	1.78

**Figure Legends**

**Figure 1** Shannon diversity index of infectious agents found in gill and liver tissue of
sockeye salmon smolts between those predated and not predated by bull trout. Asterisks
indicate a significant difference in median pathogen richness between predated and non-
predated groups (Mann Whitney U-test, $\alpha = 0.05$).

**Figure 2** PCA of gene expression of 13 genes used in diagnosing viral disease
development [40] in sockeye salmon smolt samples from 2014 and 2015 and in gill and
liver tissues. Circle size symbolizes IHNv loads (represented as the log of the estimated
copy number + 1). Red ellipses enclose the same samples (three in liver samples, with an
additional fourth in gill samples) that separate via the first PC axis and have high IHNv
loads (IHNv+), potentially indicative of an active disease state. Percentages in
parentheses indicate the percent variability explained among gene expression by that
specific axis.

**Figure 3:** Expression levels (via cycle threshold [Ct] values) of three reference genes
between years and tissues of juvenile sockeye salmon smolts. “Predated” indicates
predated samples, “Not” indicates control, or non-predated, sample. Asterisk (*) indicates
significant difference in Ct score between predated and non-predated samples for a given
reference gene (t-test, $\alpha = 0.05$).

Associate Editor Comments to Author:

Given that the authors seem to have tried hard to improve the paper but one of the reviewers still has
some concerns, it is fair to offer the authors a final opportunity to revise, but they should be aware this
is the final opportunity they will have.

***Response: Thank you, and we appreciate that our hard work has been evident. We understand that***
***this is our final opportunity, and hope the reviewer appreciates the lengths we went to meet their***
***demands. We also hope the Associate Editor and reviewer can appreciate the value of this story – that***
***even with a complicated system and limited sample size, we were able to observe this link between***
***infection and predation. Regardless, we appreciate the efforts of all of the reviewers that have been***
***involved with this manuscript.***

***All line numbers referred to in our response below correspond to the track-changes document (rather***
***than the “clean” version).***

Reviewer comments to Author:

Reviewer: 2

Comments to the Author(s)

Comment #1: In general, Furey et al. did a good job addressing my comments, especially in the
introduction. I still have some large concerns with the analysis, that I believe the authors can address
relatively quickly, if given the opportunity. I also have some concerns that the authors tend to oversell
their results in the discussion. Specifically, the results show that the only disease that appeared to
increase the susceptibility to predation was IHNv, which was only present in one year. In contrast, other
diseases that appeared to be much more prevalent, did not increase predation rates. However, the
authors did not mention this dichotomy in the discussion at all. I think this is a major oversight that can
have some major implications. I recommend the authors temper their conclusions in the discussion to
better represent that they only found that one disease increased predation rates, while other diseases
appeared to have minimal effect.

Perhaps there are differences in these diseases and the way that they influence fish behavior that
influence the fish's susceptibility to predation.

***Response #1: We appreciate the reviewer felt that overall, we did a good job in addressing their***
***comments. Focusing on IHNv's impacts to us is not “overselling” the results relative to the pathogens***
***that are not linked to mortality, but rather we focus on this result because it is so striking (and makes***
***sense, given the literature on this infectious agent). Rather than temper our conclusions, we instead***
***add text to the Discussion that clearly recognizes that most infectious agents did not result in***
***increased predation risk, which is expected (L359-367); we agree this is an important addition that we***
***overlooked. Please note that infection does not equate into disease (all of us, and animals, have***
***several infectious agents in our systems, but disease is only experienced at specific agent-load levels).***
***So our results demonstrate that most infections we saw did not result in increased predation risk,***
***rather than disease. We also add text in the Discussion (L338-358) to place some of the new results***

***(see response to comments below) in context, including increased discussion of other pathogens.***

Specific comments:

Comment #2: Line 124 – Was using fish in the best digestion condition necessary to prevent degradation
of the disease DNA? Could this have influenced your analysis later where you compared condition of the
consumed and non-consumed fish?

***Response #2: Yes, we wanted to minimize the potential degradation of the infectious agent by***
***selecting the best digestion individuals. Yes, it is possible that even worse condition individuals could***
***have experienced further degradation than we observed. We discuss the potential impacts of our***
***sampling methodology, and potential biases due to degradation, (L388-405 and 458-487).***

Comment #3: Line 195-198 – Based on the discussion that we've been having; it is apparent that this is a
complex dataset that requires some careful consideration in how it is analyzed. I appreciate that the
authors put the effort into attempting the mixed effects model, unfortunately without success.

However, I still don't believe that individual Fisher's exact test for each tissue and year is the correct way
to analyze these data. The main problem with the GLM that I suggested appeared to be the year effect,
due to the singularity issue. I will list what I think is required at a minimum and then make some
additional recommendations for some added complexity that I think would improve the analysis:

1) At a minimum the authors should fit a logistic regression where the response is whether or not a fish
was predated and the covariates are: a) whether or not that fish was infected with the single disease
(e.g. IHNV) the authors want to test, b) fish length, c) the tissue (gill or liver) tested, d) a year effect for
the diseases that occur over multiple years. Fish length needs to be included in this GLM, rather than
using a second t-test later as the authors currently do. Fitting one model to test whether disease
increases predation risk and a second to test if there is an effect of length on predation is inappropriate,
because any results from these tests will give you false precision since you are doing two separate tests
and assuming independence between them. However, it is the same fish getting eaten, so they cannot
be independent. If the authors have further difficulty getting the models to converge, I highly encourage
them to consult with a statistician or quantitative ecologist that can help them through the analysis.
2) It might also be interesting to try to fit a model that includes multiple diseases, but I recommend that
the authors only include the most prevalent diseases (e.g. *Candidatus Branchiomonas cisticola*,
*Flavobacterium psychrophilum*, *Ichthyophthirius multifiliis*, *Infectious hematopoietic necrosis virus*,
*Pacific salmon parvovirus*). That will greatly reduce the number of parameters in your model, since it
appears fairly obvious without using statistics that none of the other diseases will come out as
significant.

3) Another option, if you did want to show the effect of all disease on predation risk, would be to fit a
multivariate GLM. But that would be considerably more complex and not necessary for your purposes.

***Response #3: We have conducted additional analyses. However, they are not exactly as prescribed by***
***the reviewer, for the reasons we discuss below (as well as in the Methods; L119-217, Results L250-263,***
***and Discussion L338-358 and 426-434. The reviewer requested both a global model including both***
***years "for diseases that occur over multiple years" but also to "try a model that includes multiple***
***diseases." Even here, it is unclear how many models the reviewer actually wants presented, and***
***recognizes the complexity of our data. This also speaks to (as noted in our previous revision and***

*response to reviewers, and by the reviewer above) that because some pathogens are only found in one*
*year or the other, it is difficult to assess multiple pathogens simultaneously AND include year as a*
*covariate. It's also unclear what adding year as a covariate would provide beyond our current*
*analyses (the reader can easily assess the relative impacts of an agent on predation risk, and overall*
*prevalence, between both years).*

- *Including tissue is nonsensical as an explanatory variable for models attempting to explain probability of predation (predation binary as response variable). The coefficients from this value would simply reflect the number of samples taken for each predation group for each tissue. It would not reflect differences in the relationship between predation probability and infectious agents between tissues without including an interaction (agent1 * tissue), and given our sample size, we cannot include interactions between each agent and tissue.*
 - *Only including the "most prevalent" pathogens is also not a sound a priori modeling decision. The most virulent pathogens generally occur at lower prevalences (because in many cases, except at extremely high host densities, hosts perish rapidly and are unable to pass on the infection). As stated in our paper, there is other work pointing to IHNv affecting survival, with population-level prevalence rates at <15%. In reality, the fact that a given pathogen occurs at high prevalence is a likely indication that it is NOT virulent. Furthermore, it is simply not good science to hand-pick the pathogens to assess, particularly when the reviewer wants a more comprehensive analysis than what we have presented previously.*
 - *Similarly, although adding FL is a good idea to a modeling framework, this only acts to assess the independent impact of fish length on predation risk – understanding how the relationship between predation ~ FL is affected by pathogens would require interaction terms (which due to sample size, we cannot explore) or further post-hoc assessments such as those we provided (size distributions of infection-positive vs infection-negative fish).*

In light of this, while also attempting to provide a more comprehensive analysis as requested by the reviewer, we added the following generalized linear modelling (GLM) framework to our paper (also described in the Methods L 199-217).

- *Four global models were constructed, one for each year-tissue combination (so 2014-gill, 2014-liver, 2015-gill, and 2015-liver)*
 - *Predation status was the response variable (as requested)*
 - *Explanatory variables included: FL and presence/absence of infectious agents. Only infectious agents that were detected at least twice in a given tissue-year combination were included (this helped ensure a large enough sample size to have faith in a result in as consistent of a manner as possible). Infectious agents that were found among all samples, predated and not, were not included (as these would thus have no impact on predation risk).*
 - *Another confounding factor in smolt lengths is smolt age. Two age classes emigrate from Chilko Lake, with Age-1 smolts constituting on average ~96% of the migrating population, while age-2 are substantially larger but make up ~4% of the migration. Thus length is confounded by age. Age 2 fish were only sampled in 2014, with 8 of the 32 predated smolts being age-2 (no control fish were age 2). Thus, age-2 smolts were removed from 2014 GLM analyses, as they were only present in the predated group (and thus age and FL were confounded).*

- ***We used all subsets regression to rank candidate models via AICc. But to prevent overfitting due to our small sample sizes, the maximum number of parameters in each candidate model was limited to three (not including the intercept).***

Overall, these models still identified the main result – that IHNv strongly increased predation risk. However, some other interesting results emerged, including smaller smolts at higher risk of predation, and potential increase in predation risk associated with Ichthyophthirius multifiliis. Please see our new Results (L250-258) and Table 2) and Discussion (L338-358) on these topics. These models do represent an improvement to the paper. However, we feel these analyses work best in addition to, rather than in replacement of, our former results. This is largely due to the inability to include all pathogens within global models (and given this is the broadest published screening of infectious agents in this population to-date, it is important to publish the prevalence rates and odds-ratio associated with predation in a straightforward manner) and that we had to do further subsetting of the data to run the models.

Comment #4: Lines 220-224: See my recommendation above about testing the effects of length on IHN infection. If the authors want to disentangle the effect of fish length and disease on predation rates, these need to be included in the same model. Currently, the authors are testing the hypothesis that there is no difference in length between IHN infected fish. But, it is still possible that the consumed IHN fish were smaller than all other fish.

Response #4: Please see our new GLM analyses and response to the broader comment. We do see evidence of size-based selection, but it still appears that this effect is independent of IHNv infection (which is logical, based on the speed at which IHNv causes disease, as described in our paper). Please note that even the reviewer's suggested modelling framework would have not identified if "consumed IHN+ fish were smaller than all other fish" without including an interaction term, which our study sample size simply would not allow.

Comment #5: Line 275-280: It's unclear to me how these tests differs from the tests the authors describe on lines 220-224.

Response #5: We are confused by this comment, because lines 220-224 referred to comparisons of fish length, while lines 275-280 referred to comparisons of fish condition. No changes to the text have been made from this comment.

Comment #6: Lines 302-316: I think somewhere in here you should comment on the differences between IHNv and the other diseases. You observed an increased risk of predation with IHNv, but not with any of the other infections, based on what you know of these diseases, can you formulate some hypotheses about why you observed those results?

Response #6: In our Discussion, we do have a paragraph describing why IHNv is unique – in terms of its ability to infect, cause disease, and affect mortality of juvenile sockeye salmon. Simply, IHNv has long been known to cause acute disease and mortality, particularly in juvenile salmonids, relative to many of the other infectious agents we screened (L323-327). The infectious agents we screen are quite diverse, and thus should not be expected to behave similarly (some are viruses, others bacteria, others parasites). However, in response to this comment as well as a previous one, we have added text in the Discussion to clearly acknowledge that most infectious agents do not cause an increase in predation

**risk (L359-367).**

Comment #7: Lines 330-331: Specifically, you provide evidence that infection with one specific disease
can increase risk to fish in the wild. In fact, two other diseases, that appear to have higher prevalence in
your samples and in the system, didn't have any impact on predation. It seems like you are ignoring that
result to focus on the single positive result that you had. I find it really interesting that there appear to
be some diseases that don't increase the risk of predation. I think that dichotomy, that some diseases do
increase the risk of predation while some may not, should be addressed in the discussion.

***Response #7: It is important to make the clear distinction between an infection and disease (see L55-***
***58 and L359-360). Infection is simply when a pathogen (something that could cause disease) is***
***present. Infection can occur without disease (similar to how many with COVID19 are asymptomatic).***
***Every animal has several infections at any given time, but that does not mean they are diseased.***
***Disease is when an organism's function is affected by the presence of an infection. Although our use of***
***VDD genes allows us to identify potential smolts that are experiencing disease, the reviewer here is***
***focusing our prevalence rates of infections. It is not surprising, rather expected, that infectious agents***
***can be present without increasing predation risk. Particularly, when infectious agents are at very high***
***prevalence rates (90+% as we observe in the couple pathogens noted by the reviewer), is highly likely***
***they do not cause disease in that given host (unless we were witnessing an epidemic before our eyes)***
***or at least not strong enough disease to impact survival (think of the common cold). We have added***
***text to recognize that most infectious agents will not increase predation risk (L359-367).***

Comment #8: Lines 430: Again, you are overselling your results a little. You didn't find that 'specific
infections can be associated with higher predation risks', but rather that a single infection was
associated with a higher predation risk while multiple others were not.

***Response #8: We do not understand this comment. IHNv is a 'specific infection' – we do not claim that***
***many or all infections result in increased predation risk. We did not edit the text based on this***
***comment. In addition, the new models requested by the reviewer suggest at least one other pathogen***
***could be linked to predation.***

Comment #9: Table 1: I appreciate that the authors added the extra information that was requested, but
that generally requires adjusting the table to accommodate the additional information. This table is now
a little difficult to comprehend with the way it is arranged. They should to play around with formatting
to make it easier for the reader to digest.

***Response #9: We have made additional adjustments in Word (changing column widths throughout,***
***further reducing font size) but indeed a lot of information was asked for. We are hopeful that further***
***organization can be done at the typesetting phase, if we are fortunate enough to publish.***

Comment #10: Table 1: Do you have sample sizes for these other studies? Are there any confidence
intervals for these prevalence rates?

***Response #10: We have decided to replace info from these others studies, that felt awkward, with***
***results from provincial screening of infectious agents in this population of juvenile sockeye salmon***
***smolts from mixed tissues (via the Strategic Salmon Health Initiative). We were able to acquire these***
***data between the previous revision and now, and permission to use here. See amendments to Table 1.***
***We include ranges of sample sizes in the Table caption. However, due to space constraints (already***

***noted by the reviewer in Comment #9), we did not include confidence intervals of these prevalence***
***rates (but with proportions these can be calculated from the sample size).***

Reviewer: 1

Comments to the Author(s)

Comment: well done- really enjoy this paper

***Response: We are glad that someone did enjoy the paper. Thank you for your continued support of***
***this paper and seeing value in it.***

Appendix D

Associate Editor Comments to Author:

This paper represents something of a tricky call for the editors. On the one hand, it seems clear the authors are doing their best to meet the concerns raised by the referee, but the referee has a number of outstanding concerns regarding the statistical treatment of work. As the authors have had a number of opportunities to revise, and the referee has - likewise - had a number of opportunities to review, it is not clear how productive continued review-revise-review is going to be. Instead, we are going to make the call that the authors should do what they can to address the remaining concerns in a final revision, and this revision will be assessed by the editors alone - if the latter are satisfied that the paper is publishable, it will be accepted for publication: any remaining concerns that the reviewer and the wider community may have at this stage can then be discussed openly with the paper and data accessible to all. The editors thank the reviewers for their support and the authors for their engagement with the process.

Response: We appreciate the editor reviewing this manuscript one final time. We appreciate the editor acknowledging how much we have done in attempt to assuage the one reviewer with remaining concerns. We have taken efforts to address these final comments by the reviewer, including running GLMs again and using a model averaging approach. All line numbers referred to in our response below correspond to the track-changes document (rather than the "clean" version).

Reviewer comments to Author:

Reviewer: 2

Comments to the Author(s)

Comment #1: Now that the GLM has been conducted, I don't think the individual Fisher exact tests are needed. I think the results from these tests are repetitive and simply serve to confuse the readers. My guess is that the author's want to include these tests to highlight the odds ratios; however, as I mention in the attached file, odds ratios can be easily calculated by exponentiating the coefficients of a logistic regression.

Response #1: Although we understand the reviewer's point, we feel strongly that the Fisher exact tests still need to remain in the paper. As described in our previous response (Response #3 to this reviewer on the last revision), we have several reasons for this, that we will expand upon here.

- ***As recognized by the reviewer, we are unable to include all pathogens within global models (due to not being observed in all tissue-predation status combinations). This includes some of the most prevalent pathogens that we feel strongly are important to report. We also had to subset data further by age for models to conform. Thus the GLMs do not represent all of the samples we assessed. Collectively, these samples represent the broadest screening of pathogens in sockeye salmon smolts to date, and thus including all of these data are important.***

- ***We think presenting both the fisher exact tests and the GLM results actually will act to improve reader confidence in the results (our story is largely the same, regardless of statistical analysis), rather than confuse. Given that our data are low in power, and that subsetting/exclusion was needed for GLMs, it's nice to show that our main effect (IHNv associated with predation) stands out regardless of approach.***
- ***Although the story is largely the same regardless of approach, having both statistical approaches also helps provide some nuance. When including multiple binary explanatory variables in logistic regression (i.e. presence/absence of multiple pathogens), the coefficient for each variable is dependent upon others (what is the relationship of predation and a pathogen when all other pathogens are absent or = 0). In the case of Flavobacterium psychrophilum, this resulted in some negative coefficients in 2014 models, due to small sample size of the agent when IHNv was absent. Providing the total counts (without subsetting for GLMs) and the overall odds ratios make it clear that in general, this pathogen was more prevalent in predated samples, complementing the complexities of the GLM.***
- ***Lastly, we still feel there is value in providing the odds ratios and exact test significance in a straightforward manner for those interested in fish disease ecology. Although GLM coefficients can be used to calculate odds ratios as suggested by the reviewer, it is still another step required of the reader that can result in error. And given that GLM coefficients vary (slightly) among models for the same data (see our responses below), it makes sense to provide an odds ratio this is consistently calculated for each pathogen, independent of other effects (which then the GLM results complement well).***

Comment #2: There were some obvious problems with the coefficients of the top logistic regression models that were presented. Some of the covariates in these models have coefficients over 15, meaning they had odds ratios over 3 million!! This is obviously unrealistic. After doing a little investigating by looking at Table 1, I realized all these covariates with large coefficients either had 0% or 100% predated or not predated. That means there were either no values in the numerator or denominator of the odds ratio (just like you couldn't calculate the odds ratio for those diseases in those tissues in those years). Since the coefficient estimates in the logistic regression of the logs of the odds ratio, the coefficient estimates for these covariates aren't realistic. In other words, for your logistic regression model, you can't include any of the diseases for any of the tissues in any year that you couldn't calculate an odds ratio in table 1.

Response #2: Indeed, coefficients are difficult to interpret when a pathogen-status, predation status combination contained a zero (which occurred in both rare as well as common pathogens, the latter of which could have zero negatives in either a predated or non-predated group). We were attempting to follow the reviewer's suggestion, of including the most prevalent pathogens. We also were careful in our language of the Results and Discussion to downplay model results for pathogens with unrealistic coefficients. In response this comment, however, we have re-run all of the models (Table 2), and only include pathogens that had at least one sample in each predation status-infection-status combination. The main result is still the same – IHNv presence increases predation risk. However, there are subtle differences, such that we now provide brief discussion on Flavobacterium psychrophilum and Pacific salmon parvovirus (but we caution overinterpreting these results as they are inconsistent). We also removed text on Candidatus Branchiomonas cysticola as this infectious agent no longer is entered in

models for assessment. As mentioned in our previous version of the paper, we do not expect this agent to affect survival as it is pretty ubiquitous and in our group's work has never had links to mortality. We have edited the Methods (L211-215) to reflect these changes in the modelling as well as the Results (L260-267) and Discussion (L350-364) as needed.

Comment #3: The best practice for model selection isn't to just interpret the top model, but to either use model averaging or to pick the most parsimonious model from your top model set. Things may change after you modify which diseases to include in your models, but, currently most of your top models are subsets of one of the top models (i.e., they include all the same covariate plus some some additional ones). If that continues to be the case, you should just use that most parsimonious model as your top model.

Response #3: In response to this comment, we averaged top-ranked models (those with Delta AICc < 3) and present these in Table 2. We also added to Table 2 the AICc weight, which represents the proportion (probability) that the model is the best among the candidate set to increase interpretability, which helps further interpret models. We have done AICc weight among all models as well as recalculated for models with Delta AICc <3, which were used for model averaging. We have also added additional text to mention some of the other variables that were included in high-ranking models but not the top model for each tissue-year combination (Results L260-267 and Discussion L350-364) to fully acknowledge that there are other variables that could contribute to predation risk beyond those in the top-ranking model.

In regards to "picking the most parsimonious model from your top model set" we are a bit confused. We have ranked models based upon a measure of parsimony – AIC – that is an information criterion commonly used to assess model parsimony in ecology and biology (rather than an ad-hoc approach as suggested by the reviewer). We are not familiar with an approach where an information criterion is used to run models, and then subsequently use a separate ad-hoc approach to pick a different "top" model within that subset.